# Antibody-mediated neutralization of galectin-3 as a strategy for the treatment of systemic sclerosis

Céline Ortega-Ferreira[1,15], Perrine Soret[2,15], Gautier Robin [ORCID][3], Silvia Speca [ORCID][4,5], Sandra Hubert[6], Marianne Le Gall[3], Emiko Desvaux [ORCID][6], Manel Jendoubi[4,5], Julie Saint-Paul [ORCID][3], Loubna Chadli[7], Agnès Chomel[8], Sylvie Berger[9], Emmanuel Nony [ORCID][8], Béatrice Neau[10], Benjamin Fould[8], Anne Licznar[11], Franck Levasseur [ORCID][11], Thomas Guerrier [ORCID][4,5], Sahar Elouej[12], Sophie Courtade-Gaïani [ORCID][12], Nicolas Provost[13], The Quyen Nguyen[9], Julien Verdier[6], David Launay [ORCID][4,5,14] & Frédéric De Ceuninck [ORCID][6] ✉

Systemic sclerosis (SSc) is an autoimmune, inflammatory and fibrotic disease with limited treatment options. Developing new therapies is therefore crucial to address patient needs. To this end, we focused on galectin-3 (Gal-3), a lectin known to be associated with several pathological processes seen in SSc. Using RNA sequencing of whole-blood samples in a cross-sectional cohort of 249 patients with SSc, Gal-3 and its interactants defined a strong transcriptomic fingerprint associated with disease severity, pulmonary and cardiac malfunctions, neutrophilia and lymphopenia. We developed new Gal-3 neutralizing monoclonal antibodies (mAb), which were then evaluated in a mouse model of hypochlorous acid (HOCl)-induced SSc. We show that two of these antibodies, D11 and E07, reduced pathological skin thickening, lung and skin collagen deposition, pulmonary macrophage content, and plasma interleukin-5 and -6 levels. Moreover, E07 changed the transcriptional profiles of HOCl-treated mice, resulting in a gene expression pattern that resembled that of control mice. Similarly, pathological pathways engaged in patients with SSc were counteracted by E07 in mice. Collectively, these findings demonstrate the translational potential of Gal-3 blockade as a therapeutic option for SSc.

Galectin-3 (Gal-3) is a β-galactoside-binding lectin located in the nucleus, cytoplasm, and cell surface, but also secreted to the extracellular matrix[1]. Gal-3 exerts its biological and pathological actions, including chemoattraction and cell adhesion (membrane-associated form), cell differentiation, and apoptosis (intracellular form), through its association with several glycoprotein partners[1–5]. Under physiological conditions, extracellular Gal-3 provides essential structural and biochemical scaffolding support to surrounding cells. In pathological states, extracellular and membrane-associated Gal-3 exhibit proinflammatory and profibrotic actions, stimulating the chemoattraction of immune cells, the production of proinflammatory cytokines, the activation of fibroblasts, and the overproduction of collagen[5–8]. A pathogenic role for Gal-3 was proposed in systemic sclerosis (SSc) due to its known association with different hallmarks of this disease[9,10], namely pulmonary, cardiac, renal, and cutaneous fibrosis, inflammation, and pulmonary arterial hypertension[5,7,11]. High circulating Gal-3 levels are associated with clinical manifestations of SSc including pulmonary and peripheral vasculopathy[12,13], adverse digestive effects[13], and cardiopathy[14]. High Gal-3 levels in patients with SSc were also proposed as a prognosis factor of all-cause and cardiovascular mortality[15].

One important target organ for evaluating the potential of Gal-3 inhibitors in patients with SSc is the lung. SSc-related interstitial lung disease (SSc-ILD) is the primary cause of death in these patients[16,17]. Increased Gal-3 levels are associated with lung fibrosis, as exemplified in the bronchoalveolar lavage fluid and the plasma of patients with idiopathic pulmonary fibrosis (IPF)[18,19]. These levels also rise sharply during acute exacerbation of the disease[19], and Gal-3 inhibition has thus been suggested as a potential therapeutic target in these patients. Preclinical models have confirmed Gal-3 as an important regulator of lung fibrosis. Bleomycin-induced lung fibrosis is dramatically reduced in Gal-3 deficient mice[19]. TD139, an inhaled Gal-3 inhibitor, blocked the progression of lung fibrosis in a mouse bleomycin model, with a dose–response profile comparing favorably with that of the anti-fibrotic compound pirfenidone on total collagen deposition, and the inflammatory and fibrosis scores[20]. In a phase I/IIa study, compared with healthy controls, TD139 treatment reduced Gal-3 expression in alveolar macrophages from IPF patients and reduced plasma bio-markers relevant to lung fibrosis such as PDGF-BB, PAI-1, CCL18, and CHI3L1[21]. Altogether, these data provide a proof of principle for Gal-3 inhibition as a valuable strategy to improve lung function in SSc-ILD.

In addition, Gal-3 is involved in multi-organ fibrosis, inflammation, and vascular dysfunction, all of which strikingly reflect the clinical picture of SSc[9,10,17]. Thus, Gal-3 inhibitors may have the potential to target several features of SSc currently addressed separately by various treatments, most of them symptomatic. A pathogenic role of Gal-3 in the induction of fibrosis has been reported in several organs in addition to the lungs, including the heart, the kidneys, and the liver[5,7], prompting several preclinical and clinical studies to evaluate the therapeutic potential of Gal-3-targeting compounds in related diseases[22–27]. The deleterious action of Gal-3 is also well-established during inflammation, with the recruitment and activation of macro-phages. Gal-3 also triggers the release of cytokines and other immune signaling molecules causing organ damage and dysfunction. Furthermore, Gal-3 inhibitors were demonstrated to alleviate pulmonary vascular hypertension in different preclinical models.

Our current work shows that Gal-3 and its interactants define a strong transcriptomic fingerprint associated with SSc, highlighting the relevance of this global network as a possible target for treatment. For this purpose, we have developed blocking monoclonal antibodies, demonstrating unambiguous efficacy on several fibrotic, inflammatory, and immunological features in a relevant preclinical SSc model.

## Results

### Gal-3 signature is associated with disease severity, lung, and cardiac impairment within molecular clusters of patients with SSc

Three functional molecular clusters were identified within a cohort of patients with SSc enrolled in the PRECISESADS cross-sectional study (NCT02890121). The global characteristics of all patients and healthy volunteers (HV) are summarized in Table 1. To perform the clustering of the 249 samples obtained from patients with SSc, whole-blood transcriptomic data were analyzed using a selection flowchart described previously[28] (Supplementary Figs. 1 and 2). The analysis allowed to cluster patients into three subgroups (Fig. 1a). Cluster 1 (C1) contained 86 patients (34.5%), cluster 2 (C2) 120 patients (48.2%), and cluster 3 (C3) 43 patients (17.3%). The characteristics of patients in each cluster are reported in Table 2 and Supplementary Fig. 3. C1 was enriched in limited cutaneous SSc patients (lcSSc) and showed interferon (IFN)-dependent and intermediate inflammation profile. C3 contained a larger proportion of diffuse cutaneous SSc patients (dcSSc) and displayed a more inflammatory profile than C1, with a strong expression of IFN-related genes. C3 was also associated with an increased presence of neutrophils, and B and T cells lymphopenia based on transcriptomic and flow cytometry data (Supplementary Fig. 3e, f). Regarding clinical manifestations, C3 patients showed more

pulmonary fibrosis, arrhythmia, and higher disease activity (physician global assessment) compared with C1 and C2 (Supplementary Fig. 3b–d). This was consistent with the higher proportion of dcSSc patients in this cluster, known to present a higher disease severity than lcSSc patients[9,10]. C2 was enriched in *sine* scleroderma SSc (ssSSc) patients and showed a relatively 'healthy-like' transcriptional profile

**Table 1 | Healthy volunteers (HV) and systemic sclerosis (SSc) patients characteristics**

| | | | HV | SSc |
|---|---|---|---|---|
| **Demography** | | | | |
| Age | | n | 365 | 249 |
| | | Median (Q1;Q3) | 53 (47;59) | 60 (50;68) |
| Sex | | n | 365 | 249 |
| | Female | n (%) | 307 (84) | 212 (85) |
| Race | | n | 365 | 249 |
| | Asian | n (%) | 3 (0.8) | 2 (0.8) |
| | Black/African American | n (%) | 0 (0) | 0 (0) |
| | Caucasian/White | n (%) | 360 (98.6) | 245 (98.4) |
| | Native Hawaiian/ Pac. Isl. | n (%) | 0 (0) | 1 (0.4) |
| | Other | n (%) | 2 (0.5) | 1 (0.4) |
| **Autoantibodies** | | | | |
| Cent B | | n | 237 | 211 |
| | Positive | n (%) | 0 (0) | 105 (49.8) |
| SCL70 | | n | 238 | 208 |
| | Positive | n (%) | 1 (0.4) | 79 (38.0) |
| SSA | | n | 238 | 207 |
| | Positive | n (%) | 6 (2.5) | 36 (17.4) |
| SSB | | n | 238 | 211 |
| | Positive | n (%) | 1 (0.4) | 4 (1.9) |
| U1 RNP | | n | 238 | 208 |
| | Positive | n (%) | 12 (5) | 17 (8.2) |
| **Clinical features** | | | | |
| Pericarditis | | n | - | 232 |
| | No | n (%) | - | 225 (97.0) |
| | Past | n (%) | - | 5 (2.2) |
| | Present | n (%) | - | 2 (0.9) |
| Systemic hypertension | | n | - | 249 |
| | Yes | n (%) | - | 80 (32.1) |
| Pulmonary fibrosis | | n | - | 231 |
| | Yes | n (%) | - | 86 (37.2) |
| Skin fibrosis | | n | - | 248 |
| | No | n (%) | - | 74 (29.8) |
| | Past | n (%) | - | 5 (2.0) |
| | Present | n (%) | - | 169 (68.1) |
| DLCO | | n | - | 126 |
| | % predicted value | Median (Q1;Q3) | - | 64 (49 ;75) |
| Forced vital capacity | | n | - | 131 |
| | % predicted value | Median (Q1;Q3) | - | 94 (79;109) |
| SSc classification[a] | | n | - | 141 |
| | dcSSc | n (%) | - | 49 (34.8) |
| | lcSSc | n (%) | - | 64 (45.4) |
| | ssSSc | n (%) | - | 28 (19.8) |
| **Treatment** | | n | - | 249 |
| Hydroxychloroquine | Yes | n (%) | - | 21 (8.4) |
| Immunosuppressants | Yes | n (%) | - | 65 (26.1) |
| Biologicals | Yes | n (%) | - | 7 (2.8) |
| Steroids | Yes | n (%) | - | 59 (23.7) |
| Systemic antibiotics | Yes | n (%) | - | 5 (2.0) |

*DLCO* diffusing capacity of the lungs for carbon monoxide, *HV* healthy volunteers, *Pac. Isl.* Pacific islanders, *dcSSc* diffuse cutaneous SSc, *lcSSc* limited cutaneous SSc, *ssSSc* sine scleroderma SSc.

[a]SSc classification was performed on 141 out of 249 patients. *n*, number of patients with available information.

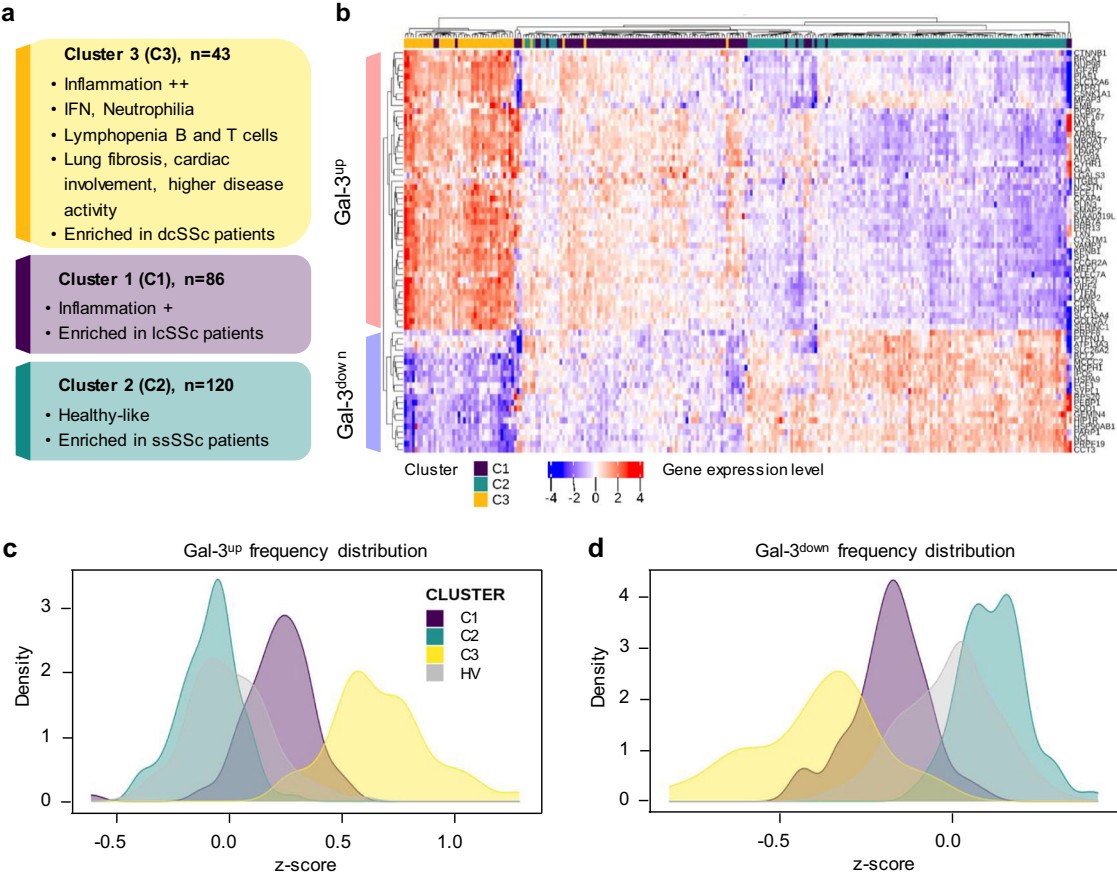

**Fig. 1 | Molecular characterization of SSc patient clusters and their Gal-3 fingerprint. a** Description of patient clusters C1, C2, and C3 characteristics based on their main molecular pathways and clinical features. **b** Gene expression heatmap of the top 69 Gal-3 interactants allowing to discriminate SSc patients clusters from each other (C3, yellow: *n* = 43; C1, purple: *n* = 86; C2, turquoise: *n* = 120). Gene expression intensities are displayed as colors ranging from blue for the lowest

expression level to red for the highest expression level as shown in the key. Each individual subject is represented by one vertical line. Differentially expressed Gal-3 interactants are represented by individual horizontal lines and listed on the right. **c**, **d** Frequency distribution plots of Gal-3$^{up}$ (**c**) and Gal-3$^{down}$ (**d**) scores for each SSc patient cluster and healthy volunteers (HV). *dcSSc, lcSSc, and ssSSc*, diffuse cutaneous, limited cutaneous, and sine scleroderma systemic sclerosis, respectively.

(Supplementary Fig. 3e), consistent with the fact that these patients are characterized by milder disease manifestation and lack of skin involvement, despite the presence of lung or cardiac malfunction[9].

To investigate a potential association between the Gal-3 network and SSc, we first established a list of known Gal-3 partners through bioinformatic queries within the Ingenuity Pathway Analysis (IPA) and GPS-Prot databases and established a non-redundant list of 307 interactants (Supplementary Data 1, column A) of which 248 could be retrieved from whole-blood RNA sequencing (RNAseq) data from the PRECISESADS SSc cohort, after exclusion of 59 genes absent from the dataset (Supplementary Data 1, column C). Among these 248 interactants, we determined the top genes allowing to discriminate patient clusters from each other, resulting in a Gal-3 fingerprint of 69 interactants represented on the heatmap in Fig. 1b. This Gal-3 fingerprint was represented by 48 upregulated genes (named Gal-3$^{up}$) and 21 downregulated genes (named Gal-3$^{down}$). For each cluster, Gal-3$^{up}$ and Gal-3$^{down}$ scores were calculated and normalized against the HV population (Fig. 1c, d). C3 patients expressed the highest Gal-3$^{up}$ and lowest Gal-3$^{down}$ scores, each almost fully discriminated from the HV z-score, whereas the score distribution of C2 patients overlapped with that of HV. C1 patients showed intermediate Gal-3$^{up}$ and Gal-3$^{down}$ scores, consistent with the intermediate disease severity and inflammation profile observed in this subgroup. The importance of the Gal-3 fingerprint was investigated further by examining the relationship between the Gal-3$^{up}$ and Gal-3$^{down}$ scores with features of the disease (Fig. 2a–c). Gal-3$^{up}$ scores were higher in dcSSc than in lcSSc and ssSSc

patients ($p$ = 0.031) and conversely, Gal-3$^{down}$ scores were lower in dcSSc patients ($p$ = 0.030) (Fig. 2a). Similarly, the Gal-3$^{up}$ scores were higher in patients with impaired lung and cardiac manifestations such as pulmonary fibrosis ($p$ = 0.029), worsening lung function ($p$ = 0.025), basilar crackles ($p$ = 0.0006) and arrhythmia ($p$ = 6.5 × 10$^{-6}$), whereas the Gal-3$^{down}$ scores were lower for these parameters ($p$ = 0.019, $p$ = 0.007, $p$ = 0.002, and $p$ = 0.001, respectively) (Fig. 2b). Overall, this Gal-3 fingerprint showed that high expression levels of Gal-3 interactants were strongly associated with impaired vital organ function in patients with SSc as well as with disease severity. Furthermore, the Gal-3$^{up}$ and Gal-3$^{down}$ scores were highly associated with changes in some specific immune cell populations in patients with SSc (Fig. 2d). The Gal-3$^{up}$ fingerprint positively correlated with the number of neutrophils and inversely correlated with both B and T lymphocytes. As expected, a mirror image was found for the Gal-3$^{down}$ fingerprint. The neutrophil-to-lymphocyte ratio, a well-recognized marker of systemic inflammation[29], was strongly correlated with the Gal-3$^{up}$ score and inversely correlated with the Gal-3$^{down}$ score (Fig. 2e), highlighting the association of Gal-3 and its partners in this target SSc population. In addition to highlighting the association of Gal-3 and its interactants in SSc, this Gal-3 fingerprint could also serve as a stratification biomarker to discriminate patients based on disease features and/or inflammatory status in a targeted treatment approach.

In a second approach, we aimed to identify which genes representative of the Gal-3 interactome were differentially expressed in patients with SSc in comparison with HV. By itself, Gal-3 expression was

**Table 2 | Descriptive analysis of SSc patients characteristics by clusters**

| | | Cluster 1 (C1) | Cluster 2 (C2) | Cluster 3 (C3) | *p*-value |
|---|---|---|---|---|---|
| **Demography** | | | | | |
| Age, years | *n* | 86 | 120 | 43 | |
| | Median (Q1;Q3) | 64 (53;73) | 58 (43;66) | 62 (52;69) | <0.001 |
| Sex | *n* | 86 | 120 | 43 | |
| Female | *n (%)* | 72 (83.7) | 109 (90.8) | 31 (72.1) | 0.014 |
| Race | *n* | 86 | 120 | 43 | 0.604 |
| Asian | *n (%)* | 0 (0) | 1 (0.8) | 1 (2.3) | |
| Black/African American | *n (%)* | 0 (0) | 0 (0) | 0 (0) | |
| Caucasian/White | *n (%)* | 85 (98.8) | 118 (98.3) | 42 (97.7) | |
| Native Hawaiian/ Pac. Isl. | *n (%)* | 1 (1.2) | 0 (0) | 0 (0) | |
| Other | *n (%)* | 0 (0) | 1 (0.8) | 0 (0) | |
| **Autoantibodies** | | | | | |
| Cent B | *n* | 73 | 103 | 35 | |
| Positive | *n (%)* | 39 (53.4) | 58 (56.3) | 8 (22.9) | 0.002 |
| SCL70 | *n* | 72 | 101 | 35 | |
| Positive | *n (%)* | 21 (29.2) | 34 (33.7) | 24 (68.6) | <0.001 |
| SSA | *n* | 72 | 100 | 35 | |
| Positive | *n (%)* | 11 (15.3) | 17 (17) | 8 (22.9) | 0.572 |
| SSB | *n* | 73 | 103 | 35 | |
| Positive | *n (%)* | 1 (1.4) | 1 (1.0) | 2 (5.7) | 0.225 |
| U1 RNP | *n* | 72 | 101 | 35 | |
| Positive | *n (%)* | 6 (8.3) | 5 (5.0) | 6 (17.1) | 0.076 |
| **Clinical features** | | | | | |
| Pericarditis | *n* | 80 | 111 | 41 | 0.749 |
| No | *n (%)* | 77 (96.2) | 109 (98.2) | 39 (95.1) | |
| Past | *n (%)* | 2 (2.5) | 2 (1.8) | 1 (2.4) | |
| Present | *n (%)* | 1 (1.2) | 0 (0) | 1 (2.4) | |
| Systemic hypertension | *n* | 86 | 120 | 43 | |
| Yes | *n (%)* | 35 (40.7) | 30 (25.0) | 15 (34.9) | 0.054 |
| Pulmonary fibrosis | *n* | 78 | 110 | 43 | |
| Yes | *n (%)* | 29 (37.2) | 32 (29.1) | 25 (58.1) | 0.004 |
| Skin Fibrosis | *n* | 86 | 119 | 43 | 0.092 |
| No | *n (%)* | 21 (24.4) | 44 (37.0) | 9 (20.9) | |
| Past | *n (%)* | 1 (1.2) | 2 (1.7) | 2 (4.7) | |
| Present | *n (%)* | 64 (74.4) | 73 (61.3) | 32 (74.4) | |
| DLCO | *n* | 36 | 68 | 22 | |
| % predicted value | Median (Q1;Q3) | 61 (51;74) | 64 (51;76) | 59 (45;72) | 0.536 |
| FVC | *n* | 38 | 71 | 22 | |
| % predicted value | Median (Q1;Q3) | 94 (71;110) | 98 (86;109) | 89 (71;99) | 0.246 |
| SSc classification[a] | *n* | 43 | 76 | 22 | 0.003 |
| dcSSc | *n (%)* | 17 (39.6) | 18 (23.7) | 14 (63.6) | |
| lcSSc | *n (%)* | 22 (51.2) | 37 (48.7) | 5 (22.7) | |
| ssSSc | *n (%)* | 4 (9.3) | 21 (27.6) | 3 (13.6) | |
| **Treatment** | *n* | 86 | 120 | 43 | |
| Hydroxychloroquine | *n (%)* | 4 (4.7) | 13 (10.8) | 4 (9.3) | 0.286 |
| Immunosuppressants | *n (%)* | 23 (26.7) | 19 (15.8) | 23 (53.5) | <0.001 |
| Biologicals | *n (%)* | 3 (3.5) | 2 (1.7) | 2 (4.7) | 0.473 |
| Steroids | *n (%)* | 19 (22.1) | 11 (9.2) | 29 (67.4) | <0.001 |
| Systemic antibiotics | *n (%)* | 0 (0) | 2 (1.7) | 3 (7.0) | nc |

*DLCO* diffusing capacity of the lungs for carbon monoxide, *FVC* forced vital capacity, *HV* healthy volunteers, *Pac. Isl.* Pacific islanders, *dcSSc* diffuse cutaneous SSc, *lcSSc* limited cutaneous SSc, *ssSSc* sine scleroderma SSc. Statistical analyses: chi-square test of independence for categorical variables and Kruskal–Wallis test for continuous variables (*nc*, not calculable).
[a]SSc classification was performed on 141 out of 249 patients. *n*, number of patients with available information.

significantly increased by 35% in C1 ($p = 2 \times 10^{-17}$) and 22% in C3 ($p = 1 \times 10^{-5}$) compared to HV, and a similar increase was found at the Gal-3 protein level in plasma (+55% in C1, $p = 7.5 \times 10^{-5}$ and + 32% in C3, $p = 0.035$). No statistical difference was found between 'healthy-like' patients from C2 and HV. Among the 248 interactants studied, 103 were found differentially expressed (fold-change (FC) ≥ |1.3 | with false discovery rate (FDR) < 0.05) in at least one cluster of patients compared to HV, thus representing 41.5% of the studied Gal-3 interactants. In this clustered analysis, 99 Gal-3 interactants were found differentially regulated in C3, 25 in C1 (including 4 C1-exclusive, 20 in common with C3, and 1 in common with C2), and 3 in C2, including 2 in common with C3 and 1 in common with C1 (Fig. 3 and Supplementary Data 2). As anticipated from the cluster-to-cluster analysis, patients from C3 were associated with a higher number of differentially expressed interactants, followed by patients from C1, and then C2. Altogether, these findings support Gal-3 as a potentially relevant therapeutic target to treat patients suffering from SSc and prompted us to engage in a strategy targeting the carbohydrate recognition domain (CRD) of Gal-3, responsible for the interaction with its natural ligands[1,30].

### Development of novel Gal-3 blocking monoclonal antibodies (mAb) with human and mouse cross-reactivity

After phage display screening, 66 unique scFv binders to recombinant human Gal-3 (hGal-3) were identified, 22 of which were cross-reactive to mouse Gal-3 (mGal-3) and also bound the CRD without showing any reactivity to human Gal-1 nor Gal-7. The lead scFvs, namely D11 and E07, were selected based on their ability to bind full-length (FL) hGal-3 as well as the CRD of hGal-3, their selectivity versus hGal-1 and hGal-7, and their cross-reactivity to mGal-3. These binding characteristics are shown in Supplementary Fig. 4.

D11 and E07 mAbs were produced and expressed in hIgG1 format. Their migration profiles on SDS-PAGE under non-reduced and reduced conditions and molecular masses are shown in Supplementary Fig. 5a, b. Both mAbs were characterized by a high level of homogeneity, superior to 99% as determined by size exclusion chromatography-UV (SEC-UV) analysis (Supplementary Fig. 5c). Temperature stability analyses showed that they displayed size exclusion chromatographic behaviors similar to trastuzumab when subjected to forced degradation at 40 °C during 2 weeks (Supplementary Fig. 6a, b), and relatively comparable onset temperatures of degradation, close to 60–61 °C for D11 and E07, versus around 64 °C for trastuzumab (Supplementary Fig. 6c). As for the parent scFvs, both mAbs retained the full capacity to bind FL-Gal-3 and its CRD as determined by ELISA, demonstrating that the site of interaction occurred in this latter domain. Furthermore, they were cross-reactive to mGal-3, and showed high selectivity towards Gal-3 over Gal-1, Gal-7 (Fig. 4a), Gal-2, -4, -8, and -10 and lower selectivity over Gal-14 (Supplementary Fig. 7a). Half-maximal effective concentrations ($EC_{50}$) for binding to Gal-3 measured by ELISA were in the sub-nanomolar range, reaching 0.37 nM (D11) and 0.07 nM (E07) for hGal-3, and 0.99 nM (D11) and 0.30 nM (E07) for mGal-3 (Fig. 4b). By comparison, $EC_{50}$ values for hGal-14 were 59.9 nM and 90.6 nM for D11 and E07, respectively (Supplementary Fig. 7b). The ability of D11 and E07 to displace asialofetuin, a prototypical Gal-3 ligand, from the CRD, was examined by competition ELISA. Half-maximal inhibitory concentrations ($IC_{50}$) were in the nanomolar range, reaching 2.18 nM and 1.50 nM for D11 and E07, respectively (Fig. 4c). This result demonstrates that both mAbs efficiently interfere with the binding of natural Gal-3 ligands within the CRD, allowing to consider them as potential regulators of the Gal-3 interactome network.

The binding kinetics of D11 and E07 mAbs were further analyzed by SPR (Fig. 4d and Supplementary Fig. 8). $K_D$ values were in the same nanomolar range for FL hGal-3 or its CRD, and their high binding potency to mGal-3 was confirmed, with $K_D$ values below 5 nM. D11 and E07 also demonstrated full cross-reactivity to rat, cynomolgus, and

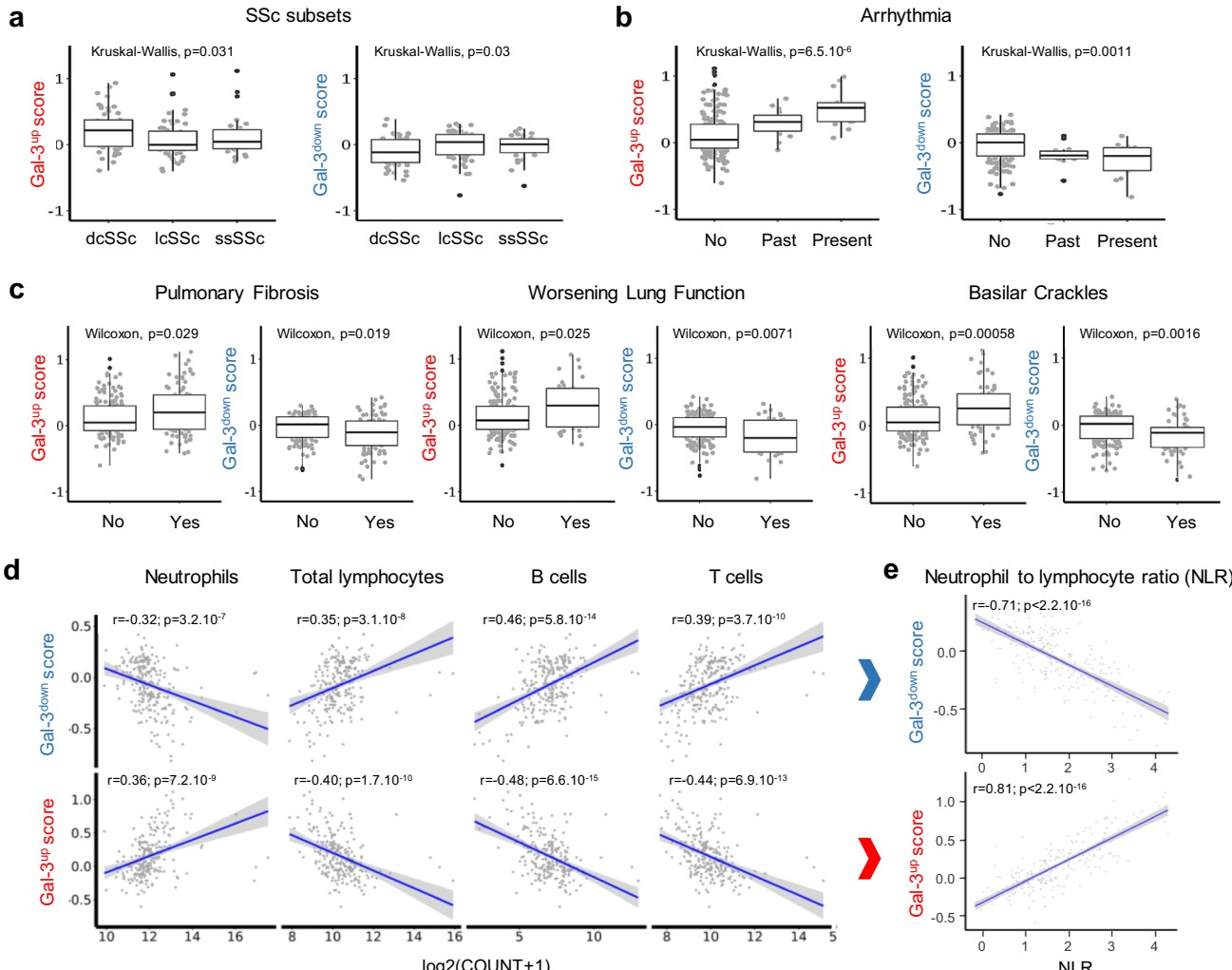

**Fig. 2 | Association between SSc patient subsets, impaired cardiac and lung functions, immune cell populations, and Gal-3 scores. a** Association between Gal-3[up] and Gal-3[down] scores and SSc patients subsets collected for 141 patients (dcSSc, diffuse cutaneous SSc: $n = 49$; lcSSc, limited cutaneous SSc: $n = 64$; ssSSc, sine scleroderma SSc: $n = 28$). **b** Association between Gal-3[up] and Gal-3[down] scores and cardiac arrythmia collected for 223 patients: no 194; past 15; present 14). **c** Association between Gal-3[up] and Gal-3[down] scores and lung dysfunction assessed by pulmonary fibrosis (data collected for 231 patients: no 145; yes 86), worsening lung function (data collected for 221 patients: no 185; yes 36), and basilar crackles (data collected for 248 patients: no 189; yes 59). Results in **a**–**c** are depicted as boxes and whiskers plots defined by minimal, maximal, and median values in each dataset. The upper whisker extends from the hinge to the largest value no further than 1.5 *

IQR from the hinge (where IQR is the inter-quartile range or distance between the first and third quartiles). The lower whisker extends from the hinge to the smallest value at most 1.5 * IQR of the hinge. Data beyond the end of the whiskers are called outlying points and are plotted individually. Statistical analyses were performed using the Kruskal–Wallis test (**a**, **b**) or the two-sided Wilcoxon test (**c**). **d** Correlation between Gal-3[up] and Gal-3[down] scores and blood immune cell populations. Data were collected from $n = 211$ patients. **e** Correlation between Gal-3[up] and Gal-3[down] scores and neutrophil to lymphocyte ratio calculated from **d**. Two-sided Pearson's correlation coefficient was used for statistical analyses performed on log-transformed data in **d** and **e**, using a linear regression model (blue line) and a 95% confidence interval represented by the gray area. All $p$-values are reported in each panel.

dog Gal-3, with $K_D$ values in the same order of magnitude as for hGal-3. Altogether, while these experiments did not allow the strict ranking of both mAbs according to their Gal-3-binding potencies, they unambiguously demonstrate that they bind to the Gal-3 CRD with high affinity and exhibit full cross-reactivity to Gal-3 of all species studied.

**Gal-3 blocking mAbs efficiently improve fibrosis and inflammation in a mouse model of HOCl-induced SSc**
The efficacy of D11 and E07 Gal-3 mAbs was evaluated on multiple readouts in the mouse model of HOCl-induced SSc and compared to TD139. Both mAbs showed good plasma exposure with similar concentrations at the end of the study with mean values of 111.7 and 101.1 μg/mL for D11 and E07, respectively (Fig. 5a). TD139 had no measurable concentrations at this time of sampling, consistent with its

low plasma half-life estimated to be about 3 h in mouse[31]. No anti-drug antibodies (ADA) could be detected in plasmas collected from mice treated with E07, and low ADA levels were measured in 5 out of 12 mice treated with D11, without any correlation with mAb plasma concentrations (Supplementary Fig. 9).

The plasma levels of Gal-3 were significantly increased by 50.2% after pathological induction with HOCl and fully reversed after D11, E07, and TD139 treatments (Fig. 5b). D11, E07 and TD139 also reduced longitudinal skin thickness, with a significant effect and gentler progressing slope starting from day 21 for D11 and E07, and maximum efficacy was reached at day 42 (Fig. 5c). All anti-Gal-3 treatments reduced the levels of skin and lung collagen content, with higher efficacy of D11 in the skin, similar to that of TD139, and higher efficacy of E07 in the lung (Fig. 5d, e). Representative images of skin and lung

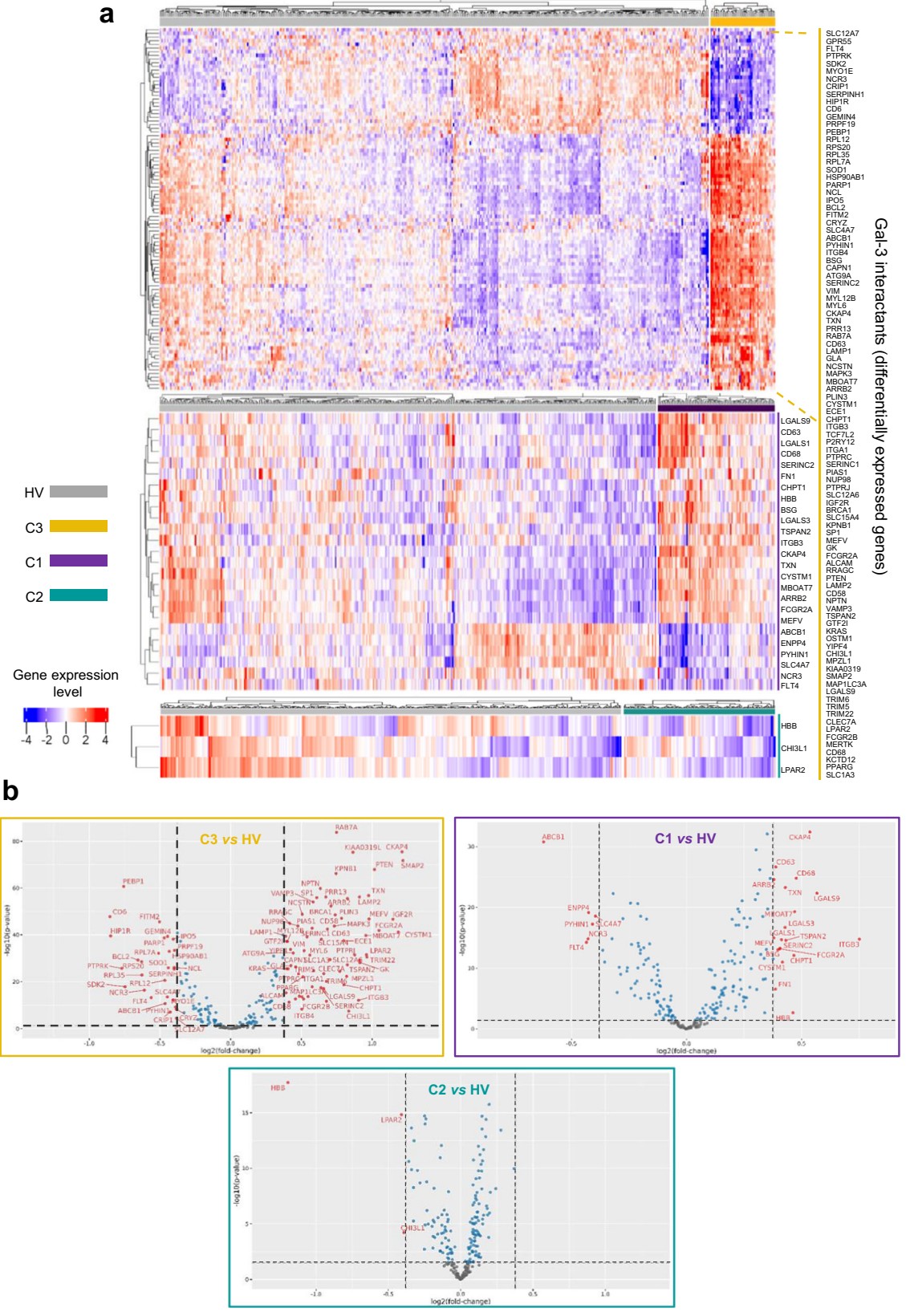

staining and collagen deposition in the different groups are shown in Supplementary Fig. 10. Immunofluorescence staining of the macrophage marker F4/80 in lungs demonstrated a 3.3-fold increase in the HOCl group compared to control mice (Supplementary Fig 11). E07 mAb and TD139 inhibited this pathological increase by 88.2% and 100% respectively. This effect was particularly visible on interstitial

macrophages after treatment with E07 mAb and TD139. Among the panel of 10 tested cytokines, IL-5 and IL-6 were the most affected by anti-Gal-3 treatments (Fig. 5f, g). Although IL-5 levels did not increase after HOCl induction, they were significantly reduced after D11, E07, and TD139 treatments, suggesting that these treatment conditions acted independently of the pathological condition. IL-6 levels were

**Fig. 3 | Gene expression heatmap and volcano plots of Gal-3 interactants differentially expressed between each SSc patient cluster and healthy volunteers (HV). a** Heatmap, gene expression intensities are displayed as colors ranging from blue for the lowest expression level to red for the highest expression level as shown in the key. Healthy volunteers (HV) are represented by the gray bar, and patients of the C3 ($n = 43$), C1 ($n = 86$), and C2 ($n = 120$) clusters are represented by the yellow (top figure), purple (middle figure) and turquoise (bottom figure) bars, respectively. Each individual subject is represented by one vertical line. Differentially expressed Gal-3 interactants are represented by individual horizontal lines and listed on the right. **b** Volcano plot representations of Gal-3 interactants

differentially expressed between C3 and HV (upper left panel), C1 and HV (upper right panel), and C2 and HV (bottom panel). The horizontal dotted line indicates the $p$-value cut-off (FDR 1.3). The vertical dotted lines indicate the fold-change cut-off values (minus 1.3 and plus 1.3). Gray dots represent the genes below the $p$-value and fold-change cut-off values; blue dots represent the genes fulfilling the $p$-value criterion but not the expected fold-change cut-off; red dots represent genes fulfilling all statistical requirements and correspond to genes identified as 'differentially expressed'. The detailed list of Gal-3 interactants differentially expressed between each cluster and HV is presented in Supplementary Data 2.

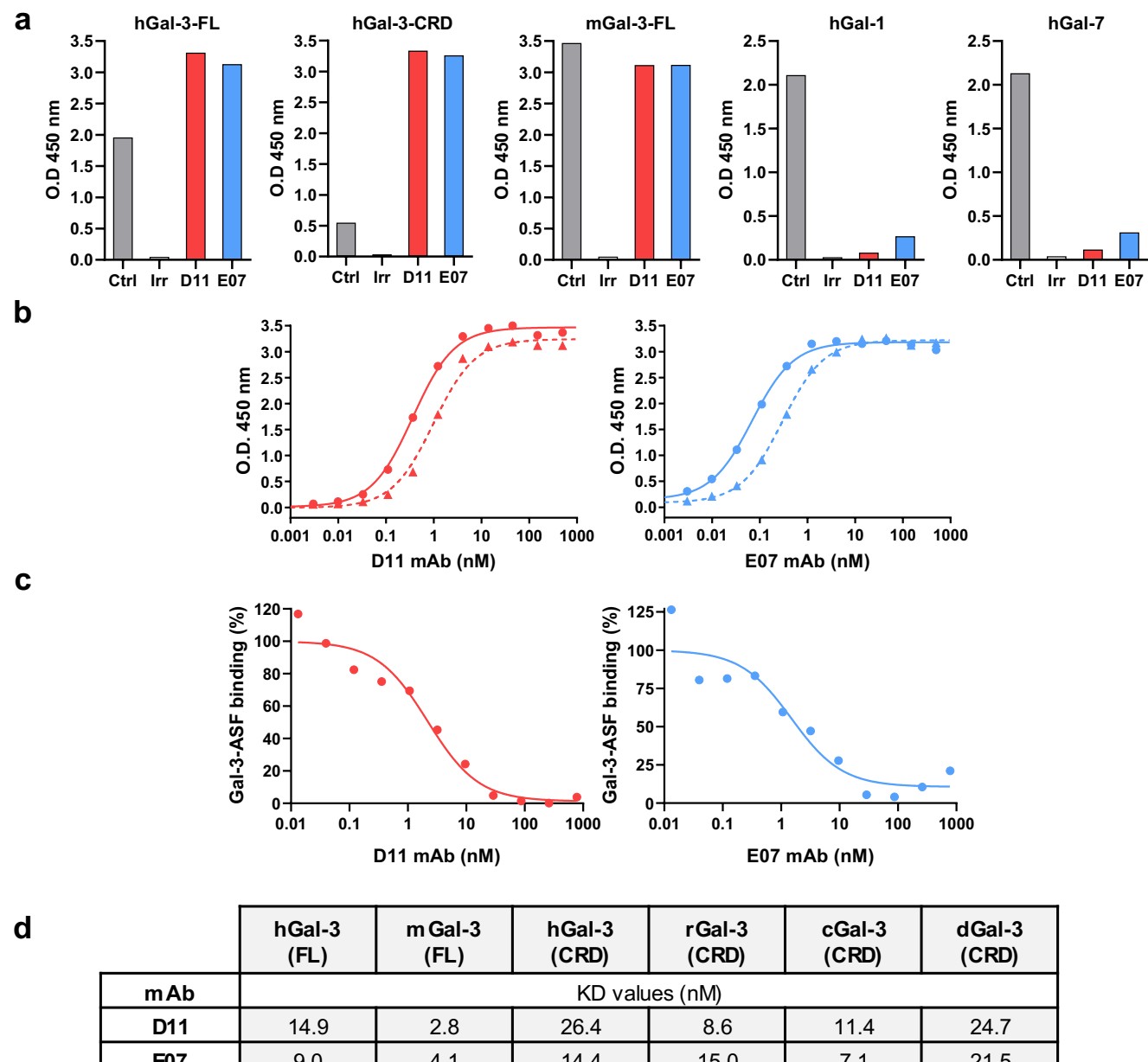

| mAb | hGal-3 (FL) | mGal-3 (FL) | hGal-3 (CRD) | rGal-3 (CRD) | cGal-3 (CRD) | dGal-3 (CRD) |
|---|---|---|---|---|---|---|
| | KD values (nM) | | | | | |
| D11 | 14.9 | 2.8 | 26.4 | 8.6 | 11.4 | 24.7 |
| E07 | 9.0 | 4.1 | 14.4 | 15.0 | 7.1 | 21.5 |

**Fig. 4 | Binding characteristics of D11 and E07 mAbs to Gal-3, Gal-1, and Gal-7. a** Binding of D11 and E07 mAbs to human (h) and mouse (m) Gal-3 constructs (FL, full-length; CRD, carbohydrate recognition domain) and their selectivity towards human Gal-1 and Gal-7, measured by ELISA. Each mAb (D11, red; E07, blue) was tested at 20 μg/mL, and the appropriate positive control (commercial antibody against Gal-3, Gal-1, or Gal-7; Ctrl, gray bars) was used at 5 μg/mL. An irrelevant IgG (Irr) was used as a negative control for each assay. **b** Dose–response ELISA of D11

and E07 mAbs binding to hGal-3 (solid lines) and mGal-3 (dotted lines). **c** Dose–response competition between asialofetuin (ASF) and D11 and E07 mAbs to hGal-3. **d** $K_D$ values of D11 and E07 mAbs to recombinant Gal-3 from multiple species (h, human; m, mouse; r, rat; c, cynomolgus; d, dog) by SPR. Data are representative of one experiment performed during the anti-Gal-3 IgG selection flow scheme.

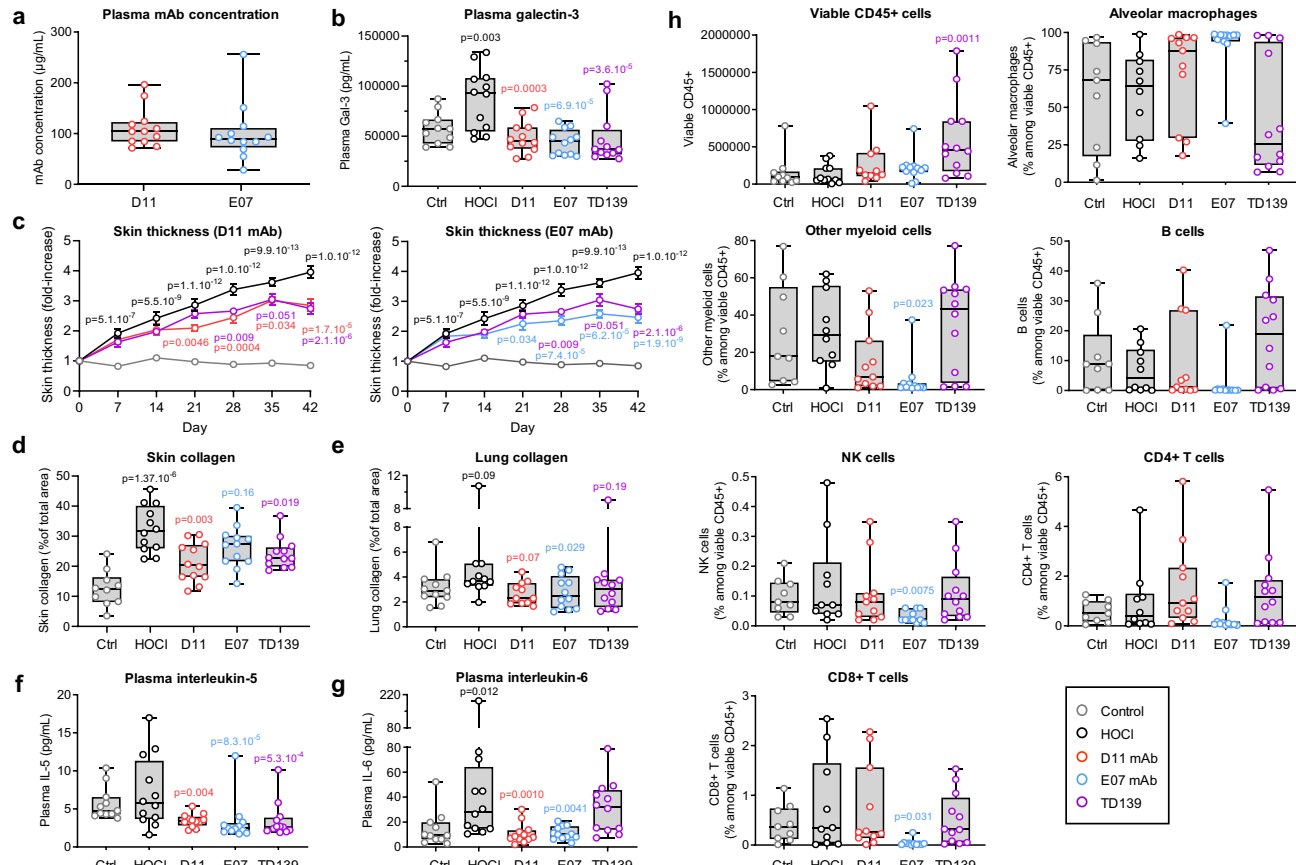

**Fig. 5 | Efficacy of anti-Gal-3 treatments in the mouse model of HOCl-induced SSc.** Control (Ctrl, light gray open circles) and HOCl (dark open circles) groups represent vehicle-receiving and pathology-induced mice, respectively. D11 (red open circles), E07 (blue open circles) and TD139 (purple open circles) groups represent anti-Gal-3 treatments administered in HOCl-induced mice as described in the Methods section. Data are representative of one experiment conducted with 11 biologically independent mice for the control group and $n = 12$ biologically independent mice for all other groups, except in specific cases due to experimental sample loss or unavailability, as detailed in the source data file. **a** Plasma Gal-3 exposure of mAbs at termination day. **b** Plasma Gal-3 levels measured at termination day. **c** Longitudinal measurements of skin thickness by external caliper. **d**, **e** Assessment of collagen deposition by Picrosirius red staining of skin and lung tissue. **f**, **g** Plasma levels of IL-5 and IL-6 measured by multiplex cytokine assay at termination day; **h** FACS immunophenotyping analysis of immune cells in the bronchoalveolar fluid at termination day. All results are depicted as boxes and whiskers plots defined by minimal, maximal, and median values in each dataset, except in **c**, for which means ± sem values are shown. Statistical analyses in **e** and **h** were performed on log-transformed data using a one-way ANOVA for comparison between the HOCl and control group, and with a one-way ANOVA for comparison between all tested items and their HOCl control group, using Dunnett's adjustment. A similar analysis was performed in **d**, without log transformation. Statistical analyses in **b**, **f**, and **g** were performed on log-transformed data using a mixed ANCOVA model for comparison between the HOCl and control group, and using a mixed ANCOVA model for comparison between all tested items and their HOCl control group, using Dunnett's adjustment. Statistical analyses in **c** were performed using a repeated measures ANOVA for comparison between the HOCl and control group, and with a one-way ANOVA for comparison between all tested items and their HOCl control group, using Dunnett's adjustment. The baseline (day minus one) was added as covariable.

increased in HOCl-treated compared with control mice, and D11 and E07 fully reversed this pathological induction. By contrast, TD139 treatment did not affect IL-6 plasma levels. The levels of other plasma cytokines are presented in Supplementary Fig. 12. Significantly, TD139 reduced the plasma levels of IL-1 and TNF-α, E07 reduced the levels of IL-2 and KC/GRO (mouse alias of human CXCL1) and D11 reduced IL-2 levels. Other cytokines were unaffected by any treatment, and IL-12p70 and IL-4 were found below the lower limit of detection in the assay.

Immunophenotyping of cell subsets in the bronchoalveolar lavage fluid was performed at the time of euthanasia (Fig. 5h). The overall amount of total immune cells measured as CD45+ cells remained unchanged between control and HOCl mice and was not altered by Gal-3 blocking mAbs. However, a high increase was observed after TD139 treatment reaching +452% of total immune cells over the HOCl condition. After subset analysis, no difference in any specific immune cell sub-population could be observed between the HOCl and control conditions. However, treatment with E07 mAb increased the ratio of protective alveolar macrophages, while

drastically decreasing the ratio of other proinflammatory myeloid cells, suggesting positive outcomes on the resolution of lung damage and inflammation. In addition, E07 treatment significantly decreased the levels of cytotoxic CD8+ T cells and NK cells compared with the HOCl group. Collectively, E07-mediated protection was associated with a shift in myeloid cells alongside a reduction of lymphoid cells that could contribute to its efficacy.

To gain further insight into the mode of action of anti-Gal-3 treatments at the molecular level, RNAseq of whole blood was performed. In total, 510 genes were differentially expressed between the HOCl and control groups (484 upregulated genes, 26 downregulated genes). These genes are listed in Supplementary Data 3 (columns A–C). A partial least-squares-discriminant analysis (PLS-DA) (Fig. 6a) showed that both groups were well discriminated. When comparing PLS-DA analyses of each anti-Gal-3 treatment and the HOCl group, a higher level of heterogeneity was observed for D11 and TD139, whereas better homogeneity was observed with E07. Remarkably, treatment with E07 resulted in the recovery of a pattern matching that of the control

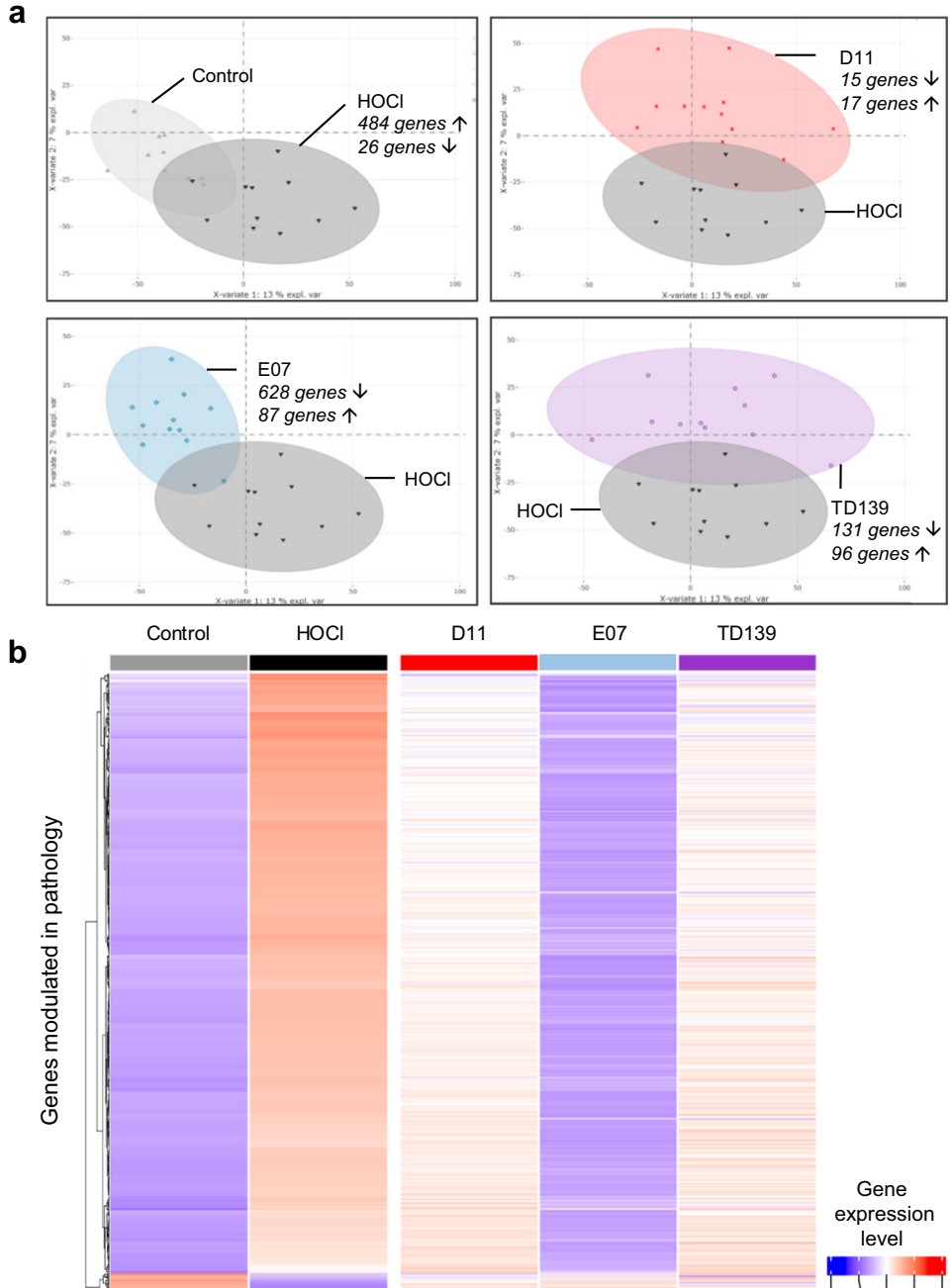

**Fig. 6 | Whole-blood cells RNAseq analysis of anti-Gal-3 treatments in the mouse model of HOCl-induced SSc. a** Partial least-squares-discriminant analysis (PLS-DA) of differential gene expression between HOCl-induced (dark gray) and untreated (control, light gray) group, or HOCl-induced groups treated with D11 (red), E07 (blue) or TD139 (purple), and HOCl untreated group. The number of upregulated (up arrow) or downregulated (down arrow) genes with false discovery rate (FDR) below 0.05 and absolute fold-change (FC) above 1.3 refers to each relevant comparative analysis. **b** Gene expression heatmap showing the 510 genes differentially expressed between the HOCl-induced and control groups, and their modulation by each anti-Gal-3 treatment. Each column represents the mean relative gene expression levels of *n* = 11 (control) or *n* = 12 (HOCl, D11, E07, TD139) mice per group. Each gene is represented by one horizontal line. Gene expression intensities are displayed as colors ranging from blue for the lowest expression level to red for the highest expression level as shown in the key. The relative FC and FDR values are reported in Supplementary Data 3.

healthy group (Fig. 6a, lower left panel). To a lesser extent, due to a higher level of heterogeneity, a similar shift and discrimination from the HOCl group was observed for D11 and TD139 treatments. The list of genes differentially expressed between each treated condition and the HOCl group is reported in Supplementary Data 3 (columns D-L). A global heatmap was generated to visualize the impact of treatment on the pathological genes differentially expressed between HOCl and

control groups (Fig. 6b). The most striking observation was that the expression patterns of 445 of these genes (representing 87.2% of the pathology-modulated genes) were fully reversed after E07 mAb treatment, as evidenced by the similar global pattern of the control and E07 groups. To a lesser extent, D11 mAb and TD139 also alleviated pathological gene expression. The list of pathological genes whose expression was reversed by E07 is presented in Supplementary Data 4.

### Translational relevance of the preclinical efficacy of E07 mAb treatment in a population of patients with SSc

The translational relevance of the preclinical effects of E07 mAb treatment at the gene expression level was investigated by comparing IPA profiles in the HOCl model and in patients from the SSc PRE-CISESADS cohort. The meaningful IPA approach allows to prediction of molecules and pathways associated with complex gene expression changes. As anticipated, E07 mAb treatment was predicted to reverse virtually all major biological entities modulated by HOCl (Fig. 7a, b), among which the well-known actors or downstream effectors of the IFN pathway IFN-λ1, IFN-α2, MAVS, DDX58 (more commonly known as retinoic acid-inducible gene I, RIG-I), IRF-3, -5, and -7 were predicted as inhibited. Other predicted E07-inhibited entities included INSR, EIF2AK2, HSPA5, EIF2AK4, RAF1, FOXO1, FOXO3, FOXM1, STAT4, MYBL2, and NFE2L2 (all of which were upregulated under HOCl conditions), and predicted E07-activated entities included ACKR2, PNPT1, IRGM1 and BACH1 (all of which were downregulated under HOCl conditions). Overall, the biological pathways predicted to be modulated by E07 drew an inverse picture to that found in pathological conditions.

The next step was to identify by unbiased analysis the predicted pathological pathways representative of each cluster of the SSc PRE-CISESADS cohort, based on whole-blood cells RNAseq data (Fig. 7c–e). Remarkably, many of the key entities identified as pathological drivers in the mouse HOCl model were represented in one or more clusters of SSc patients, highlighting the high relevance of this preclinical SSc model to reflect human pathology. These key entities included IFN-λ1, IFN-α2, IFN-γ, IRF-1, -3, -5 and -7, DDX58, MAVS, EIF2AK2, ACKR2, PNPT1, STAT1 and FOXO3. Consistent with a more 'healthy-like' clinical picture, patients from C2 were represented by a smaller network, where E07 treatment was nevertheless predicted to be able to impact major IFN pathways as well as ACKR2 (Fig. 7e). Again, consistent with an increasing level of complexity and disease severity in C1 and C3, E07 mAb demonstrated remarkable predictive efficacy in reverting disease pathways, with 11 out of 27 entities targeted in C1 (Fig. 7d) and 11 out of 23 entities targeted in C3 (Fig. 7c). Furthermore, it should be noted that all seven biological entities predicted as modulated by E07 mAb in C2 were common to C1 and C3. Similarly, IFN-λ1, IFN-α2, IRF-7, EIF2AK2, DDX58 and PNPT1 were common to C1 and C3.

In addition, we investigated how the top 25 pathological pathways in SSc patients from C3 were modulated in the mouse HOCl model, first in HOCl-induced versus control mice, then in HOCl-induced mice treated with E07 (Fig. 8). Except for the EIF2 signaling pathway, all other pathways followed a strikingly similar pattern between SSc patients and HOCl-induced mice, and E07 consistently counteracted the pathological picture. Collectively, these data converge to demonstrate the relevance of the mouse HOCl model to mimic human SSc and of a Gal-3-neutralizing strategy as a treatment for SSc patients.

## Discussion

Gal-3 was initially discovered more than 30 years ago as Mac-2, a protein expressed by macrophages under inflammation stimuli[32], then described as an important regulatory factor acting at various stages along the continuum from acute inflammation to chronic inflammation and tissue fibrogenesis[5–7,30] and involved in a wide range of diseases[33]. As a result of its pleiotropic role in multiple pathophysiological conditions, Gal-3 is sometimes described as a diagnostic or prognostic biomarker, or as a direct player involved in the cause of disease[3,34,35]. This question has remained unsolved due to the lack of Gal-3 inhibitors with adequate potency, selectivity, and bioavailability that may allow to demonstrate proper efficacy in clinical studies.

Until now, Gal-3-targeting approaches were built mainly on the high-resolution crystal structure of the CRD of Gal-3 in complex with lactose[36], knowing that this protein-ligand interface is the main component required to drive the biological functions of Gal-3.

Carbohydrate-protein interactions are relatively weak, as a result of the lack of charges and the hydrophobic surfaces of saccharides, thus reducing the likelihood of forming strong interactions with proteins. Additionally, all 15 galectin members share primary structural homology within their CRD all of them being able to bind specifically to β-galactoside sugars[30]. Nevertheless, more in-depth analyses based on nuclear magnetic resonance or X-ray crystallography methods showed that despite this high degree of conservation between CRDs of the different galectins, subtle differences in the interaction with specific ligands were involved, suggesting the possibility of finding selective Gal-3 inhibitors[37,38].

Reflecting the progress made, a thiodigalactoside compound named TD139 or GB0139, showed nanomolar affinity to Gal-3 with clear selectivity over Gal-7 but was not selective towards Gal-1[20,39]. TD139 showed positive anti-fibrotic and anti-inflammatory benefits after intratracheal administration in a mouse model of IPF[19,20], and reduced disease-associated plasma biomarkers after inhaled formulation in a phase 1/2a clinical study in IPF patients[21]. Belapectin, a galactoarabino-rhamnogalacturonan polysaccharide polymer with low micromolar binding affinity to both Gal-3 and Gal-1[40] did not meet its primary and secondary endpoints in a phase 2b study in patients with NASH, cirrhosis, and portal hypertension, and was repurposed in a subgroup of patients for the prevention of esophageal varices in NASH cirrhosis[41] (ClinicalTrials.gov NCT04365868). A large number of other approaches have focused on targeting Gal-3 with carbohydrate or noncarbohydrate-based compounds, most having been evaluated in preclinical studies and just a few of them reaching clinical stages in various indications[42–45].

To circumvent these challenges of potency, selectivity, and bioavailability, we chose to develop Gal-3 mAb able to interact with the CRD of Gal-3. Our efforts allowed the successful identification of 2 mAbs with nanomolar binding affinities to Gal-3, without any binding detected to other tested galectins, except for Gal-14 for which a moderate binding could be observed. As this protein is located intracellularly and its expression is restricted to placenta[46,47], this binding is unlikely to occur in biological systems or in a clinical setting, and it does not represent an interpretation bias in the present study. Both mAbs also showed a good PK profile after subcutaneous administration in mice. In light of these favorable characteristics, we addressed the therapeutic potential of these mAbs in SSc, by using a translational approach reconciliating data gathered in a relevant murine model of the disease and projection to patients of the SSc PRECISESADS cohort. As a lectin, Gal-3 exerts its role in biological systems through its interaction with multiple ligands. In our integrative approach, the preliminary step was thus to consider the potential modulation of these Gal-3 ligands at a molecular level, rather than Gal-3 alone. Surprisingly, such an approach has never been investigated before. Along this line, we first demonstrate that the Gal-3 network is dysregulated in SSc with 41.5% of interactants found differentially expressed between patients and HV, highlighting a close association with the disease. Further strengthening this assumption, we have identified a strong transcriptomic Gal-3 fingerprint of 69 Gal-3 interactants correlated with the gradient of severity of the disease reflected by altered lung and cardiac functions and immune and inflammatory status. Notably, the neutrophil-to-lymphocyte ratio (NLR), was strongly correlated with the Gal-3$^{up}$ score and inversely correlated with the Gal-3$^{down}$ score. Recent studies have shown that higher neutrophil counts are associated with the severity of skin and lung involvement in SSc. A negative correlation was observed between neutrophil counts and the diffusing capacity of the lungs for carbon monoxide (DLCO)[48], a marker of the severity of lung disease. Higher blood neutrophil counts and NLR were also predictive of more severe disease and increased mortality in SSc. This predictive significance of NLR for SSc severity suggested that the enrichment of neutrophils and underrepresentation of lymphocytic cells reflected the pathologic immune dysregulations

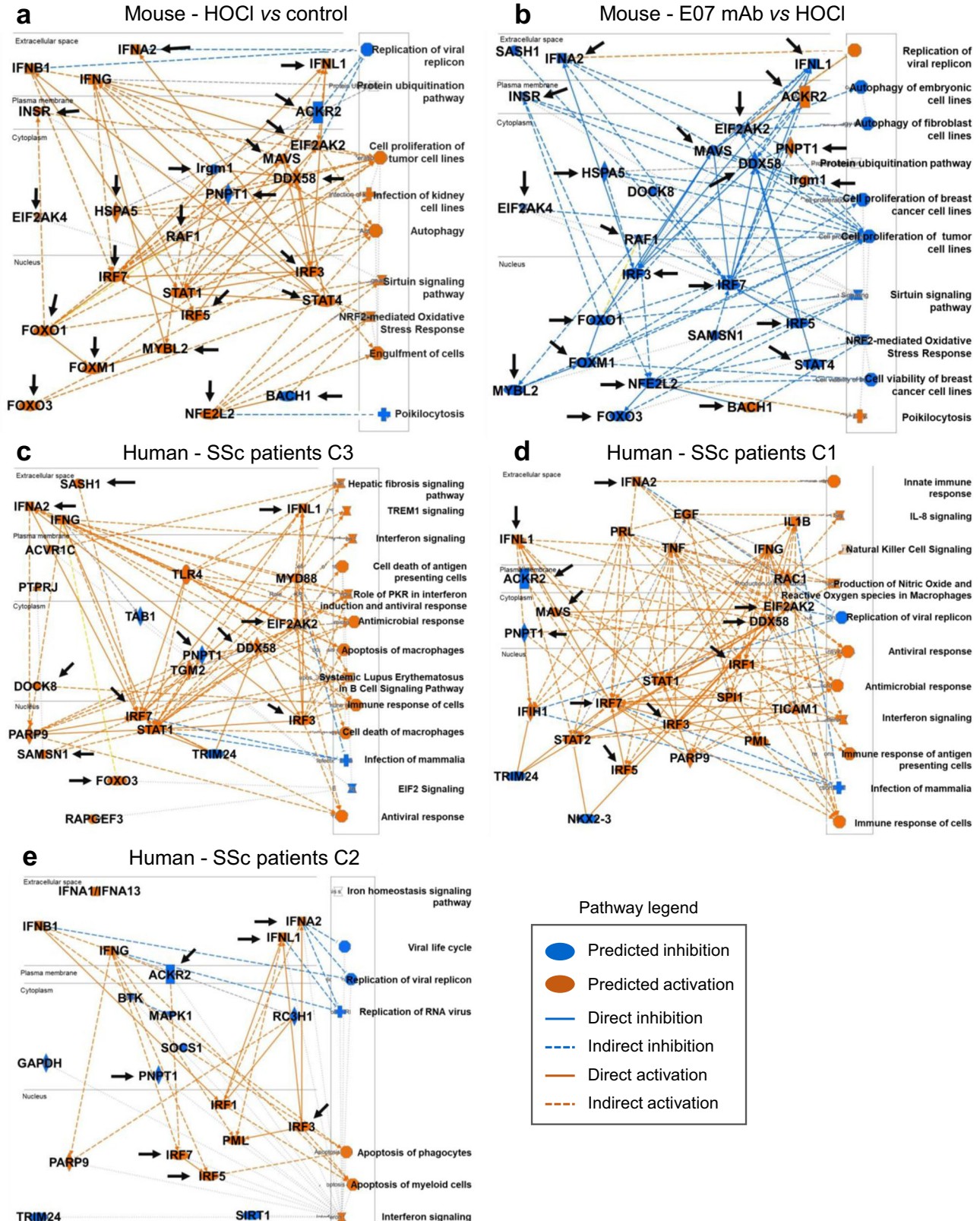

**Fig. 7 | Ingenuity pathway prediction analysis of Gal-3 entities modulated in the mouse model of HOCl-induced SSc and in patients with SSc. a** Prediction of entities modulated in HOCl-treated *versus* untreated control mice. **b** Prediction of entities modulated by E07 mAb treatment *versus* control HOCl-induced mice. Orange and blue colors indicate predicted activation or inhibition, respectively, as indicated in the key legend code. Black arrows in **a** and **b** indicate entities found in common and inversely modulated in both conditions. **c–e** Prediction of entities modulated in SSc patients of the PRECISESADS cohort. The three graphical summaries depict entities modulated in cluster 3 (**c**), 1 (**d**), and 2 (**e**), respectively. Black arrows in **c–e** indicate entities modulated in these patients and found to be inversely modulated by E07 mAb in the mouse model of HOCl-induced SSc (**b**).

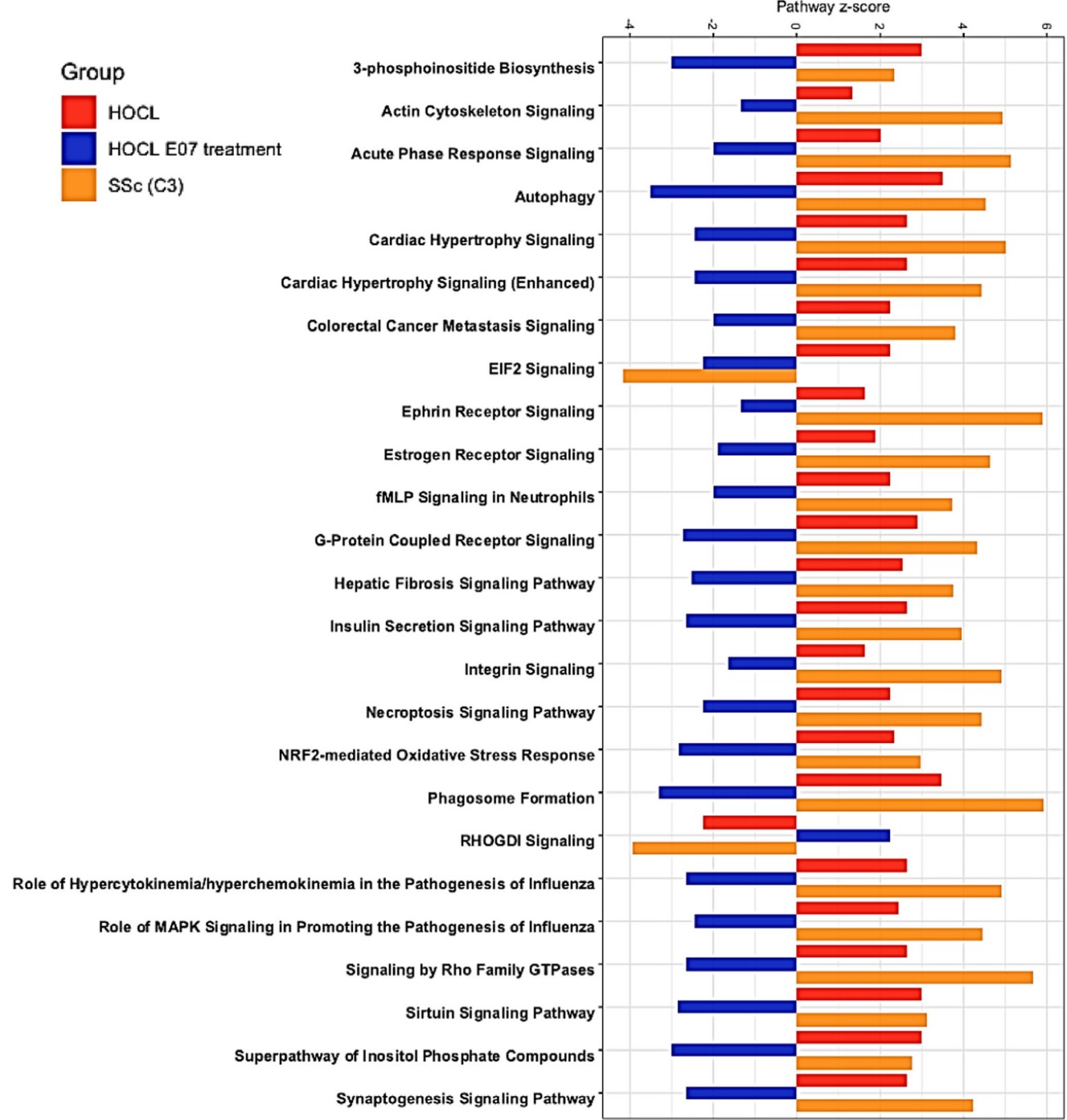

**Fig. 8 | Top 25 most representative canonical pathways modulated by E07 mAb in the mouse model of HOCl-induced SSc and in C3 SSc patients.** Horizontal bars represent *z*-score values for each canonical pathway (below 0, inhibited, above 0, activated) for each condition. Red, HOCl-induced mice versus untreated control mice; Blue, HOCl-induced mice *versus* HOCl-induced mice treated with E07 mAb; Orange, systemic sclerosis patients, cluster 3 versus healthy volunteers of the PRECISESADS cohort.

observed in SSc peripheral blood cells[49]. Taken together, these findings allow us to define the Gal-3 network at least as a relevant SSc severity biomarker.

To address the latter question, we used the mouse model of HOCl-induced systemic sclerosis, depicted to recapitulate the inflammatory and fibrotic features of the human disease much better than vasculopathy events[50–52], and most suited to reflect the pathogenic role of Gal-3. This could be verified in our hands, as all major pathological pathways altered in SSc patients were found similarly altered in HOCl-induced mice, based on RNAseq data of whole-blood cells. As a

proof-of-concept study, our two anti-Gal-3 mouse cross-reactive mAbs were evaluated in this model in comparison with TD139. Due to its short half-life and low bioavailability, TD139 was administered intratracheally, as previously described[19,20], thus limiting the scheme of administration to follow ethical guidelines and to preserve animal welfare. It was however possible to push up to eight instead of four administrations starting from the mid-point of the study, and mAbs were administered every 5 days from the beginning of the study, implying that both treatment conditions cannot be strictly compared. D11 and E07 mAbs and TD139 demonstrated efficacy at different levels,

by reducing the pathological induction of skin thickness, the skin and lung collagen content, the relative frequencies of pulmonary macrophages, and the plasma Gal-3 levels in HOCl-induced mice. Among a panel of 10 cytokines tested, the proinflammatory cytokines IL-5 and IL-6 were reduced by most treatments. Interestingly, IL-6 is known to play a major role in the pathophysiology of SSc. Tocilizumab, a mAb directed against the IL-6 receptor was approved by the FDA for the treatment of lung fibrosis in SSc[53]. Although IL-5 expression levels have been reported to be increased in BAL cells from SSc patients in association with a significant decline in forced vital capacity, the importance of this cytokine in SSc remains hypothetical[54]. Only E07 mAb showed positive outcomes on immune cell populations, revealing a capacity to reduce the bronchoalveolar fluid content of NK and CD8+ T cells and to increase the ratio of alveolar macrophages over other myeloid cells, indicative of protective function in damaged lungs[55].

Remarkably, by advancing our understanding of the actions of E07 mAb at the gene expression level, we found that most genes modulated by HOCl in whole-blood cells were reversed by E07 to the level of control untreated mice, which was also observed to a lesser extent with D11 mAb and TD139. It should be noted that the exact mode of action of anti-Gal-3 mAbs is not fully elucidated. In this respect, the relative contributions of Gal-3 interactants in the observed changes remain to be determined, which undoubtedly represents a challenging task, not only because of their high number but also due to their likely complex networking in biological systems. To circumvent this difficulty, we investigated whether the mode of action of E07 mAb could be deciphered more at the biological pathway level. The emergence of E07 as a lead pathology-attenuating agent was confirmed by IPA. The predicted inhibited pathological pathways included several arms of the IFN pathway among which type -I, and -III IFN subgroups and their downstream effectors RIG-I, MAVS, and IFN regulatory factors -3, -5, and -7. These findings looked particularly relevant from a translational perspective, considering the recent acknowledgement of IFN as a key player in SSc[56,57]. This was corroborated by pathways analyses in all SSc patient clusters of the present study, for which all of the above IFN-related effectors were identified as important molecules associated with the disease. Furthermore, the present study also allowed the identification of several key molecules involved in the pathological picture of SSc patients and predicted to be modulated inversely by E07 mAb. They were represented by IFN-λ1, IFN-α2, IRF-3, -5 and -7, MAVS, DDX58, SAMSN1, SASH1, PNPT1, ACKR2, FOXO3, and EIF2AK2. Overall, our findings strongly suggest that Gal-3 is an upstream effector of several pathological SSc players.

It should be recalled that despite recent progress with the marketing approvals of the anti-fibrotic agent nintedanib and the IL-6 receptor antibody tocilizumab for patients with SSc-ILD[53], there is still no cure for SSc. To date, several treatments are offered to patients on an individual basis depending on their various symptoms[9,10]. The results of the present study suggest that adequate anti-Gal-3 drug therapy may benefit this patient population for whom a critical medical need persists.

## Methods
### Patient population
Patients with systemic sclerosis (SSc) and healthy volunteers (HV) were obtained from the European multi-center cross-sectional study of the PRECISESADS IMI consortium which involved patients from seven systemic autoimmune diseases. The PRECISESADS cohort participants did not receive any compensation. SSc patients were aged between 50 and 68 years (median 60 years) and comprised 212 women and 37 men representing 85% and 15% of patients, respectively. Healthy volunteers (HV) were aged between 47 and 59 years (median 53 years) and comprised 307 women and 58 men, representing 84% and 16% of HV, respectively. Sex rather than gender was considered for women and

men, based on self-reporting. All patients and HV gave written informed consent for the study that was approved by local ethical committees of the 19 participating institutions: Fondazione IRCCS Ca' Granda Ospedale Maggiore, Policlinico di Milano, Comitato Etico Italy; Università degli studi di Milano, Policlinico di Milano, Comitato Etico Italy; Centre Hospitalier Universitaire de Brest, Comité de Protection des Personnes Ouest VI Brest, France; Université catholique de Louvain, Comité d'Ethique Hospitalo-Facultaire, Brussels, Belgium; Centro Hospitalar do Porto, Comissao de ética para a Saude, CES do CHP Porto, Portugal; Servicio Cantabro de Salud, Hospital Universitario Marqués de Valdecilla, Comite ético de investigacion clinical de Cantabria, Santander, Spain; Hospital Clinic I Provicia, Institut d'Investigacions Biomèdiques August Pi i Sunyer, Comité Ética de Investigación Clínica del Hospital Clínic de Barcelona, Spain; Katholieke Universiteit Leuven, Commissie Medische Ethiek UZ KU Leuven/Onderzoek, Belgium; Klinikum der Universitaet zu Koeln, Geschaftsstelle Ethikkommission, Cologne, Germany; Medizinische Hochschule Hannover, Ethikkommission, Hannover, Germany; Medical University Vienna, Ethik Kommission, Borschkegasse, Vienna, Austria; Servicio Andaluz de Salud, Hospital Universitario Reina Sofía Córdoba, Comité de Ética e la Investigación de Centro de Granada (CEI−Granada), Spain; Andalusian Public Health System Biobank, Granada, Spain; Servicio Andaluz de Salud, Complejo hospitalario Universitario de Granada (Hospital Universitario San Cecilio), Comité de Ética e la Investigación de Centro de Granada (CEI−Granada), Spain; Servicio Andaluz de Salud, Complejo hospitalario Universitario de Granada (Hospital Virgen de las Nieves), Comité de Ética e la Investigación de Centro de Granada (CEI−Granada), Spain; Servicio Andaluz de Salud, Hospital Regional Universitario de Málaga, Comité de Ética e la Investigación de Centro de Granada (CEI−Granada), Spain; Hospitaux Universitaires de Genève,. DEAS, Commission Cantonale d'éthique de la recherche Hopitaux universitaires de Genève, Switzerland; Csongrad Megyei Kormanyhivatal, University of Szeged, Hungary; Charite, Ethikkommission, Berlin, Germany.

The cross-sectional study (registered as NCT02890121 in ClinicalTrials.gov) adhered to the standards set by the International Conference on Harmonization and Good Clinical Practice (ICH-GCP) and to the ethical principles that have their origin in the Declaration of Helsinki (2013). The protection of the confidentiality of records that could identify the included subjects is ensured by the EU Directive 2001/20/EC and the applicable national and international requirements relating to data protection in each participating country. For each individual (SSc patients and HV), blood samples as well as biological and clinical information were collected, including demographic data, symptoms, comorbidities, current medications, and biological information. For each patient, disease duration and organ involvement were reported. More technical details about the sample and data collection (including inclusion/exclusion criteria and immunophenotyping) have been published previously[58].

Flow cytometry analyses were performed previously in eleven centers in the context of the PRECISESADS study, requiring multicenter harmonization of flow cytometers and procedures to integrate data. The protocol has been described previously[28,58–60], and a gating strategy is shown in Supplementary Fig. 13. After exclusion of debris, dead cells, and doublets, neutrophils (CD15hiCD16hi) were gated from CD15+ polymorphonuclear leukocytes. Lymphocytes were identified as B cells (CD19+CD3−) and T cells (CD19−CD3+). Human plasma Gal-3 was quantified using the Simple Plex assay SPCKB-PS-000490 from Biotechne (Minneapolis, USA). After quality control on transcriptomic RNAseq data[28], verification of the American College of Rheumatology (ACR)/European League Against Rheumatism (EULAR) classification criteria[61] and match of HV and patients based on age and gender, the final studied cohort after clustering comprised 249 SSc patients and 365 HV. Their characteristics are reported in Table 1.

## Molecular subgroup discovery and identification of a Gal-3 fingerprint associated with SSc clusters

Patient subgroup discovery was based on the pre-processed RNAseq data and clustering methodologies previously used for primary Sjögren syndrome patients from the PRECISESADS cohort[28]. Except when indicated, data analyses were carried out using either an assortment of R system software (http://www.R-project.org, V4.0.1) packages including those of Bioconductor v3.17, or original R code. R packages are indicated when appropriate. In order to analyze RNAseq data, first, bcl2fastq2 Conversion Software v2.20 was used to demultiplex sequencing data and convert BCL files. Quality control was obtained with FastQC tools v0.11.18 and adapters were removed with Cutadapt v1.18. Transcriptome alignment was done with STAR v2.5.2b on GEN-CODE v19 annotation (hg19) and read counts were obtained with RSEM v1.2.31. For normalizations and batch correction, read counts were normalized by the variance stabilizing transformation vst function from the DESeq2 v1.30.0R package. To reduce the effect of the RIN, a correction was applied using the ComBat function from the sva v3.38.0R package, after categorization of RIN values into 7 classes: [6.5–7.0], [7.0–7.5], [7.5–8.0], [8.0–8.5], [8.5–9.0], [9.0–9.5], [9.5–10].

Our objective was to obtain a robust classification scheme ensuring the identification of highly homogeneous SSc subgroups. The strategy used iterates unsupervised and supervised steps and is therefore designated as a "semi-supervised" approach. The training set comprised 263 SSc samples and the test set comprised 65 samples. To determine the number of clusters of patients (unsupervised step), a consensus clustering between three methods was performed: (i) Agglomerative Hierarchical Clustering (hclust function from stats v4.0.2R package) with Pearson correlation as a similarity measure and Ward's linkage method, (ii) k-means clustering (k-means function from stats R package) with four groups and (iii) Gaussian mixture clustering (mclust function from mclust v5.4.6R package). The number of clusters was determined as the best consensus between three unsupervised clusterings (https://github.com/psBiostat/GAL3_PAPER.git)[62]. From 328 patients, 249 (219 in the training and 30 in the discovery test) were considered as consensus in the clustering (Supplementary Fig. 2). The top discriminating genes were defined with randomForest function from randomForest v4.6-14R package. Heatmaps were obtained with the ComplexHeatmap v2.6.2R package. Enrichment analysis was performed using transcriptional module repertoire and generated fingerprint representations with BloodGen3Module v0.99.36.

A non-redundant list of Gal-3 interacting proteins was compiled by bioinformatics by querying the GPS-prot database[63], allowing the referencing of 210 Gal-3 interactants, and the Ingenuity Pathway Analysis database[64], allowing the referencing of 276 Gal-3 interactants. Merging the 2 databases finally led to the identification of 307 proteins, listed in Supplementary Data 1 (column A), among which 59 were found absent in datasets of the PRECISESADS SSc cohort (Supplementary Data 1, column B). Among the 248 remaining genes, we searched for those that were significantly associated with PRECISESADS SSc clusters by applying the randomForest method (randomForest function from randomForest R package[65]). To obtain a robust selection, we generated 200 bootstrap replicates. Only genes frequently chosen by randomForest over bootstrap samples were selected, which improved the stability of the results. The frequency threshold was specified to be conservative (100%) so as not to exclude any cluster-associated gene. The most robust Gal-3 fingerprint associated with SSc patient clusters was defined by 69 Gal-3 interactants, 48 of which were upregulated and 21 downregulated as compared to C3. 'Gal-3$^{up}$' and 'Gal-3$^{down}$' scores were calculated as follows:

$$\text{Score Gal-3}^{up} = \text{Median}_{j=1,\ldots,G^{up}} \left( Y_{ij} - \text{median}_{k=1,\ldots,N_{ctrl}} \left( \text{Ctrl}_{kj} \right) \right)$$

$$(1)$$

$$\text{Score Gal-3}^{down} = \text{Median}_{j=1,\ldots,G^{down}} \left( Y_{ij} - \text{median}_{k=1,\ldots,N_{ctrl}} \left( \text{Ctrl}_{kj} \right) \right)$$

$$(2)$$

Where:

$Y_{ij}$ represents the $j^{th}$ expression gene of $i^{th}$ patient,

$\text{Ctrl}_{kj}$ represents the $j^{th}$ expression gene of the $k^{th}$ HV,

$G^{up} = 48$ and $G^{down} = 21$ represent the total number of the upregulated and downregulated genes, respectively,

$N_{Ctrl}$ represents the total number of HV.

For the global differential gene expression analysis between HV and each patient subgroup, we applied a linear model (lmFit function from limma v3.46.0R package) on vst transformation gene expression dataset. The resulting p-values were adjusted for multiple hypothesis testing and filtered to retain differentially expressed genes with false discovery rate (FDR) adjusted p-values ≤ 0.05 and an absolute fold-change (|FC|) ≥ 1.3. Ingenuity Pathway Analysis (IPA) was applied to determine the most significantly dysregulated canonical pathways with Benjamini-Hochberg FDR adjusted p-values ≤ 0.05 and |FC| ≥ 1.3.

## Criteria for selection and characterization of candidate therapeutic monoclonal antibodies against Gal-3

The criteria for generating Gal-3 blocking mAb were defined as follows: ability to interact with the carbohydrate recognition domain (CRD) of Gal-3, and full-length (FL) Gal-3; ability to bind primarily to human Gal-3 (hGal-3) and also to mouse Gal-3 (mGal-3) for in vivo testing; selectivity versus close galectin family members represented by Gal-1 and Gal-7; ability to displace natural Gal-3 ligand from the CRD.

## ScFv selection, production, and binding characteristics

The naive scFv library (HuscI™, Mabqi proprietary library) was used for phage display panning. HuscI™ is a synthetic library based on a single cytosoluble hyper-stable human framework scFv termed 13R4[66–68] with side chain diversity incorporated at positions that contribute most to the antigen binding energy and least to intra-scFv contacts[69]. For the selection of scFv, recombinant His-tagged FL mGal-3, and hGal-3 were produced by Novalix (Illkirch, France) and used sequentially as baits to select human and mouse cross-reactive Gal-3 binders through successive rounds of enrichment by phage display with a human-mouse-human panning strategy. Recombinant proteins were immobilized through their His-tag on NiSO$_4$/Tris-NTA-Biotin-coated microplates following the method described previously[70]. The naive HuscI™ phage library was depleted on wells coated with streptavidin and NiSO$_4$/Tris-NTA-biotin before panning. Following the last round of panning, bacteria colonies containing the selected scFvs were picked, and grown, and the scFvs were expressed as monoclonal fragments in deep-well plates. Finally, cultures were centrifuged and polymyxin B sulfate was used for periplasmic extraction of scFv from pellets.

## ScFv binding and selectivity ELISA on human and mouse Gal-3, hGal-1, and hGal-7

Recombinant proteins hGal-3-FL-His Nter, hGal-3-CRD-His Nter, and mGal-3-FL-His Nter (Novalix, Illkirch, France) were immobilized on Nunc maxisorp plates via their His tag as described above. Recombinant proteins hGal-1 and hGal-7 (Novalix) were immobilized passively on Nunc maxisorp plates at 3 μg/mL in PBS, 100 μL/well, overnight at 4 °C, then blocked with 4% BSA. Plates were washed three times with PBS containing 0.1% Tween 20 (PBST), and periplasmic extracts were added at a 1:2 dilution in blocking buffer (PBS containing 4% of skimmed milk powder) for 1 h at room temperature. For detection of scFv via their myc-tag, a secondary anti-myc-HRP antibody (Santa Cruz sc-40) was added in blocking buffer at a 1:2000 dilution for 1 h at room temperature. Commercial antibodies directed against hGal-3 (Abcam 53082), hGal-1 (Abcam 25138), and hGal-7 (Abcam 108623) were used at 5 μg/mL as positive controls and detected with an anti-rabbit HRP

secondary antibody (Cell Signaling 7074S) diluted 1:2000. Plates were washed 3 times using PBST. TMB chromogenic substrate for HRP detection was added to the wells and the reaction was stopped with $H_2SO_4$ 1 M before optical density measurements.

## ScFv reformatting to full-length IgG

Anti-Gal-3 scFvs were reformatted to full-length antibodies (IgG1 subclass) and further engineered with the double LALA mutation in the Fc region known to strongly reduce immune effector functions both in human and mouse[71,72], and produced in CHO-K1 cells at Evitria (Zürich, Switzerland). Briefly, CHO-K1 cells were grown in eviGrow medium then transfected with eviFect and kept in eviMake2 production medium. Before protein A affinity purification using MabSelect™ SuRe™ (Cytiva), transfection supernatants were harvested by centrifugation followed by sterile filtration (0.2 μm). A dialysis was performed to formulate the IgG into DPBS leading to high-quality mAbs with endotoxin levels <1 EU/mg. For quality control, antibodies were analyzed by SDS-PAGE, size exclusion chromatography-multi angle light scattering (SEC-MALS), and mass spectrometry (MS). SDS-PAGE was performed using pre-cast NuPAGE 4–12% in MOPS buffer (ThermoFisher Scientific) at 150 V. Three micrograms of proteins were mixed with loading buffer with or without reducing agent and separated using standard conditions. The gels were then stained with Instant Blue (Expedeon, Cambridge, UK) according to the manufacturer's recommendations. Homogeneity was determined by SEC-MALS using an ACQUITY UPLC Protein BEH SEC column, 200 Å (Waters, Saint-Quentin-en-Yvelines, France) at a flow rate of 0.4 mL/min in ammonium acetate, pH 5.4 running buffer. After centrifugation at 13,000 x $g$ for 20 min, 5 μL of each purified mAb (25 μg) was injected into the column. Protein elution was monitored at 280 nm using a diode array detector (Agilent) and homogeneity was measured using a Heleos 8 + MALS detector (Wyatt, Toulouse, France) and Astra software v3.1.4 (Wyatt). For mass measurements, mAb was enzymatically deglycosylated with PNGase F (New England Biolabs, Evry, France) for 1 h at 37 °C and 1 μg of mAb was injected on a BioResolve RP 450 Å, 2.1*150 mm, 2.7 μm column (Waters) at 0.1 mL/min using a 20 to 90% acetonitrile gradient in 0.1% formic acid. Intact experimental masses were determined using a Xevo G2-XS Q-ToF (Waters) mass spectrometer coupled to a H-Class Bio UPLC system (Waters) followed by MaxEnt deconvolution. The thermal stability of mAbs was studied in comparison with a reference IgG1 approved in the EU, namely trastuzumab (EU/1/00/145/001). mAb samples stored at 4 °C or 40 °C for 2 weeks were analyzed by SEC using a BEH SEC column, 200 Å (Waters), with a of 25 mM ammonium acetate, pH 5.4 running buffer, and UV absorbance was recorded at 280 nm. Thermal stability was also assessed using nano differential scanning fluorimetry (nanoDSF) with mAb at 2 mg/mL in 50 mM histidine, 0.2 M glycine buffer, pH = 5.5. NanoDSF was averaged from duplicates using a Prometheus instrument (NanoTemper Technologies, München, Germany) from 20 to 95 °C with a 1 °C/min temperature increase and readouts at both 350 and 330 nm. Nucleotide sequences of D11 and E07 anti-Gal-3 mAbs have been filed in the patent application EP22305372 in March 2022.

## IgG ELISA on Gal-3 and selectivity toward other galectin members

Single-dose IgG ELISA on hGal-3-FL, hGal-3-CRD, mGal-3-FL, hGal-1, and hGal-7 were performed using the procedure described for scFv with IgG at 20 μg/mL. Selectivities towards human Gal-2 (R&Dsystems 9874-GA), -4 (R&Dsystems 1227-GA), -8 (R&Dsystems 1305-GA), -10 (abcam ab107951) and -14 (Novus Biologicals NBP1-50082) were determined following a similar procedure using their respective commercial antibodies as positive controls. Gal-9 was excluded from the analysis due to low-quality control. For dose–response binding ELISA on hGal-3-FL and mGal-3-FL, IgGs were tested at concentrations ranging from 0.0009 to 500 nM. A secondary anti-hFab-HRP antibody (Sigma,

A0293) was used for IgG detection using a classical TMB chromogenic procedure.

## Asialofetuin dose–response inhibition of IgG1 mAbs on human Gal-3-CRD by ELISA

Recombinant hGal-3-CRD was coated as described above at 1 μg/mL and plates were blocked with 200 μL of TBS-BSA 4 % for 2 h, then washed three times with PBST, and IgGs were added to the wells in blocking buffer at concentrations ranging from 0.0006 to 780 nM for 1 h at room temperature. Plates were washed three times using PBST and asialofetuin (Sigma A1908) was added to the wells at 8 μg/mL in PBS containing BSA 4%, for 30 min at room temperature. The binding of asialofetuin to immobilized Gal-3 was detected using an anti-asialofetuin antibody (Abcam, ab35184) labeled using an HRP conjugation kit (Abcam ab102890) diluted 1:1500.

## Binding kinetics between anti-Gal-3 mAbs and multiple Gal-3 species by surface plasmon resonance (SPR)

The interaction of anti-Gal-3 mAbs with recombinant Gal-3 from multiple species (Novalix) was monitored by SPR detection using a Biacore T-200 instrument (Biacore AB, Uppsala, Sweden). Binding studies were performed in freshly prepared, filtered, and degassed running buffer containing 10 mM HEPES, 150 mM NaCl, and 0.05% P20 (HBS-P, Cytiva Life Sciences) at 25 °C. Before the experiment, the CM5 sensor surface immobilized with anti-human Fc mAb was equilibrated with HBS-P by priming the instrument at least three times. The CM5 sensor surface was immobilized with an anti-human Fc mAb using a standard protocol. The entire sensor surface was then activated by injecting a 1:1 (v/v) mixture of 400 mM 1-ethyl-3-(3-dimethylaminopropyl)carbodiimide hydrochloride (EDC) and 100 mM NHS at a flow rate of 10 μL/min for 7 min. Monoclonal anti-human Fc antibody (Millipore AP113), 25 μg/mL in 10 mM sodium acetate pH 5, was flowed for 6 min at 10 μL/min to reach 10000 resonance units (RU) on both flow cells followed by the injection of 1 M ethanolamine, pH 8.5 for 7 min at a flow rate of 10 μL/min. HBS-P was used as the running buffer during the entire immobilization procedure. Antibody samples at 5 μg/mL were injected for 30 s at 10 μL/min. In general, 400 to 680 resonance units of antibodies were captured on the chip. During the single-cycle kinetics (SCK) analysis assays, a range from 0.5 to 50 nM of human, mouse, rat, dog, or monkey Gal-3 were injected over the mAb capture surfaces for 210 s followed by a dissociation phase of 600 s at 30 μL/min, in duplicate conditions. At the end of each cycle, capture surfaces were regenerated by 2 injections of 10 mM glycine, pH 1.5, for 30 s at 10 μL/min, followed by an extra wash with 10 mM glycine, pH 1.5. Fresh mAb was captured at the beginning of each cycle. Each analyte injection cycle was preceded by one buffer injection cycle. The double reference subtracted SCK data was fitted with a 1:1 Langmuir using the SCK binding model with the Biacore T-200 evaluation software (version 3.2). Each interaction was investigated in duplicate (using two independent analyte dilution series).

## Treatments and sampling in the mouse model of HOCl-induced SSc

Sixty 6-week-old *Balb/c AnNRj* mice, initially selected by Mac Dowell starting from a stock of inbred Albino mice (https://janvier-labs.com/en/fiche_produit/balb-cjrj_mouse/) purchased from Janvier(Le Genest-St-Isle, France) were included in the study. Only female mice were used, as described previously[73]. All mice were maintained in a specific pathogen-free (SPF) facility at the Pasteur Institute of Lille and the experimental/control groups were bred in separate cages. Mice were fed with a standard diet, given free access to water, and housed at a temperature of 22 ± 2 °C and a 35–70% humidity atmosphere, with 12 h light /12 h dark cycles. Animal experiments were performed in compliance with the European guidelines No. 68/609/EEC (approval number: #19603-2020061914271271 v6). The mouse study was

approved by the local ethics committee: Comité d'éthique en expérimentation animale (CEEA 75), Nord Pas-de-Calais, France. The model was induced by daily intradermal injection of 300 μL of HOCl into the shaved backs of mice for 5 consecutive days per week, for a duration of 6 weeks. The control group ($n = 12$) comprised PBS-receiving mice. The HOCl groups ($n = 12$ each) comprised mice receiving 100 mM $KH_2PO_4$ + 0.08% of bleach to achieve a pH of 6.2. D11 and E07 mAbs were administered by subcutaneous injection at 20 mg/kg at day minus 1 and every 5 days starting from day 5 until euthanasia. Each mAb was solubilized in sterile, endotoxin-free PBS. TD139 (Syngene, India) was administered by intratracheal instillation twice a week starting from day 21 at a concentration of 0.5 mg/kg. TD139 was dissolved in 100% DMSO and diluted in endotoxin-free PBS to obtain a final concentration of 2% DMSO and 0.25 mg/mL of TD139. Body weights were measured at day minus 1 and every two or three days until the termination day. The evolution of skin thickening was monitored by using an external caliper, every 7 days until euthanasia. Longitudinal data were analyzed using a repeated measures ANOVA for comparison between HOCl and control group (model induction) and with a one-way ANOVA for comparison between all tested items and their HOCl control group, using Dunnett's adjustment. The baseline (day minus one) was added as covariable. Statistical analyses were performed using SAS software, v9.4. Blood samples were collected from the venous sinus by retro-orbital puncture at day minus one, day 8, and 14 and processed to plasma by using lithium-heparin tubes for PK analyses and cytokine measurements. Mice were euthanized by cervical dislocation under deep $CO_2$ anesthesia. Following confirmation of death, an incision was made in the neck and the muscle layers were separated by blunt dissection and the trachea was isolated. A small incision was made in the trachea and a cannula was inserted. The airway was lavaged with 0.6 mL of PBS. PBS was left in the airway for approximately 10 s while the chest was gently massaged before the removal of bronchoalveolar lavage (BAL) fluid. This procedure was repeated twice more. Terminal blood was collected, and 100 μL was transferred in RNA-protected Animal Tubes (Qiagen) for transcriptomic analyses. The remaining volume was processed into plasma by using lithium-heparin tubes for PK analyses and cytokine measurements.

### Immunophenotyping in the mouse model of HOCl-induced SSc

For immunophenotyping, the BAL fluid from the three lavages was pooled and processed for flow cytometry. The BAL fluid was centrifuged at 400 x $g$ for 7 min at 4 °C. The supernatant was harvested and stored at −80 °C. Cell pellets were incubated for 2 min in Red Blood Cell Lysing Buffer (Hybri-Max, Sigma-Aldrich) and washed in PBS-SVF 2%-EDTA 1 mM. Cells were resuspended in 130 μL of PBS-SVF 2%-EDTA 1 mM containing 10 μg/mL Fc Block (BD Biosciences 553142, clone 2.4G2) and incubated at 4 °C for 5 min. For the Numeration of viable CD45⁺ cells in BAL fluid, cell suspensions (50 μL) were incubated with anti-CD45-APC antibody diluted 1:120 (Biolegend 103111, clone 30-F11) or the corresponding isotype control at the same dilution (Biolegend 400611, clone RTK4530) for 20 min at 4 °C, protected from light, and with propidium iodide (50 ng, Sigma-Aldrich) for 5 min. Flow-Count Fluorospheres (Beckman Coulter) were added and data were acquired on a 4-laser cytometer (CytoFLEX, Beckman Coulter) and analyzed with dedicated Kaluza software v2.1 (Beckman Coulter). For the phenotypic analysis of the cellular content in BAL fluid, cell suspensions (50 μL) were incubated with the following antibodies: anti-CD45-FITC diluted 1:200 (Biolegend 103107, clone 30-F11), anti-CD11b-APC-Cy7 diluted 1:200 (BD Biosciences 561039, clone M1/70), anti-CD3-PC7 diluted 1:200 (Biolegend 100219, clone 17A2), anti-CD4-PE diluted 1:100 (BD Biosciences 553653, clone H129.19), anti-CD8-Pacific Blue diluted 1:100 (BD Biosciences 558106, clone 53-6.7), anti-CD19-BV510 diluted 1:100 (BD Biosciences 562956, clone 1D3), anti-CD335-APC diluted 1:50 (Biolegend 137607, clone 29A1.4), or their

corresponding isotype controls at similar dilutions (FITC, PC7 and APC from Biolegend, #400634 clone RTK4530, #400617 clone RTK4530, #400511 clone RTK2758, respectively; APC-Cy7, PE, Pacific Blue and BV510 from BD Biosciences, #552773 clone A95-1, #553930 clone R35-95, #558109 clone R35-95, #562952 clone R35-95, respectively), for 20 min at 4 °C protected from light. 7-AAD (Biolegend) was added and cells were incubated for 5 min at 4 °C protected from light. Cells were washed in PBS-SVF 2%-EDTA 1 mM and fixed in 1% paraformaldehyde (PFA)/PBS solution. Data were then acquired on a 4-laser cytometer (Cytoflex, Beckman Coulter) and analyzed with Kaluza software v2.1. The gating strategy was as follows (Supplementary Fig. 14): Immune cells = CD45+ cells; B cells = CD19+ cells; NK cells = CD335+ cells; CD4+T cells = CD3+CD4+ cells; CD8+T cells = CD3+CD8+ cells. Among CD19- CD3- CD335- cells: alveolar macrophages = autofluorescence^high CD11b^{-/low} and other myeloid cells (monocytes, dendritic cells, neutrophils, and eosinophils) = autofluorescence^{low/int} CD11b^{+/high}. Statistical analyses for immune cells in BAL were determined with SAS software v9.4 using a one-way ANOVA for comparison between the HOCl-treated and control group (model induction) and with a one-way ANOVA for comparison between all tested items and their HOCl control group, using Dunnett's adjustment on log-transformed data.

### Lung and skin histology

Following completion of the BAL procedure, the thorax of mice was opened, lungs were removed, 0.5 cm skin samples were collected and all tissue samples were fixed in 4% paraformaldehyde (PFA)/PBS solution according to the standard method for paraffin embedding for histological assessment. Three 4-μm sections of skin and lung were cut from the blocks and, after rehydration, stained with Picrosirius Red to determine the level of collagen deposition. Briefly, tissue sections were stained with 0.1% Direct Red stain (Sigma-Aldrich)/0.5% Picric Acid (Sigma-Aldrich) for 60 min after deparaffinization and rehydration. Subsequently, each slide was examined and analyzed using the web-based ImageJ software v1.53t. Quantitative analysis of connective tissue deposition was performed using a threshold detection method for the grayscale image after a color deconvolution to separate the stains and the area occupied by red-stained collagen[74]. The levels of skin collagen content were analyzed using a one-way ANOVA for comparison between the HOCl-treated and control group (model induction) and with a one-way ANOVA for comparison between all tested items and their HOCl control group, using Dunnett's adjustment. For the levels of lung collagen content, the same analyses were performed on log-transformed data, adding the animal as a random factor (for replicate measurements of transversal and longitudinal slices).

The macrophage surface marker F4/80 was evaluated on 4-μm fixed lung sections by immunofluorescence assay. After permeabilization in 0.1% Triton-X 100, lung sections were incubated overnight at 4 °C with a monoclonal anti-F4/80 primary antibody (ab111101, Abcam) diluted 1:100. Then, the samples were incubated with a specific conjugated anti-mouse secondary antibody conjugated to AlexaFluor 488 (A32723, ThermoFisher Scientific) diluted 1:2000 for 1 h at room temperature. Nuclei were stained with Hoechst 33342 dye (62249, ThermoFisher Scientific). Digital images were processed with the Zeiss LSM Browser. The number of F4/80-expressing cells was quantified using QuPath software (v.0.4.1) as a percentage of the number of cells. Data were expressed as the fold-change compared to the control group. Statistical analyses were performed using a one-way ANOVA for comparison between the HOCl-treated and control group (model induction) and with a one-way ANOVA for comparison between all tested items and their HOCl control group, using Dunnett's adjustment. All statistical analyses were performed using SAS software v9.4.

### PK and ADA measurements

Plasma samples were assayed for mAb concentrations using a generic method implemented on a Gyrolab xPlore automated immunoassay

system[75]. Briefly, in-house biotinylated mouse anti-human IgG CH2 capture antibody (ThermoFisher, MA5-16929) was immobilized at 100 μg/mL on the surface of streptavidin-coated beads in Gyrolab Bioaffy CD200 (P0004180, Gyros Protein Technologies), and diluted plasma samples were loaded. Then, in-house 647-Alexafluor conjugated mouse anti-human IgG antibody (BD bioscience 555784) was used as a detection reagent at 10 nM. Sample concentrations were calculated using the Gyrolab software (Gyros Protein Technologies). Anti-drug antibodies (ADA) were measured with a generic method implemented on a Gyrolab xPlore automated immunoassay system. Mouse plasmas were coincubated with excess concentration (10 μg/mL) of D11 or E07 mAbs directly in the mixing chamber of the Gyrolab mixing CD96 (P0020455, Gyros Protein Technologies). Then, the ADA-mAb complexes were captured by biotinylated Goat anti-human IgG (Southern Biotech 2014-08) coated at 100 μg/mL on the streptavidin beads within the compact disc. Finally, 10 nM of AlexaFluor 647 goat anti-mouse IgG (H + L) cross-adsorbed secondary antibody (Thermo-Fisher A-21235) was used as a detection agent. Values were expressed as fluorescence units with a significant cut-off threshold in each experimental run determined as the mean of 3 standard deviations calculated from the mean of signals obtained from 15 control untreated mice. Statistical analyses were performed using SAS software v9.4.

## Mouse Gal-3 and cytokine assays
Mouse plasma Gal-3 was quantified using the ELISA kit DY1197 from BioTechne. Ten soluble cytokines (namely IFN-γ, IL-10, IL-1β, IL-2, IL-4, IL-5, IL-6, KC/GRO, IL-12p70, and TNF-α) were also assayed in mice plasma using the V-plex® Plus mouse proinflammatory panel 1 K15048G from Meso Scale (Discovery, Rockville, USA). Statistical analyses were performed using SAS software v9.4 on log-transformed data using a mixed ANCOVA model (baseline as covariate and animal as random) for comparison between the HOCl-treated and control groups (model induction) and using a mixed ANCOVA model for comparison between all tested items and their HOCl control group, using Dunnett's adjustment.

## Mouse RNAseq studies and canonical pathways
Total RNA was extracted from whole blood collected in RNA protect tubes (Qiagen) and proceeded for extraction using the RNeasy Protect Animal blood kit with the optional on-column DNAse I digestion. After extraction, total RNA samples were quantified on Nanodrop 2000 (ThermoFisher Scientific) and analyzed on Bioanalyzer (Agilent) using RNA 6000 Nano chips to determine RNA quality (RIN). Total RNA matching the QC criteria (RIN > 7) was included in the study. At least 250 ng of purified RNA samples were used as input for the TruSeq Strand-specific RNA-seq (poly A selection and globin depletion) according to the manufacturer's instructions (Illumina). Following the preparation of libraries, samples matching the requested criteria were sequenced on a NovaSeq 6000 sequencer (Illumina). Sequencing parameters were adapted for 2 × 150 bp configuration with at least 20 M pair-end reads per sample. Raw reads were processed using the in-house RNAExp Expression pipeline 2.1 and the quality was assessed using FastQC tools v.11.18. Then, reads were trimmed for adaptors using Cutadapt and trimmed reads were aligned to the mouse mm39 reference genome (https://www.ncbi.nlm.nih.gov/assembly/GCF_000001635.27/) using the STAR aligner. Aligned data were evaluated for quality using several quality metrics (e.g., mapping rate, %rRNA, %mRNA) using Samtools and Picard tools. Then, aligned reads per gene were quantified using FeatureCount. Read counts were normalized by the variance stabilizing transformation vst function from the DESeq2 R package v1.30.0. 55,416 genes were detected in the data. Genes not coding for a protein were filtered (29,947 genes removed). Those with 0 count over all the samples or having an expression level below 1 in

more than 95% were filtered. The final RNAseq dataset comprised 10,508 genes.

Partial Least-Squares-discriminant analysis (PLS-DA) performs a dimension reduction by preserving the covariance between the expression matrix and the outcome variable. It was applied to represent the samples while maximizing their separation with regard to the outcome. In order to identify genes differentially expressed between groups, a linear model (lmFit function from limma v3.46.0R package) based on the vst transformation gene expression dataset was applied. The resulting $p$-values were adjusted for multiple hypothesis testing and filtered to retain differentially expressed genes with a FDR adjusted $p$-value ≤ 0.05 and a |FC| value ≥ 1.3. Ingenuity pathway analysis was applied to determine the most significantly dysregulated canonical pathways with Benjamini-Hochberg FDR and FC values as above.

## Reporting summary
Further information on research design is available in the Nature Portfolio Reporting Summary linked to this article.

## Data availability
Human raw data are the property of the PRECISESADS consortium and are protected under the European General Data Protection Regulation (GDPR). PRECISESADS data are hosted by ELIXIR Luxembourg and available under controlled access at the permanent link: https://doi.org/10.17881/th9v-xt85. The access procedure is described on the ELIXIR data landing page. The PRECISESADS Consortium is committed to secure patient data access through the ELIXIR platform. This commitment was formerly given by written agreement to all patients at the end of the project and to the involved Ethical Committees. Access can be requested from the data stewardship team of ELIXIR Luxembourg via lcsb-datastewards@uni.lu. The future use of the Project database was framed according to the scope of the patient information and consent forms, where the use of patient data is open to scientific research in autoimmune diseases, not-for-profit use only. ELIXIR reviews the applicant's requests and prepares the Data Access Committee's decisions on access to Data, communicates such decisions to the Data Providers, who have 10 days to exercise their right to veto; otherwise, access is granted to the User. Sequencing data in the mouse HOCl model are available on the GEO repository under the accession number GSE226063. The source data underlying all figures and supplementary figures are provided as a Source Data file. Source data are provided in this paper.

## Code availability
The code for the clustering of patients has been deposited to Git Hub (https://github.com/psBiostat/GAL3_PAPER.git) and linked to *Zenodo*[62].

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

## Acknowledgements

The research leading to these results has received support from the Innovative Medicines Initiative Joint Undertaking under the Grant Agreement Number 115565 (PRECISESADS project), resources of which are composed of financial contributions from the European Union's Seventh Framework Program (FP7/2007–2013) and EFPIA companies' in-kind contribution. The authors thank all colleagues who participated at any given time in this global project. Iuliana Botez, Nicolas Faucher, Benjamin Chanrion, Adeline Giganti, and Jennifer Serriere-Di Bartolo contributed to paving the way for strategic thinking at the earliest stages of the project. The authors also thank Philippe Pastoureau, Laurence Laigle, Philippe Moingeon, Fabien Schmidlin, Jeanne Allinne, and Ross Jeggo for supporting the project. Laurence Laigle and Jeanne Allinne are also acknowledged for critical review of the manuscript.

## Author contributions

Authorship follows the inclusion & ethics in global research recommended by the Nature Portfolio editorial policies. C.O. and G.R. designed and supervised the discovery of Gal-3 blocking mAb. M.L.G. and J.S. conducted the mAb discovery experiments. E.N. and B.F. designed and coordinated the production and analytical qualification of mAb. S.B. and T.Q.N. were in charge of the design and interpretation of the binding kinetics experiments by SPR. L.C. and P.S. coordinated the collection of data from the SSc PRECISESADS cohort. S.C. performed queries on Gal-3 interactants in bioinformatic databases. D.L., S.S., M.J., and T.G. designed, performed, and interpreted experiments in the HOCl model. A.C. supervised and interpreted the Gal-3 and cytokine measurements in the mouse HOCl model. B.N. performed all statistical analyses referring to the HOCl study. A.L. and F.L. defined the methodology and interpretation of pharmacokinetic data. N.P. designed and supervised transcriptomic studies in the HOCl model. S.H., E.D., and J.V. executed and interpreted pathway analyses. P.S. and S.E. processed RNAseq data and performed subsequent statistical analyses in the mouse HOCl model. F.D. designed and supervised the global project strategy and wrote the manuscript with the input of all co-authors.

## Competing interests

Servier is the owner of the described therapeutic Gal-3 mAbs for which a patent application (EP22305372) has been filed. G.R., M.L.G., and J.S. are regular employees of Mabqi, a company providing contract research services and having received fees from Servier for the generation of Gal-3 neutralizing mAbs (contract number CT0081211). D.L., S.S., M.J., and T.G. are employees of the Université de Lille, which received fees from Servier for conducting evaluations in the HOCl model (contract number CT0089360). D.L. is or has been a consultant for Actelion, Boehringer Ingelheim, Octapharma, CSL Behring, Takeda, and Servier. The following authors: C.O., P.S., S.H., L.C., A.C., S.B., E.N., B.N., B.F., A.L., F.L., S.E., S.C., N.P., T.Q.N., and F.D. were regular employees of Servier at the time the research project. E.D. was a Ph.D student financed by Servier at the time of the research project. J.V. was a post-doc financed by Servier at the time of the research project.

## Additional information

[1]Servier R&D Center, Biomarker Assay Development, Translational Medicine, Gif-sur-Yvette, France. [2]Servier R&D Center, Biomarker Biostatistics, Gif-sur-Yvette, France. [3]Mabqi SAS, Montpellier, France. [4]U1286 INFINITE, Institute for Translational Research in Inflammation, Lille University, Gif-sur-Yvette, France. [5]Inserm, Lille, France. [6]Servier R&D Center, Neurosciences and Immuno-inflammation Therapeutic Area, Gif-sur-Yvette, France. [7]Servier R&D Center, Clinical Biomarker Development, Translational Medicine, Gif-sur-Yvette, France. [8]Servier R&D Center, Protein Sciences, Gif-sur-Yvette, France. [9]Servier R&D Center, Structural Sciences, Gif-sur-Yvette, France. [10]Servier R&D Center, Preclinical Biostatistics, Quantitative Pharmacology, Gif-sur-Yvette, France. [11]Servier R&D Center, DMPK Department, Translational Medicine, Gif-sur-Yvette, France. [12]Servier R&D Center, Computational Medicine, Gif-sur-Yvette, France. [13]Servier R&D Center, Molecular Genomics, Gif-sur-Yvette, France. [14]Lille University Hospital, Department of Internal Medicine and Clinical Immunology, Reference Center for Rare Systemic Autoimmune Diseases, North and North-West France (CeRAINO), Lille, France. [15]These authors contributed equally: Céline Ortega-Ferreira, Perrine Soret. ✉e-mail: frederic.deceuninck@servier.com

