## [Peer Review file · Nature Communications]

REVIEWER COMMENTS

Reviewer #1 (expert in antibody development, phage display panning):

In their manuscript "Galectin-3 blockade as a strategy for the treatment of systemic sclerosis" Dr. Ortega-Ferreira et al. describe a transcriptome biomarker signature illustrates that Galectin-3 and its interactome correlate with severe systemic sclerosis in a large patient cohort. They developed a human monoclonal antibody that blocks the binding activity of galectin-3, and that cross-react to galectin-3 of relevance to several animal model systems. Such antibodies were investigated in a mouse model of systemic sclerosis and were shown to reduce clinical symptoms. Effects on immune cells were defined. Overall, the manuscript describes a coherent story from early biomarker discovery to early pre-clinical assessment in vivo of a lead candidate antibody.

The authors develop and test a human IgG1. Is this the intended isotype of a clinical antibody or do the authors intend to exploit another isotype with limited FcR binding? How will the choice of isotype/subclass affect the biological outcome? If another antibody format is intended for a final product, data on this format should be presented.

The affinity of the identified human antibodies was determined using SPR under conditions of a high level of immobilized antibody. This may negatively affect the validity of the determined reaction rate and affinity constants. It is very strongly suggested that the assay is repeated using substantially lower levels of immobilized antibody.

The authors have performed a limited set of specificity tests with similar antigens. To ensure downstream development of the antibody, a larger set of tests should be performed to define its developability (see e.g. Jain et al. PNAS 2017). The authors should present results of a range of such tests.

The authors ought to provide a toxicology assessment of using a galectin-3-neutralizing antibody in a clinical setting. Furthermore, were any side effects identified in the mouse model?

In the interest of reproducible open science, the authors must report the sequences of the heavy and light chain variable domains of the two galectin-3 specific antibodies investigated in this study. The nucleotide sequences of these binders should be deposited in GenBank.

In the reporting summary it was stated (page 4) that “D06 and D11 mAbs were produced in CHO-K1 cells (received from ATCC)”. Is this a typo or how was E07 produced?

Reviewer #2 (expert in transcriptional networks):

General:

In this paper, Ortega-Ferreira et al conduct the detailed analysis of the blood transcriptome for the patients of systemic sclerosis (SSC), which is an autoimmune, inflammatory and fibrotic disease. In this disease, Galectin-3 (Gal-3), which is a beta-galactoside-binding lectin, plays an important role in many aspects of the pathological processes. The level of Gal-3 is also associated with the symptoms and prognosis of the patients. Making use of the detailed transcriptome data collected for PRECISESADS cross-sectional study, the authors inspected the transcriptional profiles of the patients depending on their clinical phenotypes. They further attempted to develop a monoclonal antibody in order to give a yet new treatment option for SSC. They successfully develop two monoclonal antibodies (mAbs), E07 and D11. These antibodies turned out to possess equal or better molecular characteristics, when evaluated by SPR, ELISA and other in vitro assays, than the currently most representative anti-SSC drug, TD139. While the shorter half-life of TD139 imposes a substantial problem on the administration of this drug other than the respiratory inhalation, the E07 and D11 antibodies could be administered by intradermal injections. In a treatment model of mice, in which SSC was induced by hypochlorous acid (HOCl mouse), the substantial improve of the symptoms were observed for skin thickening, lung and skin collagen disposition and plasma inflammatory cytokine levels. The blood cell profiles also showed that the HOCl-induced molecular alteration in whole blood cells could be almost completely reversed.

Overall, I think this is an important paper, conveying substantial novel insights for molecular subtyping of the patients of SSC. Above all, I appreciate that the authors successfully developed promising mAbs as a promising drug candidate. This paper should open a new field for the treatment of SSC, for which there is still no decisive option is available. The followings are the comments which I hope the authors may find helpful, when they would like to further strengthen their claims.

Major points:

1. RNA seq analysis could be done at a single cell level, firstly for the mouse model. From the present analysis, the mode of action it is not totally clear, that is, how the Galectin-3 inhibition realizes its function in a total blood cell system. It is also desirable to conduct the same single cell analysis for human samples, at least for r someepresentative cases of C1, C2 and C3. Without such an analysis, I'm afraid that a gap would remain between the first and the second parts of the paper, which is on the detailed transcriptome analysis of the patients and the characterization of the mAbs, respectively.

2. I'm concerned that there is no analysis presented for pulmonary hypertension, which I believe is one of the topical causes of death in SSc. I totally agree the pulmonary fibrosis may not be always the issue of the primary focus, still, I personally think that pulmonary hypertension should be no less important. Even it is not possible to analyze the right heart pressure of a mouse in the authors' laboratory, it may be possible that they can analyze the remodeling of the vascular endothelium in the lung, for example, by examining the lung cell-dispersed specimen by FACS. The authors can also consider the right heart/left heart ratio.

3. Generally, the assessment of the lung etiology is not always sufficiently detailed. More precisely, only collagen accumulation and BAL (bronchoalveolar lavage) are examined in this paper. While BAL mainly represents alveolar macrophage, there are many other indicators for evaluating pulmonary fibrosis. For example, it is also important to analyze interstitial macrophage as well. For this purpose, the authors could analyze the lung cell-dispersed specimen analyzed by FACS, again.

Minor Points:

4. In addition to the HOCl mice, there are a number of other types of SSc model mice. I don't know if there are any special circumstances, such as the fact that the endothelium of the blood vessels in this HOCl model may not be damaged so much. Please enrich the discussion.

5. First paragraph: The manner of describing the transcriptome analysis was somewhat confusing to me. Strictly speaking, although the description indicates the "interactome" analysis", the data is on the transcriptome and the authors do not analyze any protein-protein/lectin interactions directly. The descriptions in the Method section is rather precise.

6. Is not there any possibility that further inspection of the collected data and the bioinformatics framework would lead to identification of yet novel molecular interaction of novel aspects of transcriptional network. related to Gal-3?

7. According to Table1, most of the participants (98.6%) involved in the current cohort are Caucasian/White. I wonder if there is any difference between races or other ethnic backgrounds? Is it possible to analyze the profiles by using some public RNA-seq datasets of SSc patients, if any?

8. Is there any other sign of side effects in using mAb for Gal-3 blockade in the mouse model?

Reviewer #3 (expert in galectins):

The authors found that the mRNA levels of interactants of galectin-3 (Gal-3) correlated with systemic sclerosis (SSc) disease severity, pulmonary and cardiac malfunctions, neutrophilia and lymphopenia. Moreover, by phage display screening, they identified 66 scFv that bound to recombinant human Gal-3 and developed two mAb composed of human IgG1 Fc. They showed that they also bound to mouse Gal-3. They demonstrated that these mAbs decreased the pathological features of SSc in a mouse model induced by HOCl.

Major comments:

1. The development of biological therapeutic agents targeting Gal-3 is remarkable, since this protein has been implicated in a large number of diseases; several companies have been working for many years on development of galectin-3 inhibitors as drugs, but so far, only one has been shown to be efficacious (and FDA-approved) and it is a small molecular weight compound. The development of mAb as a Gal-3-targeting drug is challenging, as this protein functions both extracellularly and intracellularly and, for a given disease, so far, the existing information does not allow a quantitative differentiation of its contributions in these two compartments. If the protein contributes to the disease mainly through its intracellular functions, then theoretically, mAbs would not work, as they are not able to access the intracellular space. The data on the therapeutic effects of the two mAbs in a mouse model of SSc reported in this manuscript appear convincing. However, there are a number of issues that need to be addressed.

The authors need to clearly address whether these mAbs function by inhibiting the binding of extracellular Gal-3 to cells, thus inhibiting the cellular responses induced by the protein. On the other hand, these mAbs can potentially crosslink Gal-3 on the cell surfaces, resulting in suppressing of cellular responses (such as by crosslinking inhibitory receptors). To address these possibilities, the authors need to examine the amount of cell surface Gal-3 in various tissues, before and after the treatment with the mAbs. Another possible mechanism of action of these mAbs is their formation of immune complexes, which can have immunosuppressive functions. In this case, the suppressive functions of Gal-3-neutralizing mAbs can only be viewed as the properties of the immune complexes and do not reflect the functions of endogenous Gal-3.

In addition, one critical set of experiments to add involves the use of Gal-3 KO mice. This reviewer assumes this strain of mice would have a less severe phenotype, which reflects the function of endogenous Gal-3 in the pathogenesis of SSc. The author could compare the differences in the responses to HOCl between WT and KO mice to those between mice treated with mAbs and control antibodies. Moreover, the demonstration that the mAbs do not have any effect on Gal-3 KO mice would be a strong support that they work by targeting Gal-3.

2. The authors missed one critical control and that is the use of control hIgG1. An ideal one would be that binds a serum component in mice that is known not to have a pathological role in SSc. This is important, as the authors used chimeric mAbs with a human IgG1 Fc portion and these has the potential to induce anti-human IgG antibodies in mice, after multiple subcutaneous injections, which can form immune complexes with the mAbs. In this regard, the authors should determine whether the mAbs induced anti-human IgG1 antibodies in their experiments.

3. The authors showed that in their mouse model, the plasma levels of Gal-3 were significantly increased after pathological induction with HOCl, but fully reversed after D11, E07 and TD139 treatments. This needs to be clarified. One does not expect the total Gal-3 levels would be reduced after the treatment with these mAbs. In fact, the total amounts should become higher, because the immune complexes are expected to be more stable in circulation. One would not expect the Gal-3 levels to be reduced by TD-139 either. The authors need to measure the total Gal-3 amount in the serum.

4. The authors generated a list of known Gal-3 partners through bioinformatic queries within Ingenuity Pathway Analysis (IPA) and GPS-Prot databases. Gal-3 is known to bind to a large number of glycoproteins through lectin-carbohydrate interaction and also intracellular mediators through protein-protein interaction. Thus, the authors need to separate the interactants into different categories and also comment on the issue of the lack of evidence for the role of many of these interactants in the function of Gal-3.

5. In addition, many of the glycoproteins that Gal-3 has been reported to interact with might not be relevant to the pathological processes, as it might not encounter them in vivo. Thus, the rationale of relating galectin-3 interactants levels to the pathological processes in SSc should be explained, including whether interaction between galectin-3 and their partners listed is critical for development of SSc. In this regard, the authors could analyze Gal-3 WT and KO mice to establish the relevance of these interactants.

Other comments:

1. Figure 1: The logic of clustering patients into three subgroups should be explained (e.g., why choose three clusters and condition of supervised selection and clustering) and the code for the clustering should be deposited to github.

How were the Gal-3up and Gal-3down gene lists determined? Was that based on their expression levels in the C3 groups?

2. Figure 2: Are the Gal-3up scores in the subset of patients a mean of all Gal-3up genes in individual patients? How were the immune cell population counts defined? What genes were used to calculate the "counts"? The code used should be deposited to github.

3. Figure 3: The author may show a volcano plot for the differential expression gene profile by comparing the two groups as mentioned in the text.

4. The authors need to provide explanations on why the two mAbs have different effects.

Reviewer #4 (expert in systemic sclerosis):

Comments for authors,

In this manuscript, the authors performed cluster analysis using RNA sequencing using whole blood, and Gal-3 and its related factors were related to clinical parameters in SSc (systemic sclerosis) patients. Then, they discovered novel Gal3-neutralizing antibodies and used them for the HOCl-induced SSc mouse model to evaluate the therapeutic potential of their Gal-3-neutralizing antibodies. Although these contents are interesting, there are some concerns about the interpretation of the results and clinical aspects as listed below.

[Major points]

1.

Whole blood samples were analyzed in this study. They highlight that Gal-3 expression is strongly correlated with neutrophilia and lymphopenia. However, this might require careful interpretation. Given that whole blood is rich in neutrophils and lymphocytes, changes in their relative proportions must greatly affect their RNAseq results, especially in case Gal-3 is expressed in neutrophils or lymphocytes. In fact, there seems to be literature indicating that Gal-3 are expressed and function in neutrophils. For these reasons, I think they should be a little more cautious when discussing the correlation with neutrophils/lymphocytes from a clinical point of view.

2.

There are further concerns related to blood cells (e.g. data in Fig2D etc.). They indicated that neutrophilia and lymphopenia are associated with cluster 3 and associated with disease activity. Cytopenia in patients with SLE, another collagen disease, is an important activity indicator. However,

regarding SSc, there must not be a strong consensus that neutrophilia and lymphopenia are useful in the evaluation and monitoring of SSc patients in clinical practice. This is one of the reasons why I am concerned about the previous point regarding whole blood sample analysis and I feel uncomfortable with the description of neutrophilia and lymphopenia.

3.

As the authors have already noted in the introduction, Gal-3 is well-documented to be important in many fibrotic diseases and systemic sclerosis. Regarding the potential of Gal-3 as a therapeutic target for SSc, as they indicated in the title, there are some papers showing that bleomycin-induced lung fibrosis was reduced in Gal-3 deficient mice or by the treatment with TD139, an inhaled Gal-3 inhibitor. Furthermore, as described in the introduction, a phase I/IIa clinical trial was also conducted with TD139 in IPF patients. I agree that there are new points in terms of new analysis methods, the HOCl model, and the discovery of novel antibodies. However, in light of those previous reports, focusing on Gal-3 and SSc in this work may not be so novel. In this sense, I feel unsatisfactory with whether the title properly expresses the content including the novelty.

4.

Their functional molecular cluster analysis revealed that C3 contained a larger proportion of diffuse cutaneous SSc patients (dcSSc) and displayed a higher inflammation profile than other clusters. In addition, regarding clinical manifestations, C3 patients showed more pulmonary fibrosis, arrhythmias, and higher disease activity compared with other clusters. Moreover, they investigated the importance of the Gal-3 fingerprint by examining the relationship between the Gal-3 up and Gal-3 down scores with features of SSc. As a result, it became clear that the expression level of a high number of Gal-3 interactants was strongly associated with impaired vital organ function in SSc patients and with the level of disease severity. As the authors may know, the antibody profile of systemic sclerosis is known to correlate well with clinical characteristics, and it is expected that the pathology should differ for each antibody. For example, complications of ILD (SSc-ILD) are frequently seen in SSc patients with anti-Scl70 antibodies. On the other hand, anti-centromere antibody-positive patients are frequently classified as lcSSc, and they sometimes have complications of pulmonary hypertension with relatively high frequency. Considering these facts, it is difficult to determine whether the correlation between severity and high activity in the C3 cluster is purely due to complications and fibrosis, or due to differences in pathology depending on antibody profiles. In other words, for example, if the observed characteristics of C3 cluster are based on the pathological features of anti-Scl70 antibody-positive, it is possible that the patients without anti-Scl70 antibody but classified to C3 cluster do not correlate well with activity or complications. In fact, looking at Table 2, about 70% of C3 patients are anti-Scl70-positive, so they should be careful about the effects of each antibody. If I can propose the investigation methods to clarify these possibilities, the results obtained in Figures 1 and 2 can be analyzed again in a similar way by separating each category of patients (C1-C3) into anti-Scl70-positive patients and anti-centromere-positive patients. If a similar trend is observed in these analyses, the results they have already shown should be independent of antibody differences.

5.

There is one more point to make regarding auto-antibodies. As the authors know, representative auto-antibodies specific to SSc include anti-Scl70, centromere, and RNA polymerase III antibodies. In Table 2, there is no description of RNA polymerase III antibodies. Among the patients enrolled in this project, were any patients positive for this antibody? This study seems to include many Caucasians. The proportion of patients with RNA polymerase III antibodies should not be small in Caucasians, and this antibody is well-known to be a risk factor for rapid skin sclerosis and renal crisis. In this study, they argue that Gal-3 and its interacting factors are deeply involved in disease activity and inflammation. Therefore, I think that this should be examined and discussed.

6.

There are more questions in terms of neutrophils and lymphocytes. As they may know, there are various treatments for SSc, such as steroids and other immunosuppressants. In Table 1, they described that about 25% of each of the patients in this study were treated with steroids and immunosuppressants. Patients with high disease activity may be taking these therapies. Steroids often lead to neutropenia. And, immunosuppressants sometimes lead to lymphopenia. Considering these, it is likely that the therapeutic agent may have influenced the features seen in the C3 cluster such as neutrophilia and lymphopenia. Therefore, they may not be able to rule out the possibility that these correlations are not intrinsic and may even reflect the effects of therapeutic agents.

[Minor points]

7.

They used the HOCl mouse model for the Gal3-neutralizing antibody, but there are other models such as the bleomycin model. Why did you use the HOCl model this time? Have you examined the BLM model in the same way?

8.

Data for IL-5 and IL-6 are shown in Figure 5, etc. If so, I think it is necessary to mention in the explanation or introduction how these cytokines are important in the pathology of SSc. In fact, in some countries, biologics targeting IL-6 have already been clinically applied to SSc-ILD.

9.

In Fig. 5, cytokines are reduced by E07. Is this related to changes in lymphocyte counts? And/Or does E07 somehow directly affect cytokine production in Gal3-expressing cells? This paper has many descriptive parts. Therefore, I can advise that the quality of this paper would be improved if it is possible to add an examination using in vitro experiments to address these questions.

10.

They showed that E07 treatment significantly decreased the levels of cytotoxic CD8+ T cells and NK cells. This is interesting. As the authors may know, cytotoxic CD8+ T cells and NK cells are very important cells from the point of view of tumor immunity. This means that if E07 can be used in clinical practice in the future, it may exacerbate the progression of cancer in cancer-bearing patients. As mentioned above, there are reports that malignant tumors occur at a high rate in anti-RNA polymerase III antibody-positive patients. I think it would be better to discuss this concern.

11.

In terms of the Methods section, the mouse model description "Mouse model of HOCl-induced SSc" seems too long. This includes many experiments such as FCM and BAL. I think that the description of the mouse model and the individual analysis methods should be described separately.

REVIEWER 1: Expert in antibody development, phage display, phage display panning

In their manuscript "Galectin-3 blockade as a strategy for the treatment of systemic sclerosis" Dr. Ortega-Ferreira et al. describe a transcriptome biomarker signature illustrates that Galectin-3 and its interactome correlate with severe systemic sclerosis in a large patient cohort. They developed a human monoclonal antibody that blocks the binding activity of galectin-3, and that cross-react to galectin-3 of relevance to several animal model systems. Such antibodies were investigated in a mouse model of systemic sclerosis and were shown to reduce clinical symptoms. Effects on immune cells were defined. Overall, the manuscript describes a coherent story from early biomarker discovery to early pre-clinical assessment in vivo of a lead candidate antibody.

The authors develop and test a human IgG1. Is this the intended isotype of a clinical antibody or do the authors intend to exploit another isotype with limited FcR binding? How will the choice of isotype/subclass affect the biological outcome? If another antibody format is intended for a final product, data on this format should be presented.

Thank you for your comment that allows to correct an omission regarding the format of the antibodies used in this study. D11 and E07 mAbs have been mutated in their Fc region as LALA antibodies, two mutations known to reduce effector functions and aiming at reducing potential immunogenicity in clinical trials. The correct format has been amended in the revised manuscript (Materials and Methods, section 'ScFv reformatting to full-length IgG', lines 573-575), and 2 references related to the advantages of introducing LALA mutations in human and mouse IgG have been added (64, 65). We also confirm that the IgG1 isotype is the intended format for further development.

The affinity of the identified human antibodies was determined using SPR under conditions of a high level of immobilized antibody. This may negatively affect the validity of the determined reaction rate and affinity constants. It is very strongly suggested that the assay is repeated using substantially lower levels of immobilized antibody.

We agree, a low capture level allows to avoid the problems of mass transport limitation and modern approaches recommend to have a Rmax value of 100-150 RU. A Rmax value below 200 RU is still considered a low density of ligand, except for low molecular weight ligands for which a Rmax value below 20 RU is recommended. In our experiments the Rmax ranged between 74 and 190 RU, close to the recommended values. If the level of immobilization is too high, the risk is that the signal becomes linearized, because of the presence of a mass transport limitation, which is not the case in our experiments. Ditto for the dissociation, at a high level of immobilization/capture, a risk of rebinding and slow dissociation exists, which is also not the case in our experiments.

All sensorgrams corresponding to the data reported in Figure 4d have been added in the revised manuscript as Supplementary Fig. 8. Below, we also provide the key parameter values for each SPR condition, allowing to show the strength of the results. These values have also been reported in the source data file.

D11 mAb							
	ka (1/Ms)	kd (1/s)	KD (M)	Rmax (RU)	kt (RU/Ms)	tc	U value
human Gal-3 (FL)	1.66E+06	2.47E-02	1.49E-08	130.2	1.61E+08	5.18E+07	2
mouse Gal-3 (FL)	6.35E+05	1.78E-03	2.80E-09	183.6	3.73E+07	1.20E+07	1
hGal-3 (CRD)	1.62E+06	4.28E-02	2.64E-08	169.5	2.95E+08	9.50E+07	2
rat Gal-3 (CRD)	2.11E+06	1.80E-02	8.55E-09	179	2.47E+08	7.95E+07	1
cyno Gal-3 (CRD)	3.38E+06	3.86E-02	1.14E-08	169.4	1.52E+08	4.91E+07	12
dog Gal-3 (CRD)	9.48E+05	2.35E-02	2.47E-08	131.5	2.31E+11	7.42E+10	2
E07 mAb							
	ka (1/Ms)	kd (1/s)	KD (M)	Rmax (RU)	kt (RU/Ms)	tc	U value
hGal-3 (FL)	3.24E+06	2.92E-02	9.00E-09	189.9	7.72E+07	2.48E+07	9
mGal-3 (FL)	9.83E+05	4.07E-03	4.14E-09	149.9	2.84E+07	9.13E+06	2
hGal-3 (CRD)	2.61E+06	3.77E-02	1.44E-08	121.4	1.59E+08	5.10E+07	4
rat Gal-3 (CRD)	3.31E+06	4.97E-02	1.50E-08	88.1	7.59E+07	2.44E+07	9
cyno Gal-3 (CRD)	7.54E+06	5.35E-02	7.09E-09	115.8	9.65E+07	3.11E+07	15
dog Gal-3 (CRD)	7.46E+05	1.60E-02	2.15E-08	74.1	2.06E+12	6.64E+11	7

The authors have performed a limited set of specificity tests with similar antigens. To ensure downstream development of the antibody, a larger set of tests should be performed to define its developability (see e.g. Jain et al. PNAS 2017). The authors should present results of a range of such tests.

Thank you for this suggestion. In addition to Gal-1 and Gal-7, we have performed an additional set of experiments to check the selectivity of D11 and E07 mAbs towards Gal-2, -4, -8, -10 and -14. Recombinant Gal-9 was excluded from the analysis because of low quality control.

This figure has been added in the revised manuscript as Supplementary Figure 7. The methods section has been updated with these additional galectins (lines 607-611), and references of the antibodies have been added in the reporting summary file. These new results are commented in the results section (lines 236-237 and 240-241) and in the discussion (lines 403-406). The moderate binding observed for Gal-14 is not an issue. This protein is located intracellularly and its expression is restricted to placenta (references 45 and 46 added in the discussion), implying that this binding is unlikely to occur in biological systems or in a clinical setting, and that it does not represent an interpretation bias in the present study. We also performed additional temperature stability studies of D11 and E07 mAbs in comparison with a reference IgG1 antibody approved in the EU, namely trastuzumab (herceptin). These results have been implemented in the revised manuscript as Supplementary Fig. 6, together with methods and results added in the corresponding sections (lines 597-604, and 228-232 respectively). Both mAbs displayed size exclusion chromatographic behaviors similar to trastuzumab when subjected to forced degradation at 40°C during 2 weeks (Supplementary Fig. 6b), and relatively comparable onset temperatures of degradation, close to 60-61°C for D11 and E07, *versus* around 64°C for trastuzumab (Supplementary Fig. 6c).

The authors ought to provide a toxicology assessment of using a galectin-3-neutralizing antibody in a clinical setting. Furthermore, were any side effects identified in the mouse model?

In life observations were recorded daily and body weight variation was measured every two days starting from day minus 1 to day 42 in each group of the HOCl mouse study. No change in body weight

was observed in the HOCl control group compared to the vehicle control group, neither in mice treated with D11, E07 or TD139 compared to the HOCl control group. No abnormal behavioural sign was observed in the anti Gal-3 treated groups. There was no mortality in any of the anti Gal-3 treated groups. In an independent study conducted for 9 days in rats, subcutaneous administration of D11 or E07 up to 200 mg/kg did not induce any histological evidence of systemic toxicity at up to 200 mg/kg (microscopic examination of Heart, Ileum, Kidneys, Liver, Lungs, and Spleen), and there was no abnormal macroscopic finding. There was no mortality in any of the anti Gal-3 treated groups.

In the interest of reproducible open science, the authors must report the sequences of the heavy and light chain variable domains of the two galectin-3 specific antibodies investigated in this study. The nucleotide sequences of these binders should be deposited in GenBank.

Nucleotide sequences have been filed in the patent application (EP22305372) in March 2022. The scientific community will be able to freely access and reproduce studied antibodies from the sequences disclosed in the above patent application.

In the reporting summary it was stated (page 4) that “D06 and D11 mAbs were produced in CHO-K1 cells (received from ATCC)”. Is this a typo or how was E07 produced?

Thank you for your vigilance, this was a typo. It has been corrected in the reported summary: D11 and E07 mAbs were produced in CHO-K1 cells (received from ATCC)”

REVIEWER 2: Expert in transcriptomics and transcriptional networks

General: In this paper, Ortega-Ferreira et al conduct the detailed analysis of the blood transcriptome for the patients of systemic sclerosis (SSC), which is an autoimmune, inflammatory and fibrotic disease. In this disease, Galectin-3 (Gal-3), which is a beta-galactoside-binding lectin, plays an important role in many aspects of the pathological processes. The level of Gal-3 is also associated with the symptoms and prognosis of the patients. Making use of the detailed transcriptome data collected for PRECISESADS cross-sectional study, the authors inspected the transcriptional profiles of the patients depending on their clinical phenotypes. They further attempted to develop a monoclonal antibody in order to give a yet new treatment option for SSC. They successfully develop two monoclonal antibodies (mAbs), E07 and D11. These antibodies turned out to possess equal or better molecular characteristics, when evaluated by SPR, ELISA and other in vitro assays, than the currently most representative anti-SSC drug, TD139. While the shorter half-life of TD139 imposes a substantial problem on the administration of this drug other than the respiratory inhalation, the E07 and D11 antibodies could be administered by intradermal injections. In a treatment model of mice, in which SSC was induced by hypochlorous acid (HOCl mouse), the substantial improve of the symptoms were observed for skin thickening, lung and skin collagen disposition and plasma inflammatory cytokine levels. The blood cell profiles also showed that the HOCl-induced molecular alteration in whole blood cells could be almost completely reversed. Overall, I think this is an important paper, conveying substantial novel insights for molecular subtyping of the patients of SSC. Above all, I appreciate that the authors successfully developed promising mAbs as a promising drug candidate. This paper should open a new field for the treatment of SSC, for which there is still no decisive option is available. The followings are the comments which I hope the authors may find helpful, when they would like to further strengthen their claims.

Major points:

1. RNA seq analysis could be done at a single cell level, firstly for the mouse model. From the present analysis, the mode of action it is not totally clear, that is, how the Galectin-3 inhibition realizes its function in a total blood cell system. It is also desirable to conduct the same single cell analysis for human samples, at least for some representative cases of C1, C2 and C3. Without such an analysis, I'm afraid that a gap would remain between the first and the second parts of the paper, which is on the detailed transcriptome analysis of the patients and the characterization of the mAbs, respectively.

Thank you for your comment. The PRECISESADS cross-sectional study (registered as NCT02890121 in ClinicalTrials.gov) has assessed the transcriptomic data of patients with various autoimmune diseases including patients with SSc from the whole blood compartment. Single-cell analyses were not part of the study protocol. We thus stuck on whole blood analysis in the mouse model of HOCl-induced SSc, with the objective to maintain consistency for the reconciliation of preclinical and clinical data, and evaluate the translational strength of our therapeutic mAbs. Remarkably, many of the key entities

identified as pathological drivers in the mouse HOCl model were represented in one or more clusters of the PRECISESADS SSc patients (see Figure 7), highlighting the high relevance of this preclinical model to reflect the human pathology using a whole blood transcriptomic approach. By comparing pathways in the human clusters and the mouse model, we were able to show that E07 mAb demonstrated remarkable predictive efficacy in reverting 11 out of 27 entities targeted in C1 and 11 out of 23 entities targeted in C3. Furthermore, all seven biological entities predicted as modulated by E07 mAb in C2 were common to C1 and C3.

Further related to the relevance of this approach, it is known that most cell types implicated in autoimmune pathologies infiltrate tissues from blood. Previous analyses in the PRECISESADS cohort have shown that whole blood molecular signatures correlate very well with disease activity (Barturen *et al.*, Integrative analysis reveals a molecular stratification of systemic autoimmune diseases. Arthritis Rheum 2021, cited in the current manuscript as reference 55). Inversely and counter-intuitively, single cell analyses in SSc patients did not reveal major global and consistent changes between SSc patient groups and healthy subjects in blood or skin immune subsets (Gur *et al.* Cell 2022).

2. I'm concerned that there is no analysis presented for pulmonary hypertension, which I believe is one of the topical causes of death in SSc. I totally agree the pulmonary fibrosis may not be always the issue of the primary focus, still, I personally think that pulmonary hypertension should be no less important. Even it is not possible to analyze the right heart pressure of a mouse in the authors' laboratory, it may be possible that they can analyze the remodeling of the vascular endothelium in the lung, for example, by examining the lung cell-dispersed specimen by FACS. The authors can also consider the right heart/left heart ratio.

We agree that pulmonary arterial hypertension (PAH) is an important issue in SSc. The prevalence of SSc-associated PAH varies between 8 and 12% (Saygin *et al.*, Open Access Rheumatol. 2019) and PAH has been reported the second cause of death after interstitial lung disease (Elhai *et al.*, Ann. Rheum. Dis 2017) and is associated with a poor prognosis (Lefevre *et al.*, Arthritis Rheum. 2013). The gold standard for the treatment of pulmonary hypertension in SSc consists of vasoactive drugs like endothelin1-receptor or PDE5 inhibitors (Humbert *et al.*, Eur. Respir J. 2022), and not anti-inflammatory nor anti-fibrotic drugs. In the current study, we focused on the well-established pro-inflammatory and pro-fibrotic actions of Gal-3, also known to reflect the main features of the disease. To this aim, we chose the HOCl model, which is known to recapitulate the inflammatory and fibrotic characteristics of SSc, and where Gal-3 could be envisaged as a possible strategy, much more than vasculopathy features. This model is also not known to be associated with pulmonary hypertension. For those reasons, PAH was not measured in our experiments.

Nevertheless, to investigate your question, we performed an additional RT-qPCR analysis looking at the expression level of well-known endothelial markers in the lungs of control and HOCl mice. In good coherence with the fact that the HOCl model is not/poorly associated with the vasculopathy events observed in SSc, no difference between control and HOCl mice could be observed for any of the markers examined.

3. Generally, the assessment of the lung etiology is not always sufficiently detailed. More precisely, only collagen accumulation and BAL (bronchoalveolar lavage) are examined in this paper. While BAL mainly represents alveolar macrophage, there are many other indicators for evaluating pulmonary fibrosis. For example, it is also important to analyze interstitial macrophage as well. For this purpose, the authors could analyze the lung cell-dispersed specimen analyzed by FACS, again.

Thank you for your suggestion that allowed us to enrich the content of our study with the quantification of lung tissue macrophages. While lung cell-dispersed specimen were no more available for FACS analysis, we examined the presence of the macrophage surface marker F4/80 by immunofluorescence. This analysis has been added in the revised manuscript as Supplementary Fig. 11 and commented in the Results section (lines 275-281) and in the discussion (lines 437-438). A 3.3-fold increase was found in HOCl versus control conditions, and D11 mAb, E07 mAb or TD139 inhibited this pathological increase by 43.6%, 88.2% and 100% respectively. While these values reflected a decrease of both lung alveolar and interstitial macrophages, the effect was particularly obvious on interstitial macrophages after treatment with E07 mAb and TD139. The methodology has been added in the methods section (lines 726-735).

Minor Points:

4. In addition to the HOCl mice, there are a number of other types of SSc model mice. I don't know if there are any special circumstances, such as the fact that the endothelium of the blood vessels in this HOCl model may not be damaged so much. Please enrich the discussion.

Animal models of SSc are generally chosen according to the question asked, as SSc is an heterogeneous disease, where drugs can target one or more specific pathways including inflammation, fibrosis and/or vasculopathy (Morin F. *et al.*, *Curr. Pharm. Des.* 2015). As we aimed to assess the possible role of Gal-3 blockade, predicted to target primarily fibrosis and inflammation, we selected the model believed most suited to answer this question. The HOCl model is a well known model recapitulating the inflammation and fibrotic features of the disease (Ledoult E. *et al.* *Sci. Rep.*, 2022 ; Meng M. *et al.* *Front Immunol.* 2019, Artlett CM, *Open Access Rheumatol.* 2019) much better than its vasculopathy features. This point has been added in the discussion (lines 424-426) together with 2 additional references in the revised manuscript (48, 49).

5. First paragraph: The manner of describing the transcriptome analysis was somewhat confusing to me. Strictly speaking, although the description indicates the “interactome” analysis”, the data is on the transcriptome and the authors do not analyze any protein-protein/lectin interactions directly. The descriptions in the Method section is rather precise.

Thank you for your comment. We fully acknowledge. To avoid any confusion, the term ‘interactome’ has been removed and replaced by a more adequate formulation, each time it was irrelevant.

6. Is not there any possibility that further inspection of the collected data and the bioinformatics framework would lead to identification of yet novel molecular interaction of novel aspects of transcriptional network. related to Gal-3?

All in-depth analysis of the collected data and the bioinformatics analysis has been shared in the present manuscript and are exemplified in figure 7. Pathway analyses allowed to identify key entities linked to modulation of the transcriptional network related to Gal-3 in SSc patients of the PRECISESADS cohort: IFN- λ 1, IFN- α 2, IFN- γ , IRF-1, -3 and -5, DDX58, MAVS, EIF2AK2, ACKR2, PNPT1, STAT1 and FOXO3.

7. According to Table1, most of the participants (98.6%) involved in the current cohort are Caucasian/White. I wonder if there is any difference between races or other ethnic backgrounds? Is it possible to analyze the profiles by using some public RNA-seq datasets of SSc patients, if any?

Thank you for your comment. Indeed, the PRECISESADS SSc cohort is mainly represented by caucasian/white patients. Following your question, we identified a Geodataset (accession number GSE179153), reporting the global gene expression of peripheral blood cell samples collected from a subset of 49 systemic sclerosis patients enrolled in the GENISOS cohort, as well as unaffected controls. This cohort includes more ethnic groups than the PRECISESADS cohort: 14.3% are african american, 18.4% are hispanic, 2.0% are asian and 65.3% are caucasian. We analyzed the expression of Gal-3

interactants in this cohort and did not find any specific pattern associated with ethnicity, as shown in the heatmap below, where no clustering related to this parameter was found.

8. Is there any other sign of side effects in using mAb for Gal-3 blockade in the mouse model?

In life observations were recorded daily and body weight variation was measured every two days starting from day minus 1 to day 42 in each group of the HOCl mouse study. No change in body weight was observed in the HOCl control group compared to the vehicle control group, neither in mice treated with D11, E07 or TD139 compared to the HOCl control group. No abnormal behavioural sign was observed in the anti Gal-3 treated groups. There was no mortality in any of the anti Gal-3 treated groups. In an independent study conducted for 9 days in rats, subcutaneous administration of D11 or E07 up to 200 mg/kg, did not induce any histological evidence of systemic toxicity at up to 200 mg/kg (microscopic examination of Heart, Ileum, Kidneys, Liver, Lungs, and Spleen), and there was no abnormal macroscopic finding. There was no mortality in any of the anti Gal-3 treated groups.

REVIEWER 3: Expert in galectins

The authors found that the mRNA levels of interactants of galectin-3 (Gal-3) correlated with systemic sclerosis (SSc) disease severity, pulmonary and cardiac malfunctions, neutrophilia and lymphopenia. Moreover, by phage display screening, they identified 66 scFv that bound to recombinant human Gal-3 and developed two mAb composed of human IgG1 Fc. They showed that they also bound to mouse Gal-3. They demonstrated that these mAbs decreased the pathological features of SSc in a mouse model induced by HOCl.

Major comments:

1. The development of biological therapeutic agents targeting Gal-3 is remarkable, since this protein has been implicated in a large number of diseases; several companies have been working for many years on development of galectin-3 inhibitors as drugs, but so far, only one has been shown to be efficacious (and FDA-approved) and it is a small molecular weight compound. The development of mAb as a Gal-3-targeting drug is challenging, as this protein functions both extracellularly and intracellularly and, for a give disease, so far, the existing information does not allow a quantitative differentiation of its contributions in these two compartments. If the protein contributes to the disease mainly through its intracellular functions, then theoretically, mAbs would not work, as they are not able to access the intracellular space. The data on the therapeutic effects of the two mAbs in a mouse model of SSc reported in this manuscript appear convincing. However, there are a number of issues that need to be addressed.

The authors need to clearly address whether these mAbs function by inhibiting the binding of extracellular Gal-3 to cells, thus inhibiting the cellular responses induced by the protein. On the other hand, these mAbs can potentially crosslink Gal-3 on the cell surfaces, resulting in suppressing of cellular responses (such as by crosslinking inhibitory receptors). To address these possibilities, the authors need to examine the amount of cell surface Gal-3 in various tissues, before and after the treatment with the mAbs. Another possible mechanism of action of these mAbs is their formation of immune complexes, which can have immunosuppressive functions. In this case, the suppressive functions of Gal-3-neutralizing mAbs can only be viewed as the properties of the immune complexes and do not reflect the functions of endogenous Gal-3.

Thank you for your comment. As you mentioned above, the existing information does not allow a quantitative differentiation of the contributions of Gal-3 in the extracellular and intracellular compartments. The complexity of its mode of action, still to be considered empirical to date, lies in the fact that it interacts with a plethora of proteins present intracellularly, at the cell membrane, in the extracellular matrix and in biological fluids. While looking at the impact of our mAbs on its membrane-associated partners specifically is difficult from a technical point of view, we examined the distribution of the 69 Gal-3 interactants whose expression allowed to discriminate patient clusters from each other represented on the heatmap in Fig. 1b. We did not find any specific pattern associated with location, as shown in the heatmap below, where no clustering related to this parameter was found. While this is an indirect answer to your challenging question, it indicates that the expression of disease-associated Gal-3 interactants was not associated with one or another compartment in particular. Concerning your point on the possible formation of immune complexes, we emphasize that our mAbs are Fc-silent (LALA mutated, see answer to reviewer 1). Furthermore, the formation of immune complexes would trigger immune responses and inflammation, at the opposite of the results observed in the present study.

In addition, one critical set of experiments to add involves the use of Gal-3 KO mice. This reviewer assumes this strain of mice would have a less severe phenotype, which reflects the function of endogenous Gal-3 in the pathogenesis of SSc. The author could compare the differences in the responses to HOCl between WT and KO mice to those between mice treated with mAbs and control

antibodies. Moreover, the demonstration that the mAbs do not have any effect on Gal-3 KO mice would be a strong support that they work by targeting Gal-3.

You raise here an interesting question, but we do not have access to Gal-3 KO mice. However, one point to be considered is our objective to generate monoclonal antibodies to block extracellular Gal-3 only, rather than multi-compartmental Gal-3. Gal-3 KO mice would recapitulate the phenotype of mice devoid of Gal-3 in the intracellular compartment as well, which is believed poorly if ever associated to fibrosis and inflammation, thus out of our stated objective. The results obtained with the E07 mAb in the present study, capable of reversing the expression of 87.2% of the HOCl-modulated genes demonstrated that the pathology is mostly driven by extracellular Gal-3 interactants.

Regarding your second point, you are right in saying that the KO model would help in demonstrating that our Gal-3 mAbs are fully Gal-3 selective. Your comment is also related to reviewer's 1 comment. To address this question, we have performed an additional set of experiments to measure the potential binding of D11 and E07 mAbs to other galectin family members. These new results are commented in the results section and in the discussion. The moderate binding observed for Gal-14 is not an issue. This protein is located intracellularly and its expression is restricted to placenta (references 45 and 46 added in the discussion), implying that this binding is unlikely to occur in biological systems or in a clinical setting, and that it does not represent an interpretation bias in the present study. A protein blast analysis also showed that no significant identity was found between the CRD of Gal-3 or full-length Gal-3 with non-galectin proteins, emphasizing the low probability of such off target recognition.

2. The authors missed one critical control and that is the use of control hIgG1. An ideal one would be that binds a serum component in mice that is known not to have a pathological role in SSc. This is important, as the authors used chimeric mAbs with a human IgG1 Fc portion and these has the potential to induce anti-human IgG antibodies in mice, after multiple subcutaneous injections, which can form immune complexes with the mAbs. In this regard, the authors should determine whether the mAbs induced anti-human IgG1 antibodies in their experiments.

Thank you for your comment that allows to enrich the manuscript with some results answering your question. ADAs were measured using a generic method (GYROS) as exemplified below.

The revised manuscript has been impleted with the following sections:

- Methods: lines 743-751

Anti-drug antibodies (ADA) were measured with a generic method implemented on a Gyrolab xPlore automated immunoassay system. Mouse plasmas were coincubated with excess concentration (10 µg/mL) of D11 or E07 mAbs. Then, the ADA-mAb complexes were captured by biotinylated Goat anti-human IgG (Southern Biotech 2014-08) coated on the streptavidin beads within the compact disc. Finally, 10 nM of Alexa Fluor 647 goat anti-mouse IgG (H+L) cross-adsorbed secondary antibody (Invitrogen A-21235) was used as detection agent. Values were expressed as fluorescence units with a significant cut-off threshold in each experimental run determined as the mean of 3 standard deviations calculated from the mean of signals obtained from 15 control untreated mice.

- Results: lines 265-268

No anti-drug antibodies (ADA) could be detected in plasmas collected from mice treated with E07, and low ADA levels were measured in 5 out of 12 mice treated with D11, without any correlation with mAb plasma concentrations (Supplementary Fig. 9).

- Supplementary Figure 9:

3. The authors showed that in their mouse model, the plasma levels of Gal-3 were significantly increased after pathological induction with HOCl, but fully reversed after D11, E07 and TD139 treatments. This needs to be clarified. One does not expect the total Gal-3 levels would be reduced after the treatment with these mAbs. In fact, the total amounts should become higher, because the immune complexes are expected to be more stable in circulation. One would not expect the Gal-3 levels to be reduced by TD-139 either. The authors need to measure the total Gal-3 amount in the serum.

You are right, this observation is counter-intuitive. Neutralizing Gal-3 with a therapeutic mAb, or a small compound in the case of TD139, should stabilize Gal-3 and thus increase its levels in biological fluids (a reflect of target occupancy). One hypothesis here is that anti-Gal-3 mAbs used as detection reagents in the commercial ELISA kit are hampered in their capacity to bind to / and recognize Gal-3 when it is already bound to high affinity therapeutic mAbs. In other words, the Gal-3 ELISA detects free Gal-3 better than bound Gal-3, that may also be a reflect of target occupancy.

Similar observation has been made with the Gal-3 inhibitor TD139 both in human (Hirani N, *et al.* Eur Respir J. 2021). Another possible explanation is that Gal-3 levels are decreased at a certain stage of disease alleviation, as a reflect disease improvement (in other words, Gal-3 not only acts as a therapeutic target, but also as a biomarker of fibrosis and inflammation).

4. The authors generated a list of known Gal-3 partners through bioinformatic queries within Ingenuity Pathway Analysis (IPA) and GPS-Prot databases. Gal-3 is known to bind to a large number of glycoproteins through lectin-carbohydrate interaction and also intracellular mediators through protein-protein interaction. Thus, the authors need to separate the interactants into different categories and also comment on the issue of the lack of evidence for the role of many of these interactants in the function of Gal-3.

Thank you for this suggestion. We have now added the location of all Gal-3 interactants listed by alphabetical order in supplementary Table 1 of the revised manuscript (columns A and B). An additional sorting by location is proposed in supplementary Table 1, columns D and E. As mentioned in our answer to your comment 1 above, we did not find any specific pattern associated with location, as shown in the heatmap, where no clustering related to this parameter was found.

Concerning your second point on the lack of evidence for the role of many of these interactants in the function of Gal-3, we emphasize that we looked at the differential gene expression levels of these interactants in patients clusters (Fig 1b), where 69 were found discriminating, and in patients *versus* healthy controls (Figure 3 and supplementary Table 2), where 103 out of 248 interactants were found differentially expressed. We believe that we do not overinterpret our findings by referring to a gene signature or fingerprint rather than to a protein level or function. We would also like to emphasize that stringent criteria were fixed to retain differentially expressed genes with false discovery rate (FDR) adjusted p-values ≤ 0.05 and an absolute fold-change ($|FC| \geq 1.3$), implying that genes for which differential expression values are moderately below these cut-off values may also be of biological significance. Related to this last point, volcano plots newly added in Figure 3b of the revised manuscript help understanding this fact: a number of genes discarded with this level of stringency were nevertheless found very close to the minus 1.3 or plus 1.3 FC threshold, and most genes out of the FC threshold had FDR values largely below 0.05.

5. In addition, many of the glycoproteins that Gal-3 has been reported to interact with might not be relevant to the pathological processes, as it might not encounter them *in vivo*. Thus, the rationale of relating galectin-3 interactants levels to the pathological processes in SSc should be explained,

including whether interaction between galectin-3 and their partners listed is critical for development of SSc. In this regard, the authors could analyze Gal-3 WT and KO mice to establish the relevance of these interactants.

As we mentioned above, we do not have access to Gal-3 KO mice. You are right in saying that all interactants may not all be involved in the pathological processes in SSc. This is the reason why we analyzed key pathways and entities involved in these processes, highlighted in figures 6 and 7.

Other comments:

1. Figure 1: The logic of clustering patients into three subgroups should be explained (e.g., why choose three clusters and condition of supervised selection and clustering) and the code for the clustering should be deposited to github.

The number of clusters was determined as the best consensus between three unsupervised clusterings. This has been added in the Methods section of the revised manuscript (lines 504-506).

The github code has been deposited at https://github.com/psBiostat/GAL3_PAPER.git and is also mentioned in the Methods section.

How were the Gal-3up and Gal-3down gene lists determined? Was that based on their expression levels in the C3 groups?

Thank you for allowing to clarify. Yes, this was based on their expression levels in C3. This precision has been given in the revised manuscript (line 520).

The most robust Gal-3 fingerprint associated with SSc patient clusters was defined by 69 Gal-3 interactants, 48 of which were upregulated and 21 downregulated as compared to C3. 'Gal-3up' and 'Gal-3down' scores were calculated as follows:

Score 'Gal-3up' = Median $_{j=1, \dots, Gup}$ (Y $_{ij}$ – median $_{k=1, \dots, Nctrl}$ (Ctrl $_{kj}$))

Score 'Gal-3down' = Median $_{j=1, \dots, Gdown}$ (Y $_{ij}$ – median $_{k=1, \dots, Nctrl}$ (Ctrl $_{kj}$))

2. Figure 2: Are the Gal-3up scores in the subset of patients a mean of all Gal-3up genes in individual patients? How were the immune cell population counts defined? What genes were used to calculate the "counts"? The code used should be deposited to github.

The Gal3up scores were calculated based on the median of Gal3up genes in individual patients and centered by control levels, as described in the formula mentioned above. Details about immune cell population counts were described in the publication from Barturen G. *et al.*, referred in the present manuscript as reference 55. The Methods section has been amended (line 491): 'More technical details about the sample and data collection (including inclusion / exclusion criteria and immunophenotyping) have been published previously⁵⁵.

Regarding your comment on genes used to calculate the counts, please note that data represented in Figure 2 are only referring to flow cytometry counts. No genes were used. In these conditions there is no relevance for a github code because data were generated from flow cytometry data.

3. Figure 3: The author may show a volcano plot for the differential expression gene profile by comparing the two groups as mentioned in the text.

Thank you for this helpful suggestion. Volcano plots related to the differential expression of Gal-3 interactants between HV and C3, C1 and C2, respectively, have been added in Figure 3 of the revised manuscript. The legend of figure 3 has been amended accordingly.

4. The authors need to provide explanations on why the two mAbs have different effects.

D11 and E07 mAbs had similar qualitative effects on several readouts, including reduction in plasma Gal-3 levels, skin thickness, skin and lung collagen content, IL-5 and IL-6 levels. D11 also reversed the

expression of genes induced in HOCl conditions in a qualitative manner similar to that observed with E07 but with lower potency (Fig. 6b). There was also a trend, similar to the effect observed with E07, on the content of alveolar macrophages and other myeloid cells (Fig. 5), and F4/80-positive macrophages in lung tissue (Supplementary Fig. 11). Both mAbs have quite similar affinities for Gal-3, but one possibility remains that a subtle different positioning within the Gal-3 CRD could result in different qualitative or quantitative binding of some Gal-3 interactants. Another possible explanation is that more heterogeneity was observed at the gene expression level after D11 versus E07 treatment (see PLSDA analysis, Fig 6a.), resulting in lower statistical power.

Reviewer #4 (expert in systemic sclerosis): Comments for authors

In this manuscript, the authors performed cluster analysis using RNA sequencing using whole blood, and Gal-3 and its related factors were related to clinical parameters in SSc (systemic sclerosis) patients. Then, they discovered novel Gal3-neutralizing antibodies and used them for the HOCl-induced SSc mouse model to evaluate the therapeutic potential of their Gal-3-neutralizing antibodies. Although these contents are interesting, there are some concerns about the interpretation of the results and clinical aspects as listed below.

[Major points]

1. Whole blood samples were analyzed in this study. They highlight that Gal-3 expression is strongly correlated with neutrophilia and lymphopenia. However, this might require careful interpretation. Given that whole blood is rich in neutrophils and lymphocytes, changes in their relative proportions must greatly affect their RNAseq results, especially in case Gal-3 is expressed in neutrophils or lymphocytes. In fact, there seems to be literature indicating that Gal-3 are expressed and function in neutrophils. For these reasons, I think they should be a little more cautious when discussing the correlation with neutrophils/lymphocytes from a clinical point of view.

Thank you for your comment. We did not correlate neutrophilia and lymphopenia with the expression of Gal-3, but with that of its fingerprint, also showing correlation of this fingerprint with the neutrophil to lymphocyte ratio taken as a marker of inflammation. Gal-3 is ubiquitous and similarly expressed both in lymphocytes and neutrophils as shown in the Protein Atlas datasets.

In the case your question was about the Gal-3 fingerprint rather than Gal-3 itself, we emphasize that no single cell RNAseq data were generated in the frame of the PRECISESADS consortium. Thus, the relative expression of 'Gal-3down' genes and 'Gal-3up' genes between neutrophils and lymphocytes could not be assessed in healthy and patient clusters from this SSc cohort. We believe that our statement does not overstate interpretation of this correlation, mentioning that '*this Gal-3 fingerprint could also serve as a stratification biomarker to discriminate patients based on disease features and/or inflammatory status in a targeted treatment approach*'.

2. There are further concerns related to blood cells (e.g. data in Fig 2D etc.). They indicated that neutrophilia and lymphopenia are associated with cluster 3 and associated with disease activity. Cytopenia in patients with SLE, another collagen disease, is an important activity indicator. However, regarding SSc, there must not be a strong consensus that neutrophilia and lymphopenia are useful in the evaluation and monitoring of SSc patients in clinical practice. This is one of the reasons why I am concerned about the previous point regarding whole blood sample analysis and I feel uncomfortable with the description of neutrophilia and lymphopenia.

We thank the reviewer for this comment. In a recently published study (Chikhouné *et al.*, J. Clin. Med. 2022), we showed that total white blood cells and neutrophilia were associated with a more severe SSc phenotype. A recent study also showed that higher blood neutrophil count and neutrophils to lymphocytes ratio were predictive of more severe disease course and increased mortality in SSc (Wareing *et al.*, Arthritis Care Res 2022).

3. As the authors have already noted in the introduction, Gal-3 is well-documented to be important in many fibrotic diseases and systemic sclerosis. Regarding the potential of Gal-3 as a therapeutic target for SSc, as they indicated in the title, there are some papers showing that bleomycin-induced lung fibrosis was reduced in Gal-3 deficient mice or by the treatment with TD139, an inhaled Gal-3 inhibitor. Furthermore, as described in the introduction, a phase I/IIa clinical trial was also conducted with TD139 in IPF patients. I agree that there are new points in terms of new analysis methods, the HOCl model,

and the discovery of novel antibodies. However, in light of those previous reports, focusing on Gal-3 and SSc in this work may not be so novel. In this sense, I feel unsatisfactory with whether the title properly expresses the content including the novelty.

You are right in saying that Gal-3 inhibitors were active in the mouse IPF model of bleomycin-induced lung fibrosis and that TD139 is currently evaluated in a clinical trial in IPF patients. These studies are mentioned in the discussion (lines 389-392) and referred as references 19 and 20, and 21, respectively. Our study is the first one investigating the role of Gal-3 and its interactants in a cohort of SSc patients, an autoimmune disease different from IPF, and in a relevant preclinical SSc model, through a multidimensional and translational approach. We believe that the title is more appropriate if sticking to systemic sclerosis. In this way, we do not over-interpret results of the present study.

4. Their functional molecular cluster analysis revealed that C3 contained a larger proportion of diffuse cutaneous SSc patients (dcSSc) and displayed a higher inflammation profile than other clusters. In addition, regarding clinical manifestations, C3 patients showed more pulmonary fibrosis, arrhythmias, and higher disease activity compared with other clusters. Moreover, they investigated the importance of the Gal-3 fingerprint by examining the relationship between the Gal-3 up and Gal-3 down scores with features of SSc. As a result, it became clear that the expression level of a high number of Gal-3 interactants was strongly associated with impaired vital organ function in SSc patients and with the level of disease severity. As the authors may know, the antibody profile of systemic sclerosis is known to correlate well with clinical characteristics, and it is expected that the pathology should differ for each antibody. For example, complications of ILD (SSc-ILD) are frequently seen in SSc patients with anti-Sci70 antibodies. On the other hand, anti-centromere antibody-positive patients are frequently classified as lcSSc, and they sometimes have complications of pulmonary hypertension with relatively high frequency. Considering these facts, it is difficult to determine whether the correlation between severity and high activity in the C3 cluster is purely due to complications and fibrosis, or due to differences in pathology depending on antibody profiles. In other words, for example, if the observed characteristics of C3 cluster are based on the pathological features of anti-Sci70 antibody-positive, it is possible that the patients without anti-Sci70 antibody but classified to C3 cluster do not correlate well with activity or complications. In fact, looking at Table 2, about 70% of C3 patients are anti-Sci70-positive, so they should be careful about the effects of each antibody. If I can propose the investigation methods to clarify these possibilities, the results obtained in Figures 1 and 2 can be analyzed again in a similar way by separating each category of patients (C1-C3) into anti-Sci70-positive patients and anti-centromere-positive patients. If a similar trend is observed in these analyses, the results they have already shown should be independent of antibody differences.

Thank you for this comment. As you mention, anti-Sci70 positive antibodies were more frequent in patients from cluster 3, more represented by diffuse cutaneous SSc and showing higher disease complications together with a higher Gal-3 score compared to other clusters. Conversely, patients from clusters 1 and 2, more associated with lcSSc and intermediate and low Gal-3 scores, respectively, had higher proportion of anti-centromere antibody positivity. To answer your question, we examined the Gal-3^{up} and Gal-3^{down} scores in each cluster according to the rate of positivity of each of these antibodies. The distribution of each cluster was independent of the rate of positivity, see below. To be read as follows, with the example of C3 up scores: highest Gal-3 score, more patients with anti SCL70 than anti Cent B, but no association between the rate of positivity and the Gal-3 score.

5. There is one more point to make regarding auto-antibodies. As the authors know, representative auto-antibodies specific to SSc include anti-Scl70, centromere, and RNA polymerase III antibodies. In Table 2, there is no description of RNA polymerase III antibodies. Among the patients enrolled in this project, were any patients positive for this antibody? This study seems to include many Caucasians. The proportion of patients with RNA polymerase III antibodies should not be small in Caucasians, and this antibody is well-known to be a risk factor for rapid skin sclerosis and renal crisis. In this study, they argue that Gal-3 and its interacting factors are deeply involved in disease activity and inflammation. Therefore, I think that this should be examined and discussed.

Unfortunately, RNA polymerase III antibodies were not part of the antibody panel measured in the PRECISESADS cohort.

6. There are more questions in terms of neutrophils and lymphocytes. As they may know, there are various treatments for SSc, such as steroids and other immunosuppressants. In Table 1, they described that about 25% of each of the patients in this study were treated with steroids and immunosuppressants. Patients with high disease activity may be taking these therapies. Steroids often lead to neutropenia. And, immunosuppressants sometimes lead to lymphopenia. Considering these, it is likely that the therapeutic agent may have influenced the features seen in the C3 cluster such as neutrophilia and lymphopenia. Therefore, they may not be able to rule out the possibility that these correlations are not intrinsic and may even reflect the effects of therapeutic agents.

Thank you for your comment.

In the inclusion criteria, patients treated with high doses of immuno-suppressants for the 3 months prior to recruitment, or cyclophosphamide or belimumab in the past 6 months were not eligible for the study. Reference 55 in the section 'Methods, patient population' provides all details regarding the selection of patients for this study.

More precisely, exclusion criteria were :

- Patients with stable doses of steroids >15 mg/day for the last 3 months or with IV corticosteroids in the last 3 months
- Patients under immunosuppressant treatment in the last 3 months prior to recruitment and patients with combined therapy using two or more immunosuppressants
 - Methotrexate \geq 25 mg/week
 - Azathioprine \geq 2.5 mg/kg/day
 - Cyclosporine A > 3 mg/kg/day
 - Mycophenolate Mofetil > 2 g/day
- Treatment with cyclophosphamide (any dose or route of administration) or belimumab in the past 6 months
- Patients on depletative therapy such as rituximab in the last year

[Minor points]

7. They used the HOCl mouse model for the Gal3-neutralizing antibody, but there are other models such as the bleomycin model. Why did you use the HOCl model this time? Have you examined the BLM model in the same way?

Animal models of SSc are generally chosen according to the question asked, as SSc is a heterogeneous disease, where drugs can target one or more specific pathways including inflammation, fibrosis and/or vasculopathy (Morin *et al.*, *Curr. Pharm. Des.* 2015). As we aimed to assess the possible role of Gal-3 blockade, which is reported to target fibrosis and inflammation, we chose a model more likely to mirror the human pathology. The HOCl model is a well known model recapitulating inflammatory and fibrotic components of the disease (Servettaz *et al.*, *J Immunol* 2009; Ledoult *et al.*, *Sci. Rep.* 2022) more than the vasculopathy part. This is now mentioned in the discussion, lines 424-426. The bleomycin model is depicted to be more representative of idiopathic pulmonary fibrosis than SSc (Tashiro *et al.*, *Front. Med.* 2017; Moore *et al.*, *Am. J. Physiol.* 2008), and care should be taken in extrapolation of drugs successfully tested in this model due to partial reversibility of bleomycin-induced fibrosis over time (Moeller *et al.*, *Int. J. Biochem. Cell. Biol.* 2008). In our study, pathological pathways in the mouse HOCl model showed outstanding similarity with human SSc (Fig. 8), demonstrating the relevance of this model for our approach.

8. Data for IL-5 and IL-6 are shown in Figure 5, etc. If so, I think it is necessary to mention in the explanation or introduction how these cytokines are important in the pathology of SSc. In fact, in some countries, biologics targeting IL-6 have already been clinically applied to SSc-ILD.

You are right, IL-6 plays a major role in the pathophysiology of SSc, leading to tocilizumab, a mAb directed against the IL-6 receptor, to be approved by the FDA for the treatment of lung fibrosis in SSc. This was mentioned at the end of discussion of the original manuscript together with reference 50 (lines 466-468). It has been reinforced by a comment on lines 440-442. The role of IL-5 in SSc is more hypothetical, although an earlier study reported that IL-5 expression levels were increased in BAL cells from SSc patients, in association with significant decline in forced vital capacity over time (Atamas *et al.*, Arthritis Rheum. 1999). Since then, IL-5 has been poorly studied and was not reported as an important actor in SSc. This is now mentioned in the revised manuscript (lines 442-444) together with the associated reference 51. The role of IL-5 has been more studied in the context of eosinophilia diseases such as severe eosinophilic asthma, eosinophilic granulomatosis with polyangiitis or hypereosinophilic syndrome.

9. In Fig. 5, cytokines are reduced by E07. Is this related to changes in lymphocyte counts? And/OR does E07 somehow directly affect cytokine production in Gal3-expressing cells? This paper has many descriptive parts. Therefore, I can advise that the quality of this paper would be improved if it is possible to add an examination using *in vitro* experiments to address these questions.

IL-5 and IL-6 levels were also reduced by D11, and IL-5 levels were reduced by TD139. For D11 and TD139, there was no observed change in lymphocytes, making very unlikely that these 2 outcomes are related. There was also no correlation between IL-5 or IL-6 levels and lymphocytes.

Regarding your second point, we emphasize that *in vitro* assays are not relevant to examine the complexity of the mode of action of Gal-3 in biological systems, because its multiple interactants are only partly represented in isolated cells. Thus, the states of balance/imbalance of biological functions depends on the global Gal-3 interactome, its distribution in different compartments, and the communication between cells and tissues.

10. They showed that E07 treatment significantly decreased the levels of cytotoxic CD8+ T cells and NK cells. This is interesting. As the authors may know, cytotoxic CD8+ T cells and NK cells are very important cells from the point of view of tumor immunity. This means that if E07 can be used in clinical practice in the future, it may exacerbate the progression of cancer in cancer-bearing patients. As mentioned above, there are reports that malignant tumors occur at a high rate in anti-RNA polymerase III antibody-positive patients. I think it would be better to discuss this concern.

Thank you for your comment. Indeed, several studies have shown that anti-RNA-PolIII antibodies were associated with cancer in SSc. Genetic alterations of the POLR3A gene were identified in the tumors of SSc patients with cancer and anti-RNA-PolIII antibodies. This mutated auto-antigen in be the *primum movens* triggering autoimmunity in these patients, inducing cellular and humoral responses, with the production of anti-RNA-PolIII autoantibodies suggesting that anti-RNA-PolIII antibodies were not responsible for the occurrence of cancer but were a consequence of it. It explains the recommendations that an associated cancer must be looked for in the case of diffuse cutaneous SSc with anti-RNA polymerase III antibodies (Hachulla E. *et al.*, Orphanet J Rare Dis. 2021). In patients with paraneoplastic SSc, there are no specific recommendation other than treating the underlying cancer and managing the symptoms of SSc with drugs allowed by the oncologist. Concerning the risk of cancer development in non-paraneoplastic SSc patients, international recommendations do not preclude the use of strong immunosuppressants such as cyclophosphamide, rituximab or even autologous stem cell transplantation (Kowal-Bielecka O. *et al.*, Ann Rheum Dis. 2017) due to a theoretical increase risk of cancer. As for these medications, the theoretical risk of cancer development with anti Gal-3 treatments will be part of the long-term safety development plan and carefully monitored at early clinical stages and following post-market surveillance.

11. In terms of the Methods section, the mouse model description "Mouse model of HOCl-induced SSc" seems too long. This includes many experiments such as FCM and BAL. I think that the description of the mouse model and the individual analysis methods should be described separately.

Thank you for your suggestion. This section has been split accordingly to improve clarity and readability.

REVIEWERS' COMMENTS

Reviewer #1 (expert in antibody development, phage display):

This reviewer was no longer available for review and was therefore replaced by Reviewer #5.

Reviewer #2 (expert in transcriptomics and transcriptional networks):

First of all, I appreciate the substantial work which has been made by the authors for this revision. Thanks to the careful extensive analyses and the deepened discussion, the manuscript has been significantly improved. Almost all of the concerns which I have raised in the previous round of the review has been completely addressed. I also understand the remaining parts, if any, should be left for their future work. In fact, I foresee substantial additional work still waiting before the achievements as presented here should be eventually brought to the bed side of patients. I sincerely hope the authors even accelerate their efforts towards that goal.

Reviewer #3 (expert in galectins):

Major comments:

Although the authors provided additional data and responded to many of the reviewers' comments, in this reviewer's view, a great deal of issues remain. The main issue is how galectin-3 contributes to SSc is not known and the mechanism of action of galectin-3-blocking antibodies can only be speculated. It is important to point out galectin-3 is not a cytokine and it is known to function intracellularly. A large number of extracellular functions have been reported for this protein, but they were demonstrated with the use of exogenously added protein and may not be relevant to how endogenous galectin-3 works (although one certainly cannot exclude the possibility that they are relevant).

The authors described a large number of interactants. However, these include both cell surface proteins and intracellular proteins that galectin-3 is known to bind to. The galectin-3 blocking monoclonal antibodies obviously work by binding to extracellular galectin-3, and not intracellular one. Although the authors responded to this comment and provided additional data, it is very hard to envision how the

antibody would affect many of the intracellular galectin-3 binding partners, although additional investigations might discover heretofore unknown mechanisms (of how galectin-3 works).

In this regard, it is to be noted, galectin-3 probably binds to all the different cell types and it is known that it binds to many cell surface glycoproteins on some of the cell types studied. Many of these “interactants” might not be relevant to the function of galectin-3.

The authors appear to relate the mechanism of action of their antibodies to blocking galectin-3 in circulation, but did not provide any data on how these circulating galectin-3 contributes to the pathogenesis of SSc, other than citing references on how exogenously added galectin-3 cause cellular responses, which has an issue as mentioned above. Galectin-3 likely exists in complex with many glycoconjugates (e.g., Cedefur et al. Different affinity of galectins for human serum glycoproteins: galectin-3 binds many protease inhibitors and acute phase proteins *Glycobiology* 18:384-94, 2008). In fact, these complexes might be relevant to how galectin-3 is pathogenic in SSc, although there is no information in the literature to support this notion.

This reviewer suggested in the previous review, “the authors need to examine the amount of cell surface Gal-3 in various tissues, before and after the treatment with the mAbs”. The authors did not address this point. This reviewer understands this is a very challenging problem to address, but the issue is we really do not know how and whether extracellular galectin-3 contributes to the pathogenesis of SSc (or any other diseases), and thus without additional work addressing the issue, one really does not know these mAbs work.

This reviewer also suggested that the authors look at galectin-3 knockout mice. The authors responded by stating, “we do not have access to Gal-3 KO mice”. However, this strain is readily available from Jackson Laboratories and its availability is listed on the webpage of Consortium for Functional Glycomics, and has been used in numerous publications.

The authors found their monoclonal antibodies cross react with galectin-14, but commented (page 11, line 404), “As this protein is located intracellularly and its expression is restricted to placenta, this binding is unlikely to occur in biological systems or in a clinical setting”. In this reviewer’s view, galectin-14, like other galectins, including galectin-3, while being located intracellularly, as the authors correctly stated, must be present extracellular as well (although to this reviewer’s knowledge this has not been addressed in the literature). Also, although galectin-14 has more limited tissue distribution, when it is released into the extracellular space and being present in circulation, it will probably bind to many cells and tissues, like galectin-3 does. Thus, the statement “this binding is unlikely to occur in biological systems or in a clinical setting” might not be correct. As a matter of fact, one might make the same statement on galectin-3, on the basis of the comments this reviewer made above.

Overall, the development of mAbs that target galectin-3 and are shown to be effective in targeting a systemic disease, in an animal model, is remarkable, especially knowing the complexity of galectin biology. As this reviewer pointed out above, galectin-3 is not a cytokine and the existing information suggests that “neutralizing antibodies” would not be effective in treatment of diseases in which this protein might play role. This might be the reason, so far there has been no biologics that are developed for treatment of diseases by targeting galectin-3, despite the fact this protein has been implicated in the pathogenesis of a number of diseases. One of the reasons might be that galectin-3 functions mainly intracellularly. In this regard, for example, a recent paper by Nguyen et al. (Galectin-3 deficiency ameliorates fibrosis and remodeling in dilated cardiomyopathy mice with enhanced Mst1 signaling *Am J Physiol-Heart Circ Physiol* 316: 316: H45–H60, 2019) concluded, “Gal-3 is largely localized within cardiomyocytes rather than the ECM. It remains unknown whether currently available Gal-3 inhibitors like MCP are able to pass through cellular membrane, which would be essential for inhibiting intracellularly localized Gal-3”. It is to be noted also, the only currently FDA-approved drug targeting galectin-3 (TD-139, as the authors cited, also known as GB0139) is a small molecule that is known to have access to intracellular space. This does not mean the mAbs the authors are reporting are not be expected to be developed into therapeutic antibodies. To the contrary, this reagent might be very useful to dissect the mechanism of how galectin-3 functions (and is thus worthy of being reported). However, the authors need to be extra critical and thoughtful in analyzing their data, especially with regard to the interactants, and in articulating the possible mechanism of action of their mAbs, as well as the challenging issues and possible pitfalls.

Other comments:

Page 5, Line 197: “In a second approach, we aimed at identifying which genes representative of the Gal-3 interactome were differentially expressed in SSc patients --. By itself, Gal-3 expression was significantly increased by ----, and a similar increase was found at the Gal-3 protein level ---”. The authors should clarify how the Gal-3 protein level was determined. Was it its concentration in the serum?

Page 8, Line 299, “However, treatment with E07 mAb significantly increased the ratio of alveolar macrophages, while drastically decreasing the ratio of other myeloid cells, suggesting positive outcomes on the resolution of lung damage and inflammation”. The authors might want to elaborate on how the results “suggest positive outcomes on the resolution—”.

This reviewer mentioned that in the previous review, “Another possible mechanism of action of these mAbs is their formation of immune complexes, which can have immunosuppressive functions. In this case, the suppressive functions of Gal-3-neutralizing mAbs can only be viewed as the properties of the immune complexes and do not reflect the functions of endogenous Gal-3”. In the rebuttal, the authors commented “-- the formation of immune complexes would trigger immune responses and

inflammation, at the opposite of the results observed in the present study". However, it is important to point out there exists an inhibitory Fcγ receptor (FcγRIIB), which is the most broadly expressed FcγR and expressed by almost all leukocytes.

Supplemental Figure 7: Galectin-9 is not included. In the rebuttal to one of Reviewer 1's comments, the authors stated, "Recombinant Gal-9 was excluded from the analysis because of low quality control". It is critical to include galectin-9 and this issue needs to be clarified.

Reviewer #4 (expert in systemic sclerosis):

The authors have responded to all of the questions I have raised. While I do not have any major concerns at this time, there are additional comments in the second point on the blood cells (neutrophilia/lymphocytes) in the SSc. I understand what the author explained, but I still do not think that the clinical significance of neutrophils/lymphocytes in SSc has already gained a broad consensus among collagen disease doctors (the papers presented in their comments appear to be very recent). Then, considering that lymphocyte and neutrophil data are presented as one of the main data in this paper, I highly recommend the authors politely explain the clinical significance of white blood cells and neutrophilia in SSc in the text with showing some recently published studies, as the authors wrote in the comments to me (I apologize if this sufficient explanation has already been posted in the manuscript somewhere). These will help the readers to understand the clinical significance of their findings related to blood cells.

Reviewer #5 (expert in antibody discovery and phage display):

Additional Reviewer

The manuscript describes the isolation by phage display of several antibodies to Galectin-3, with two lead antibodies identified that bind to human Galectin-3, and cross-react with mouse, cyno and dog Galectin-3. There is no cross-reactivity to other Galectins other than minimal reactivity to Galectin-14. The authors proceed to describe a very thorough characterisation of the antibodies, both for biophysical characteristics (purity, aggregation, heat stability, binding affinities) and for efficacy in in vivo models.

I have been asked to review the rebuttal of the authors to Reviewer 1's comments, due to the unavailability of Reviewer 1. Like Reviewer 1, I am also an expert in antibody discovery and phage display. Overall, I believe that the authors have sufficiently responded to the reviewer's comments, and should be commended that they have also performed additional experiments in order to meet these comments. Specific feedback on each of the Reviewer's comments and the authors' rebuttals are given below:

Reviewer 1's first question was regarding the intended isotype of the antibodies, in relation to the authors' report that a human IgG1 isotype was used, and they asked if an effector-less isotype should be used instead. The authors have acknowledged that there was an omission in the manuscript and that the antibody isotype was in-fact a LALA variant of the IgG1 isotype which has reduced effector function. Mention of the LALA variant, with references, has been added to the manuscript.

Reviewer 1's second comment was a concern that the SPR experiments captured too much antibody and risked inaccurate results. The authors have rebutted that while the experiment could have been run at lower capture level, to avoid the risk of mass transport limitations, the sensorgrams are still showing curvature and are therefore still valid. The authors correctly state that if there was mass transport there would be linearity in the association phase. The authors have now added the sensorgrams along with the calculated parameters in the supplementary section so that readers can assess the accuracy.

Reviewer 1 then suggested to extend the specificity tests to include more antigens other than Gal-1 and Gal-7, and the authors have responded with additional assays to test Gal-2, Gal-4, Gal-8, Gal-10 and Gal-14. Although they saw some cross-reactivity with Gal-14, they explain why this wouldn't be an issue for therapeutic use due to its expression being limited to the placenta.

Reviewer 1 has asked about a toxicology assessment of using the antibodies in a clinical setting and also asked if there were any side-effects in the mouse model. The authors have only responded to the second question – confirming that there were no side-effects. Although the authors have not responded to the comment about toxicology assessment, I don't believe this is a necessary inclusion in this paper which is describing the early-stage discovery and characterisation of the antibodies. Toxicology assessments could include tissue cross-reactivity studies, and should be considered for future work.

Reviewer 1 has asked that the sequences of the antibodies be included, and the authors have responded that the sequences have been filed in a patent. The patent reference could be included in the manuscript so it can be found easily by readers.

The authors have corrected a typo picked up by Reviewer 1.

REVIEWERS' COMMENTS

Reviewer #1 (expert in antibody development, phage display):

This reviewer was no longer available for review and was therefore replaced by Reviewer #5.

Reviewer #2 (expert in transcriptomics and transcriptional networks):

First of all, I appreciate the substantial work which has been made by the authors for this revision. Thanks to the careful extensive analyses and the deepened discussion, the manuscript has been significantly improved. Almost all of the concerns which I have raised in the previous round of the review has been completely addressed. I also understand the remaining parts, if any, should be left for their future work. In fact, I foresee substantial additional work still waiting before the achievements as presented here should be eventually brought to the bed side of patients. I sincerely hope the authors even accelerate their efforts towards that goal.

Thank you for your encouraging comments and acknowledging all changes made in the revised manuscript.

Reviewer #3 (expert in galectins):

Major comments:

Although the authors provided additional data and responded to many of the reviewers' comments, in this reviewer's view, a great deal of issues remain. The main issue is how galectin-3 contributes to SSc is not known and the mechanism of action of galectin-3-blocking antibodies can only be speculated. It is important to point out galectin-3 is not a cytokine and it is known to function intracellularly. A large number of extracellular functions have been reported for this protein, but they were demonstrated with the use of exogenously added protein and may not be relevant to how endogenous galectin-3 works (although one certainly cannot exclude the possibility that they are relevant).

Thank you for your comment. We fully agree that the mode of action of Gal-3 still remains an enigma, not only for SSc but also for a wide range of other diseases where it has been implicated. As you mention, the difficulty lies in the fact that Gal-3 interacts with multiple partners and has a multi-compartmental location. While in our point of view, the question of the mode of action of Gal-3 is far from being solved (due to this network complexity, and differences between diseases), our study demonstrates a clear association between a Gal-3-based transcriptomic signature and disease severity in SSc patients, and further demonstrates that Gal-3 blockade reduces the severity of skin and lung lesions, among other outcomes, in a murine model of this human disease.

The authors described a large number of interactants. However, these include both cell surface proteins and intracellular proteins that galectin-3 is known to bind to. The galectin-3 blocking monoclonal antibodies obviously work by binding to extracellular galectin-3, and not intracellular one. Although the authors responded to this comment and provided additional data, it is very hard to envision how the antibody would affect many of the intracellular galectin-3 binding partners, although additional investigations might discover heretofore unknown mechanisms (of how galectin-3 works).

In this regard, it is to be noted, galectin-3 probably binds to all the different cell types and it is known that it binds to many cell surface glycoproteins on some of the cell types studied. Many of these "interactants" might not be relevant to the function of galectin-3.

We fully agree with your comment and acknowledge that only part of the Gal-3 interactants may be involved in the pathological processes of SSc. This is the reason why we analyzed key pathways and entities involved in these processes, as highlighted in figures 7 and 8. In our study, pathological pathways in the mouse HOCI model showed outstanding similarity with human SSc (Fig. 8), demonstrating the relevance of this model for our approach.

The authors appear to relate the mechanism of action of their antibodies to blocking galectin-3 in circulation, but did not provide any data on how these circulating galectin-3 contributes to the pathogenesis of SSc, other than citing references on how exogenously added galectin-3 cause cellular responses, which has an issue as mentioned above. Galectin-3 likely exists in complex with many glycoconjugates (e.g., Cedefur et al. Different affinity of galectins for human serum glycoproteins: galectin-3 binds many protease inhibitors and acute phase proteins *Glycobiology* 18:384-94, 2008). In

fact, these complexes might be relevant to how galectin-3 is pathogenic in SSc, although there is no information in the literature to support this notion.

Thank you for your comment. After careful checking, we did not find reference to the fact the mode of action of our mAbs could be preferentially linked to Gal-3 blockade in the circulation. Again, referring to your questions / our answers in the points above, and in the previous round of review, we are cautious and concerned with not over-interpreting our findings, but rather keeping a mandatory part of empiricism inherent to the complexity of the Gal-3 network.

This reviewer suggested in the previous review, “the authors need to examine the amount of cell surface Gal-3 in various tissues, before and after the treatment with the mAbs”. The authors did not address this point. This reviewer understands this is a very challenging problem to address, but the issue is we really do not know how and whether extracellular galectin-3 contributes to the pathogenesis of SSc (or any other diseases), and thus without additional work addressing the issue, one really does not know these mAbs work.

We took good note of your comment. As mentioned, looking at the impact of our mAbs on its membrane-associated partners specifically would be technically demanding, if ever possible, and would require a full dedicated study *per se*. As previously stated, part of the mode of action of our mAbs could be deciphered from pathway analyses shown in Figures 7 and 8, also allowing to gain insights in their promising translational potential.

This reviewer also suggested that the authors look at galectin-3 knockout mice. The authors responded by stating, “we do not have access to Gal-3 KO mice”. However, this strain is readily available from Jackson Laboratories and its availability is listed on the webpage of Consortium for Functional Glycomics, and has been used in numerous publications.

We have taken good note of your comment. Considering that the use of such Gal-3 KO mice would address the question of mAbs selectivity (ie, do mAbs act at least partly by hitting antigens other than Gal-3 ?), ethical guidelines recommend to avoid the use of animals as far as possible for the development of therapeutic candidates. These questions are classically addressed at later development stages by *in vitro* or *in silico* means, as part of the mAbs safety package.

The authors found their monoclonal antibodies cross react with galectin-14, but commented (page 11, line 404), “As this protein is located intracellularly and its expression is restricted to placenta, this binding is unlikely to occur in biological systems or in a clinical setting”. In this reviewer’s view, galectin-14, like other galectins, including galectin-3, while being located intracellularly, as the authors correctly stated, must be present extracellular as well (although to this reviewer’s knowledge this has not been addressed in the literature). Also, although galectin-14 has more limited tissue distribution, when it is released into the extracellular space and being present in circulation, it will probably bind to many cells and tissues, like galectin-3 does. Thus, the statement “this binding is unlikely to occur in biological systems or in a clinical setting” might not be correct. As a matter of fact, one might make the same statement on galectin-3, on the basis of the comments this reviewer made above.

As you mention, we did find any report mentioning the presence of Gal-14 in the extracellular compartment. Due to the fact that this protein is located intracellularly and that its expression is restricted to placenta, as mentioned in the discussion with the accompanying recent references [(Si, Y. et al. Structure-function studies of galectin-14, an important effector molecule in embryology. FEBS J. 288, 1041-1055 (2022); Oravec, O. et al. Placental galectins regulate innate and adaptive immune responses in pregnancy. Front. Immunol. 13, 1088024 (2022)], we believe that we do not over-interpret by mentioning that this binding is unlikely to occur in biological systems or in a clinical setting.

Overall, the development of mAbs that target galectin-3 and are shown to be effective in targeting a systemic disease, in an animal model, is remarkable, especially knowing the complexity of galectin biology. As this reviewer pointed out above, galectin-3 is not a cytokine and the existing information suggests that “neutralizing antibodies” would not be effective in treatment of diseases in which this protein might play role. This might be the reason, so far there has been no biologics that are developed for treatment of diseases by targeting galectin-3, despite the fact this protein has been implicated in the pathogenesis of a number of diseases. One of the reasons might be that galectin-3 functions mainly intracellularly. In this regard, for example, a recent paper by Nguyen et al. (Galectin-3 deficiency ameliorates fibrosis and remodeling in dilated cardiomyopathy mice with enhanced Mst1 signaling Am

J Physiol-Heart Circ Physiol 316: 316: H45–H60, 2019) concluded, “Gal-3 is largely localized within cardiomyocytes rather than the ECM. It remains unknown whether currently available Gal-3 inhibitors like MCP are able to pass through cellular membrane, which would be essential for inhibiting intracellularly localized Gal-3”. It is to be noted also, the only currently FDA-approved drug targeting galectin-3 (TD-139, as the authors cited, also known as GB0139) is a small molecule that is known to have access to intracellular space. This does not mean the mAbs the authors are reporting are not expected to be developed into therapeutic antibodies. To the contrary, this reagent might be very useful to dissect the mechanism of how galectin-3 functions (and is thus worthy of being reported). However, the authors need to be extra critical and thoughtful in analyzing their data, especially with regard to the interactants, and in articulating the possible mechanism of action of their mAbs, as well as the challenging issues and possible pitfalls.

Thank you for your comment. Several of your points have been addressed in a recent review published by our group (Bouffette S., Botez I., De Ceuninck F., Targeting galectin-3 in inflammatory and fibrotic diseases. Trends Pharmacol. Sci. 2023 Aug;44(8):519-531), now added as reference 30. In particular, the paucity of biological agents against Gal-3 (but also of chemical compounds) was mainly related to the difficulty to generate successful candidates targeted to the CRD of Gal-3. This domain is physiologically suited to bind natural polysaccharide ligands, is located at the surface of the protein being solvent-exposed and contains highly polar amino acids. The ‘flat’ structure of the Gal-3 CRD is thus considered as a very difficult druggable target. In the present study, the synthetic naive scFv library used for phage display panning (based on a hyper-stable human framework scFv with side chain diversity), together with a rigorous human-mouse-human panning strategy, allowed the successful identification of D11 and E07 mAbs.

To answer your second point (*‘one of the reasons might be that galectin-3 functions mainly intracellularly’*), the results obtained with the E07 mAb in the present study, capable of reversing the expression of 87.2% of the HOCl-modulated genes demonstrated that the pathology is mostly driven by extracellular Gal-3 interactants. Although this is still a matter of debate, it is also commonly recognized that fibrosis and inflammation are mostly driven by extracellular Gal-3 (reviewed in Bouffette S., Botez I., De Ceuninck F., Targeting galectin-3 in inflammatory and fibrotic diseases. Trends Pharmacol. Sci. 2023 Aug;44(8):519-531).

Being fully in agreement with your last point (*‘to be extra critical with regard to the interactants, and the possible mechanism of action of anti Gal-3 mAbs, as well as the challenging issues and possible pitfalls’*), we have added the following statement in the discussion: *‘It should be noted that the exact mode of action of anti Gal-3 mAbs is not fully elucidated. In this respect, the relative contributions of Gal-3 interactants in the observed changes remains to be determined, which undoubtedly represent a challenging task, not only because of their high number, but also due to their likely complex networking in biological systems. To circumvent this difficulty, we investigated whether the mode of action of E07 mAb could be deciphered more at the biological pathway level.’*

Other comments:

Page 5, Line 197: “In a second approach, we aimed at identifying which genes representative of the Gal-3 interactome were differentially expressed in SSc patients --. By itself, Gal-3 expression was significantly increased by ----, and a similar increase was found at the Gal-3 protein level ---”. The authors should clarify how the Gal-3 protein level was determined. Was it its concentration in the serum?

Thank you for allowing to clarify this point. This was plasma Gal-3. This has been precised in the final manuscript. The methodology is also reported in the Methods section: Human plasma Gal-3 was quantified using the Simple Plex assay SPCKB-PS-000490 from Biotechne (Minneapolis, USA).

Page 8, Line 299, “However, treatment with E07 mAb significantly increased the ratio of alveolar macrophages, while drastically decreasing the ratio of other myeloid cells, suggesting positive outcomes on the resolution of lung damage and inflammation”. The authors might want to elaborate on how the results “suggest positive outcomes on the resolution—“.

Thank you for your comment. Although this was implicit to us, the term ‘protective’ has been added before ‘alveolar macrophages’, and ‘proinflammatory’ has been added before ‘myeloid cells’. It is also mentioned in the discussion that: ‘Only E07 mAb showed positive outcomes on immune cell populations, revealing a capacity to reduce the bronchoalveolar fluid content of NK and CD8+ T cells, and to increase

the ratio of alveolar macrophages over other myeloid cells, indicative of protective function in damaged lungs', supported by reference 56.

This reviewer mentioned that in the previous review, "Another possible mechanism of action of these mAbs is their formation of immune complexes, which can have immunosuppressive functions. In this case, the suppressive functions of Gal-3-neutralizing mAbs can only be viewed as the properties of the immune complexes and do not reflect the functions of endogenous Gal-3". In the rebuttal, the authors commented "-- the formation of immune complexes would trigger immune responses and inflammation, at the opposite of the results observed in the present study". However, it is important to point out there exists an inhibitory Fcγ receptor (FcγRIIB), which is the most broadly expressed FcγR and expressed by almost all leukocytes.

Thank you for your comment. As mentioned in the manuscript, both D11 and E07 mAbs have been engineered with a LALA mutation in their Fc part, known to abolish the interaction between mAbs and Fc receptors.

Supplemental Figure 7: Galectin-9 is not included. In the rebuttal to one of Reviewer 1's comments, the authors stated, "Recombinant Gal-9 was excluded from the analysis because of low quality control". It is critical to include galectin-9 and this issue needs to be clarified.

Thank you for your comment. Unfortunately, the commercial anti Gal-9 mAb did not show any positive signal to recombinant Gal-9, and a positive ELISA signal was found when testing recombinant Gal-9 with an irrelevant antibody. While performing additional selectivity experiments to better characterize the profile of D11 and E07 mAbs was undertaken in response to reviewer's 1 request, we emphasize that broader selectivity assays are usually conducted at later stages in our global development strategy, and that selectivity to thousands of additional antigens including Gal-9 will be assessed as part of the mAbs safety package. To be noted, Gal-9 inhibition is known to increase T cell proliferation and survival, an effect not observed with D11 and E07 mAbs in the current study.

Reviewer #4 (expert in systemic sclerosis):

The authors have responded to all of the questions I have raised. While I do not have any major concerns at this time, there are additional comments in the second point on the blood cells (neutrophilia/lymphocytes) in the SSc. I understand what the author explained, but I still do not think that the clinical significance of neutrophils/lymphocytes in SSc has already gained a broad consensus among collagen disease doctors (the papers presented in their comments appear to be very recent). Then, considering that lymphocyte and neutrophil data are presented as one of the main data in this paper, I highly recommend the authors politely explain the clinical significance of white blood cells and neutrophilia in SSc in the text with showing some recently published studies, as the authors wrote in the comments to me (I apologize if this sufficient explanation has already been posted in the manuscript somewhere). These will help the readers to understand the clinical significance of their findings related to blood cells.

We thank the reviewer for this comment. We have now added the following statement in the discussion: *'Recent studies have shown that higher neutrophil counts were associated with the severity of skin and lung involvement in SSc. A negative correlation was observed between neutrophil counts and diffusing capacity of the lungs for carbon monoxide (DLCO)⁴⁹, a marker of the severity of lung disease. Higher blood neutrophil counts and neutrophils to lymphocytes ratio were also predictive of more severe disease and increased mortality in SSc. This predictive significance of NLR for SSc severity suggested that enrichment of neutrophils and underrepresentation of lymphocytic cells reflected the pathologic immune dysregulations observed in SSc peripheral blood cells⁵⁰.*

Reviewer #5 (expert in antibody discovery and phage display):

Additional Reviewer

The manuscript describes the isolation by phage display of several antibodies to Galectin-3, with two lead antibodies identified that bind to human Galectin-3, and cross-react with mouse, cyno and dog Galectin-3. There is no cross-reactivity to other Galectins other than minimal reactivity to Galectin-14.

The authors proceed to describe a very thorough characterisation of the antibodies, both for biophysical characteristics (purity, aggregation, heat stability, binding affinities) and for efficacy in in vivo models.

I have been asked to review the rebuttal of the authors to Reviewer 1's comments, due to the unavailability of Reviewer 1. Like Reviewer 1, I am also an expert in antibody discovery and phage display. Overall, I believe that the authors have sufficiently responded to the reviewer's comments, and should be commended that they have also performed additional experiments in order to meet these comments. Specific feedback on each of the Reviewer's comments and the authors' rebuttals are given below:

Reviewer 1's first question was regarding the intended isotype of the antibodies, in relation to the authors' report that a human IgG1 isotype was used, and they asked if an effector-less isotype should be used instead. The authors have acknowledged that there was an omission in the manuscript and that the antibody isotype was in-fact a LALA variant of the IgG1 isotype which has reduced effector function. Mention of the LALA variant, with references, has been added to the manuscript.

Reviewer 1's second comment was a concern that the SPR experiments captured too much antibody and risked inaccurate results. The authors have rebutted that while the experiment could have been run at lower capture level, to avoid the risk of mass transport limitations, the sensorgrams are still showing curvature and are therefore still valid. The authors correctly state that if there was mass transport there would be linearity in the association phase. The authors have now added the sensorgrams along with the calculated parameters in the supplementary section to that readers can assess the accuracy.

Reviewer 1 then suggested to extend the specificity tests to include more antigens other than Gal-1 and Gal-7, and the authors have responded with additional assays to test Gal-2, Gal-4, Gal-8, Gal-10 and Gal-14. Although they saw some cross-reactivity with Gal-14, they explain why this wouldn't be an issue for therapeutic use due to its expression being limited to the placenta.

Reviewer 1 has asked about a toxicology assessment of using the antibodies in a clinical setting and also asked if there were any side-effects in the mouse model. The authors have only responded to the second question – confirming that there were no side-effects. Although the authors have not responded to the comment about toxicology assessment, I don't believe this is a necessary inclusion in this paper which is describing the early-stage discovery and characterisation of the antibodies. Toxicology assessments could include tissue cross-reactivity studies, and should be considered for future work.

Thank you for this comment. We confirm that toxicology assessment including TCR (tissue cross reactivity studies) and other regulatory tox studies performed in the context of mAb development are planned as future work.

Reviewer 1 has asked that the sequences of the antibodies be included, and the authors have responded that the sequences have been filed in a patent. The patent reference could be included in the manuscript to it to be found easily by readers.

Thank you for your comment. The patent reference (EP22305372) has been added in the 'Methods' and 'Competing interests' sections.

The authors have corrected a typo picked up by Reviewer 1.

Thank you for your comments and acknowledging all changes made in the revised manuscript.